# The dynamics of pattern matching in camouflaging cuttlefish

Theodosia Woo[1,4], Xitong Liang[1,3,4], Dominic A. Evans[1], Olivier Fernandez[1], Friedrich Kretschmer[1], Sam Reiter[1,2,5 ✉] & Gilles Laurent[1,5 ✉]

Many cephalopods escape detection using camouflage[1]. This behaviour relies on a visual assessment of the surroundings, on an interpretation of visual-texture statistics[2–4] and on matching these statistics using millions of skin chromatophores that are controlled by motoneurons located in the brain[5–7]. Analysis of cuttlefish images proposed that camouflage patterns are low dimensional and categorizable into three pattern classes, built from a small repertoire of components[8–11]. Behavioural experiments also indicated that, although camouflage requires vision, its execution does not require feedback[5,12,13], suggesting that motion within skin-pattern space is stereotyped and lacks the possibility of correction. Here, using quantitative methods[14], we studied camouflage in the cuttlefish *Sepia officinalis* as behavioural motion towards background matching in skin-pattern space. An analysis of hundreds of thousands of images over natural and artificial backgrounds revealed that the space of skin patterns is high-dimensional and that pattern matching is not stereotyped—each search meanders through skin-pattern space, decelerating and accelerating repeatedly before stabilizing. Chromatophores could be grouped into pattern components on the basis of their covariation during camouflaging. These components varied in shapes and sizes, and overlay one another. However, their identities varied even across transitions between identical skin-pattern pairs, indicating flexibility of implementation and absence of stereotypy. Components could also be differentiated by their sensitivity to spatial frequency. Finally, we compared camouflage to blanching, a skin-lightening reaction to threatening stimuli. Pattern motion during blanching was direct and fast, consistent with open-loop motion in low-dimensional pattern space, in contrast to that observed during camouflage.

Cephalopod camouflage consists of matching the animal's appearance to that of its substrate and typically contains two-dimensional (2D) and three-dimensional (3D) components. Although both components are technically textural[4,15,16], in this field the term 'texture' is often applied only to 3D features, caused, for example, by the contraction of skin papillae[5,17]. We studied here the 2D features of camouflage and therefore refer to them as skin patterns and to the process as pattern matching. Pattern matching does not consist of a faithful reproduction of the substrate's appearance but, rather, of the visually initiated statistical estimation and generation of that appearance[5]. These sophisticated operations are carried out instinctively[18] by the brain of animals that diverged from us more than 550 million years ago[19], well before large brains existed. The generation of 2D skin patterns relies on a motor system that controls the expansion state of up to several million pigment cells (chromatophores) embedded in the animal's skin[5], among other specialized cell types[17,20]. The expansion state of each chromatophore depends on a radial array of muscles controlling the size of a central pigment sac[21] and, therefore, on the activity of one to a few motoneurons,

the dendrites and somata of which lie in the animal's central brain[6,7]. The generation of a skin pattern therefore results from the appropriate coordination and control of tens of thousands of motoneurons by a system that interprets complex visual scenes[2,3,18].

We recently developed methods to track the instantaneous expansion state of tens of thousands of chromatophores in the behaving cuttlefish *S. officinalis*—a master of camouflage[14]. Here we improve on these techniques and report a new complementary analysis to describe quantitatively the space, dynamics and reliability of camouflage patterns and, through this, gain insights into its control system. To this end, objective measurements are critical because camouflage evolved to exploit perceptual clustering by observers, so as to fool them[22,23]. Earlier efforts to categorize camouflage patterns suggested that they belong to a small number of classes[8–11], a surprising result given the size of this system. However, a recent study using artificial backgrounds suggested that patterns, quantified as the differential expression of a set of pattern components, do not readily cluster in a low-dimensional projection[24]. Using natural and artificial 2D backgrounds (Methods

[1]Max Planck Institute for Brain Research, Frankfurt, Germany. [2]Okinawa Institute of Science and Technology Graduate University, Okinawa, Japan. [3]Present address: School of Life Sciences, Peking University, Beijing, China. [4]These authors contributed equally: Theodosia Woo, Xitong Liang. [5]These authors jointly supervised this work: Sam Reiter, Gilles Laurent. ✉e-mail: samuel.reiter@oist.jp; gilles.laurent@brain.mpg.de

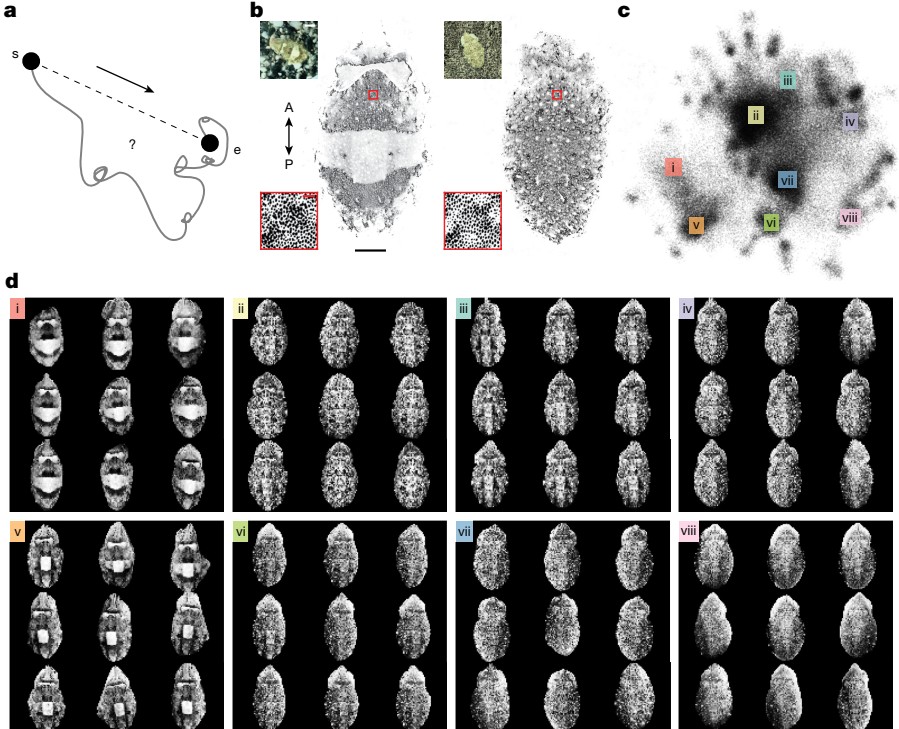

**Fig. 1 | Camouflage-pattern space. a**, A path to camouflage, from starting (s) to ending (e) skin patterns, could be direct and ballistic (dashed line) or meandering, with successive accelerations and decelerations (grey). **b**, Two examples of camouflage skin patterns, typically classified as disruptive (left) and mottled (right). Insets: magnification of the area on the mantle indicated by a red square. Bottom insets: high-resolution segmented images. Top insets: lower-resolution wide-field images. The apparatus and stimuli are shown in

Extended Data Figs. 1 and 2. Scale bars, 10 mm (main images), 20 mm (top insets) and 0.5 mm (bottom insets). A, anterior; P, posterior. **c**, Skin-pattern space was visualized using a 2D UMAP embedding of skin patterns produced by one representative animal of ten analysed. *n* = 215,577 images. Naturalistic and artificial backgrounds are shown in Extended Data Fig. 2b,c. **d**, Nine representative images were taken from each of the eight regions of skin-pattern space in **c**.

and Extended Data Figs. 1 and 2), we acquired a dense videographic sampling of the animal's generative pattern repertoire and analysed motion within skin-pattern space (Fig. 1a).

## Skin-pattern space is high-dimensional

To quantitatively assess camouflage pattern space, we presented a series of natural images to cuttlefish using printed fabric, filming cuttlefish skin at both high and low resolution. Figure 1b shows the processed high-resolution images of one cuttlefish on two backgrounds. These images were acquired using an array of 17 high-resolution cameras, synchronized with a single low-resolution camera for a global view (Fig. 1b (colour insets)). Low-resolution and high-resolution image sets were used to generate what we name skin-pattern-representation and chromatophore-representation spaces, respectively. We gathered over 200,000 low-resolution cuttlefish images from 27 h of behavioural videos of an animal on our background set (Extended Data Fig. 2b,c). We then used a pretrained neural network to parameterize skin patterns (Methods and Extended Data Figs. 1 and 2). The skin-pattern space is displayed in a 2D uniform manifold approximation and projection (UMAP) embedding (Fig. 1c), and selected patterns corresponding to different regions of that space (i–viii) are illustrated in Fig. 1d. Whereas patterns within each window seemed of a kind, their precise realizations differed. The smallest variations were due to chromatophore flickering (detected in high-resolution data; Supplementary Video 3) and small local fluctuations. However, larger variations represented different instantiations of a skin pattern (Fig. 1d). Having tested the explanatory power of linear and nonlinear methods for dimensionality estimation (Methods), we opted for a linear method—parallel analysis[25,26]. Parallel analysis reports the number of principal components (PCs)

with statistically significant explanatory power (versus a null distribution based on independently shuffled data). This approach indicated 59.4 ± 1.23 relevant dimensions (Extended Data Fig. 3a–e), although parallel analysis often underestimates the true dimensionality of a linear space above 20 dimensions[27] (Methods).

The apparent high dimensionality of camouflage patterns hinted that a reasonably close relationship might exist between backgrounds and skin patterns. As natural backgrounds themselves are difficult to parametrize simply[28], we tested this hypothesis in several ways. In the first, we used a set of 30 natural images (Extended Data Fig. 2a) and measured the correlation between background and final skin pattern along the PCs of skin-pattern space (Fig. 2a). They were significantly correlated (PC1–3, $P < 10^{-10}$) in all of the animals tested. In the second, we tested spatial frequency, a simple texture metric in image analysis. Using checkerboards as backgrounds[29] (Extended Data Fig. 2c), we observed, as others had previously[8,11,29], that a coarse sampling of spatial frequencies (half-periods, 0.04–20 cm) led to only a few clusters of correlated skin patterns. Observing that this sampling of spatial frequencies was too sparse, we added 16 checkerboard sizes in an intermediate range (Fig. 2b). A clear trend then emerged, linking monotonically background and skin-response spatial frequencies. Decomposition of chromatophore space using Leiden clustering identified groupings of chromatophores (components; Methods) of which the expansion was positively or negatively related to background spatial frequency (Fig. 2c and Extended Data Fig. 4a–c). *Sepia* camouflage can therefore smoothly and predictably transition from one pattern to another, when challenged with appropriate sets of backgrounds. This sensitivity was expressed differently over individual pattern components, resulting in an elaborate relationship between visual stimulus and skin patterning. We examined other metrics of pattern matching, as well as low-level

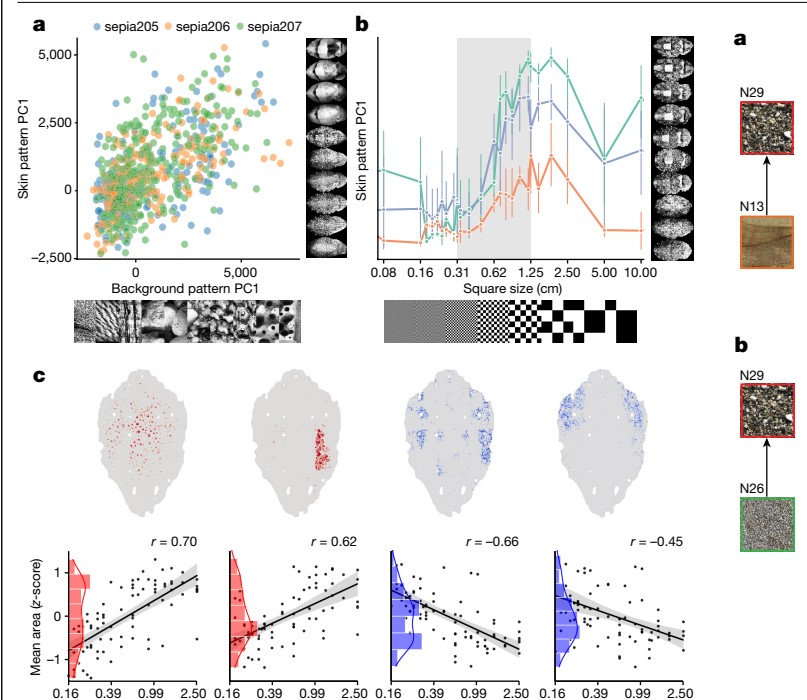

**Fig. 2 | The relationship between camouflage and natural or checkerboard backgrounds. a**, The correlation between camouflage patterns and natural background images in skin-pattern space (stimuli N0–N29; $n = 3$ animals, >8 trials per stimulus; Methods). PC1 (accounting for $17.5 \pm 0.8\%$ of the variance) shows significant stimulus–response correlation (Pearson's $r = 0.62, 0.64, 0.54$; $P < 10^{-22}$). In the three analysed animals, the first 3 (animal S205), 3 (animal S206) and 2 (animal S207) PCs are significantly correlated ($35.8 \pm 5.1\%$ variance, Pearson's $r = 0.56 \pm 0.05, P < 10^{-15}$). **b**, Skin patterns evoked by checkerboards of different spatial frequencies (square sizes, 0.04–20 cm, only 0.08–10 cm shown) reveal a monotonic gradient of intermediate responses. PC1 shows a statistically significant stimulus–response relationship within the shaded region (0.31–1.25 cm; linear regression $r^2 = 0.50 \pm 0.04, P \le 0.0001; n = 3$ animals, 4–8 trials per stimulus). In the three analysed animals, the first 4 (animal 1), 2 (animal 2) and 4 (animal 3) of the top 50 PCs are statistically significant ($r^2 = 0.40 \pm 0.03, P \le 0.0001$). The error bars show the 95% confidence intervals. **c**, Four clusters of co-varying chromatophores (components), of which the state depends positively (red) or negatively (blue) ($P \le 0.05$) on the stimulus, in one representative animal of three analysed. $n = 4$–8 trials per stimulus. Each point represents the mean steady-state response sampled at 25 Hz over 46 s. Top, cluster locations. Bottom, correlations between the mean chromatophore area and checkerboard period.

## Transitions are tortuous and intermittent

We examined the paths taken through skin-pattern space when an animal changed camouflage in response to changes between three backgrounds (N13, N26, N29; Extended Data Fig. 2b). Background changes occurred every 5–10 min (Methods). In some trials, the animal lay still during background changes (Supplementary Videos 1 and 2). In other trials, the background change induced the animal to swim to a new position, while adopting a new camouflage. Camouflage-trajectory durations were equally distributed in the two conditions (Extended Data Fig. 3d,e).

Three trajectories through skin-pattern space (pattern transitions), taken from the same animal in response to the same background

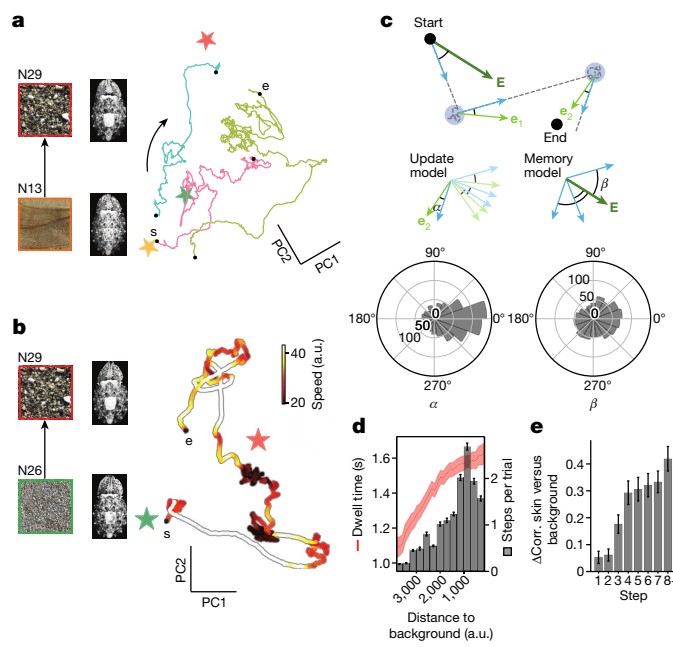

**Fig. 3 | Dynamics of camouflage transitions. a**, Exploratory trajectories (lengths, 40–126 s) in skin-pattern space (PC1, 15.2%; PC2, 13.1%) in response to the same background switches (N13 to N29). The stars represent the three background textures (N13 (orange), N26 (green) and N29 (red)). **b**, Speed profile of pattern change (colour) of one trajectory (length: 220 s) in response to background switch from N26 to N29. **c**, Test of two motion-direction models (update and memory) for motion in skin-pattern space. The dark green vectors point to the end goal from the starting point; the light green vectors point to the end goal from each intermediate slow point; the blue vectors show the actual motion direction when exiting each slow point. Data support the update model: the distribution of $\alpha$ is significantly biased to 0 (Rayleigh test, $P < 10^{-10}$), but not that for $\beta$ (Rayleigh test, $P > 0.01$). $n = 85$ trajectories, 3 animals on 3 backgrounds (N13, N26 and N29). **d**, The number of transitions (steps) between slow points per trial (grey) and the dwell time at slow points (red) increase as the skin pattern becomes more similar to the background. $n = 868$ slow points from 85 trajectories in 3 animals. Data are mean ± s.e.m. The $x$ axis shows the distance (in top two PCs) from the skin pattern at each slow point to the background pattern (bins of 285 arbitrary units (a.u.)). **e**, The correlation (corr.) between skin and background patterns increases as the number of transitions (steps) between slow points increases (Methods). Ordinate plots change (Δ) in correlation between the skin and background compared with at behaviour onset. $n = 85$ trajectories, 3 animals.

image statistics (including Fourier, Weibull[30], contrast and skewness) and their combinations. None (spatial frequency included) matched the predictive power of a high-dimensional visual texture parameterization (Extended Data Fig. 4d–f).

change, are shown as projections into the PC1–2 plane (Fig. 3a)—they were tortuous and differed across trials, typical of our results. The instantaneous velocity of pattern change (Methods) also varied along each path (Fig. 3b and Extended Data Fig. 5). In regions of greatest tortuosity (but not only there), the speed of pattern change decreased, to pick up again until a next deceleration, before eventually converging to a stable camouflage. The direction of motion at the exit of each low-velocity region pointed towards the final camouflage pattern (Fig. 3c (left); Rayleigh test, $P = 1.1 \times 10^{-54}$, $n = 85$ trials, 3 animals, 3 backgrounds; Methods), rather than in a direction parallel to the direct path linking the starting and final camouflage patterns (Fig. 3c (right)). This indicates that the animal updated its heading on its course through pattern space. The number of successive low-velocity regions increased as the animal skin approached its target pattern (Fig. 3d (grey)), as did the dwell time in each such region (Fig. 3d (red)). These results suggested that the path to a camouflage contained successive error-correction steps, as confirmed by direct measurements (Fig. 3e).

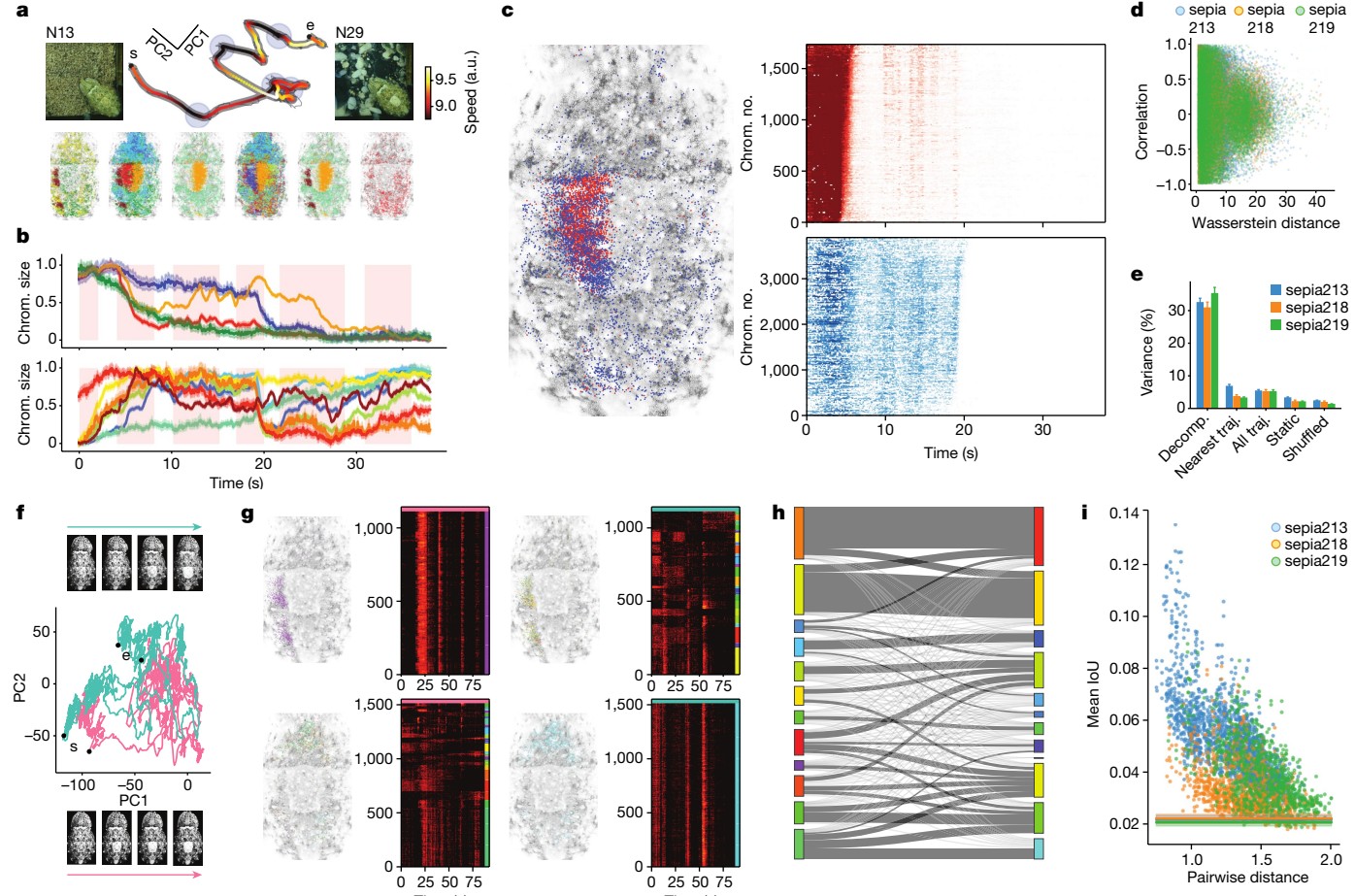

**Fig. 4 | Organization and reorganization of chromatophore groupings during pattern transitions. a**, Speed profile of a transition (background: N13 to N29) in chromatophore space (PC1–2: 15.6%, 9.8%) contained five slow points (blue, top). Groupings of chromatophores (coloured, bottom) that changed together (pattern components) during transitions between these points (1 of 3 analysed animals). **b**, Chromatophores (chrom.) in pattern components shrank (top row) or expanded transiently (bottom) during the transition in **a**. The pink shading shows the time of motion between slow points. **c**, Interdigitated groups of 1,736 (red) and 3,903 (blue) chromatophores, located in left half of the dorsal square, show different activity (right; average in **b**, top). The heat maps show the size of individual chromatophores (rows, z-scored). **d**, The correlation of activity between pattern components is not linked to their physical separation (Pearson's $r_{56,928} = -0.043$, $P = 4.34 \times 10^{-25}$; 3 animals; Wasserstein distance; Extended Data Fig. 7c–e). **e**, The variance explained (200 PCs) by the dataset in which PCs are defined. Decomp., same dataset; nearest, the most similar transition; all, all transitions, downsampled; static, activity at static patterns;

shuffled, randomized groupings. $n = 21$, 18 and 21 trajectories (traj.) from 3 animals, 3 backgrounds (N13, N26 and N29). **f**, Trajectories (the same PCs as in **a**) and images for two similar transitions (teal, 87.2 s; pink, 87.1 s; backgrounds are the same as in **a**). **g**, Chromatophores ($n = 1,123$) that co-varied in the pink trial (purple cluster, g1, left) split into many clusters in the teal trial. Chromatophores ($n = 1,532$) that co-varied in the teal trial (teal cluster, g2, right) split in the pink trial. The heat maps show the size of individual chromatophores (rows, z-scored). **h**, Pattern-component reorganization. Groupings are based on activity in the pink (left) and teal (right) trials in **f** and **g**. The line thickness is proportional to number of shared chromatophores. **i**, The fractions of chromatophores that grouped consistently across pairs of trials. The mean intersection over union (IoU) of chromatophore groupings decreases as the distance between the transition pairs increases ($n = 32.3 \pm 0.5$, $31.2 \pm 0.9$, $33.2 \pm 0.4$ clusters; 44, 32, 30 transitions; animals and backgrounds are as described in **e**; Extended Data Fig. 7f,g). The lines show shuffled groupings.

## Variable composition of camouflage patterns

We next used high-resolution imaging to identify large pattern components[18] that might reflect the higher levels of a hypothesized control hierarchy in the chromatophore system. Using Leiden community detection (Methods and Extended Data Fig. 6) over the pattern-motion segments of a camouflage change (that is, in between low-velocity regions), we identified clusters of co-varying chromatophores (Fig. 4a,b (colour coded)). The identified components were neither trivial nor did they match manually annotated components identified from static images[5,18]. The two components in Fig. 4c (red and blue, left) overlapped within the central square but differed from one another in their activities (Fig. 4c (right)), indicating that a seemingly singular feature—the dorsal square, characteristic of many disruptive patterns—is composed

of interspersed subcomponents, each capable of independent control. Generally, the degree of pairwise correlation between components was independent of their spatial overlap (Fig. 4d; Pearson's $r_{56,928} = -0.043$, $P = 4.34 \times 10^{-25}$, 3 animals). Individual components could be tight and clumpy, or loose and distributed. Our pattern decomposition had high explanatory power only if the components had been derived from the same trajectories (Fig. 4e). Performance declined when components were extracted from different trajectories or patterns (Fig. 4e and Extended Data Fig. 6), hinting that each trajectory in chromatophore space (that is, each realization of a camouflage) uses a different arrangement of components.

We examined these arrangements in more detail by tracking, at the chromatophore resolution, two camouflage-pattern trajectories from the same animal, initiated after the same background switch (from

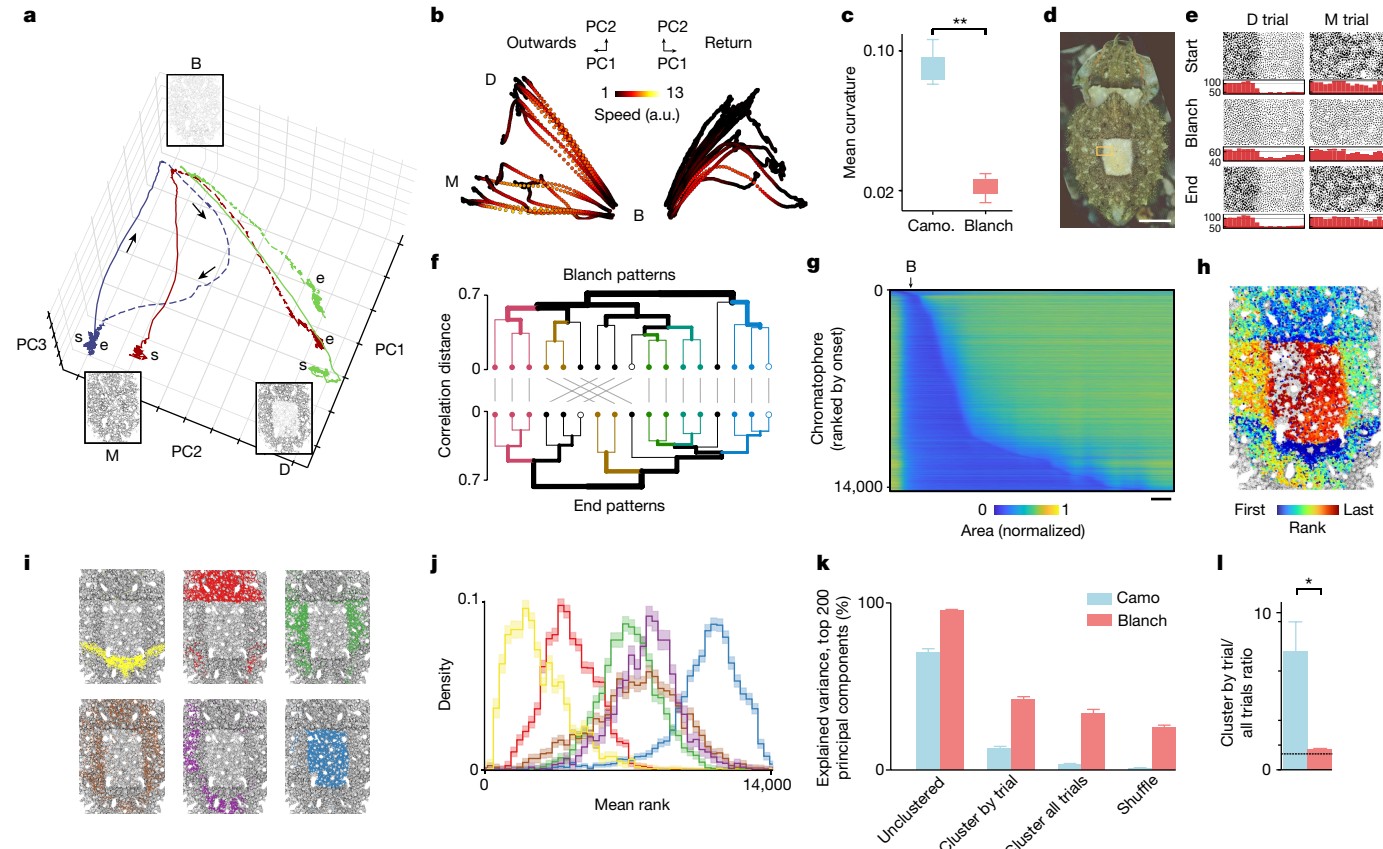

**Fig. 5 | Transition to and from the blanched state. a**, Chromatophore-space trajectories from camouflages (disruptive (D), background N29; mottled (M), background N13) to blanched (B) in response to approaching visual stimuli (sepia219 is shown throughout, except in **c**, **k** and **l**; other animals are shown in Extended Data Figs. 8–10). The solid and dashed lines show the motion to and from the blanched state, respectively. **b**, The outwards and return paths of all trajectories (*n* = 17, sepia219). Note the slower returns. The colour shows the speed of pattern change. **c**, Blanching (blanch) trajectories are straighter (lower mean curvature, 75-PC space) than camouflage (camo.) transitions (Student's *t*-test, camo. versus blanch, 3 animals each, *P* = 0.0017). **d**, Whole-mantle, 'disruptive' camouflage. Scale bar, 10 mm. **e**, Chromatophore segmentation in a cropped region (indicated by the yellow box in **d**) at the trial start, at maximum blanching and at the trial end, with corresponding marginal histograms (red). D and M, camouflage at trial onset as in **a**. Note that traces of the start and end patterns can be seen at the blanch timepoint. **f**, Hierarchical clustering of whole-mantle patterns during (top) and after (bottom) blanching reveals conserved subtrees (colours). The open circles show the trials in **e**

(D (black); M (blue); *n* = 17 trials, cophenetic correlation = 0.26; *P* = 0.015, Mantel test). **g**, Chromatophore size over time (single trial), ordered by the time of recruitment during return from blanching. Scale bar, 4 s. **h**, Chromatophores coloured by recruitment-time rank (as in **g**), suggesting a non-random, compartmentalized sequence. **i**, Leiden clustering (17 trials) reveals six components (colours). **j**, The density distribution of the chromatophore mean rank over trials for each component (on the basis of **i**), showing a reliable sequence (Kruskal–Wallis, *H* = 67.3, *P* = 3.7 × 10⁻¹³), all pairs being significantly different (post hoc multiple hierarchical permutation tests, *P* = 0.001). The shading shows the binned s.d. **k**, The explained variance after reduction to components derived from the 200-PC baseline (unclustered), the individual trajectories (cluster by trial), the whole dataset (cluster all trials) and a shuffled dataset (shuffle). *n* = 3 animals per dataset. **l**, The ratio of explained variance between cluster by trial and cluster by all trials. Blanching components are more generalizable across trials (two-tailed permutation *t*-test, camouflage versus blanching, 3 animals each, *P* = 0.0475).

N13 to N29) and that looked similar to the naked eye (top and bottom image rows). These trajectories have neighbouring starting and ending points and occupy overlapping regions in chromatophore space (pink and teal) (Fig. 4f). A set of about 1,100 chromatophores, defined by their covariation in the pink trajectory, formed one component (Fig. 4g (top left, purple)). The same chromatophores, analysed again but over the teal trajectory (right), now defined over 15 components. This analysis was repeated with a different chromatophore set, this time chosen from the teal trajectory (Fig. 4g (bottom right, cyan)). Here also, this component split into smaller ones in the other trajectory (left). The intricacy of this reorganization is summarized in Fig. 4h, in which the left and right margins represent the components generated by analysing one or the other trajectory. Subsets of chromatophores that belonged to one component joined a different component a few moments later, even (as here) when the camouflage changes were not distinguishable by eye. Across pairs of trajectories, the fraction

of chromatophores classified as belonging to the same components decreased as the distance between trajectories increased (Fig. 4i and Extended Data Fig. 7; Pearson's *r*₃,₈₈₂ = −0.619, *P* = 0.0; 3 animals). Thus, camouflage-pattern components are not stable entities and can be defined only over specific segments of activity.

## Pattern trajectories during blanching

Cephalopods often turn pale (blanch[18,31]) when they perceive a threat. These changes appear to be for conspicuous 'deimatic' display rather than camouflage, because they converge to similar patterns whatever the background (Supplementary Video 4). We therefore used blanching as a comparison for pattern-change dynamics during camouflage. Figure 5a shows 3 out of 17 blanching responses to a looming visual stimulus (Methods) in one animal displaying two different initial camouflages (Extended Data Fig. 8). In this PC projection of chromatophore

space, the three trajectories converged from their starting states to the same neighbourhood of chromatophore space, a blanched deimatic pattern (B), before typically returning to their initial camouflage (blue and green trajectories) or (only once in these 17 trials) to a different one (red). The blanching motion was fast; recovery was slower (Fig. 5b and Extended Data Fig. 8b) with gradual deceleration. We compared the curvature of camouflage and blanching trajectories in 2–200 PC dimensions; blanching paths were always more direct than those taken in camouflage, and required fewer dimensions to account for the same variance (Fig. 5c and Extended Data Fig. 9).

In the experiment in Fig. 5a,b, the animal returned to the neighbourhood of its pre-blanching state in 16 out of 17 trials, suggesting that information about its initial state remained (Extended Data Fig. 8). Indeed, although blanching trajectories converged towards the same state, they remained separable near B—their point of convergence. This is illustrated in a magnified view of the edge of the dorsal square (Fig. 5d,e): in the blanched state, the edge of the square was detectable (with reduced contrast) in the D (disruptive) trial, but not in the M (mottled) trial, consistent with their respective starting and ending patterns. The predictability of the return pattern from the blanched state is illustrated in the tanglegrams in Fig. 5f and Extended Data Fig. 8d, based on the correlations between blanched and ending patterns. The rapid chromatophore shrinking (blanching) followed by slower expansion is shown in Fig. 5g, in which chromatophores are ordered by expansion onset. By mapping the ranked chromatophores back onto the mantle, we observed that they formed reliable non-random patterns (Fig. 5h), confirmed as reliable components by community-detection clustering (Fig. 5i,j and Extended Data Fig. 10). The contrast between the repeatability of skin-pattern restoration after blanching (Fig. 5k,l) and the variability of camouflage pattern composition (Fig. 4f–i) supports the hypothesis that camouflage and blanching are under differential control.

## Discussion

Our results paint a complex picture of camouflage control. First, possibly consistent with the high resolution of chromatophore motor control[5,14], skin-pattern space is high-dimensional—the same backgrounds led to many different instantiations of a given skin pattern that are difficult to distinguish by eye. Second, camouflage smoothly covaries with ranges of natural or artificial visual textures. Skin patterns were composed of components, or chromatophore clusters, independently recruited[24], and displaying different sensitivities and responses. The *Sepia* visual system must therefore represent visual textures in some detail, probably in the optic lobes[32], and the animal's camouflage strategy is adapted to matching high-dimensional background targets. Third, the paths (in skin-pattern space) taken during a camouflage change are tortuous, intermittent—consisting of alternating pattern motion and relative stability—and not stereotyped. The number of pauses and their duration increased as convergence neared. The correlation between skin and background patterns increased as the number of pattern-motion steps increased. At each intermittent motion onset, pattern motion aimed towards the target camouflage, reflecting knowledge of the animal's instantaneous state rather than the memorization of its initial motion direction at the onset of the behaviour. Together these results suggest that camouflage relies on feedback during the approach to an adaptive pattern, more akin to correction of hand reaching movements in primates[33,34] or of tongue reaching in rodents[35] than to ballistic motion towards a memorized target. Fourth, trajectories between camouflages involve pattern components defined by chromatophore co-variation; these components could be large or small, tight or loose, suggesting a multiscale control system. However, different trajectories between similar pairs of camouflages invoked different (in numbers and composition) pattern components, suggesting control flexibility. Owing to such flexibility, describing body

pattern as the combination of around 30 fixed pattern components[24] may underestimate the complexity and dimensionality of camouflage pattern space. Identifying the smallest consistent components of camouflage patterns was not possible and will probably require very large datasets. Fifth, blanching evoked by threats to camouflaging animals retained a trace (at chromatophore resolution) of the initial camouflage. The animal usually returned to its initial state after withdrawal of the threat, through paths decomposable into reliable components. This suggests that blanching co-occurs with camouflage. Blanching represents the shrinking of chromatophores caused by the relaxation of the chromatophore muscles. By contrast, the return to a camouflage pattern requires the differential expansion of chromatophores by the contraction of those same muscles. Thus, blanching could be generated by a transient and general inhibition of the chromatophore motor drive, downstream of the camouflage control level; however, because recovery from blanching reveals components with different dynamics, this putative inhibition probably acts upstream of the motoneurons (at an intermediate level of chromatophore control) rather than directly on them.

In conclusion, camouflage in *Sepia* appears to be both very flexible and to follow non-stereotypical paths when analysed at cellular resolution. The dynamics of its output suggest the use of feedback to converge onto a chosen camouflage. Regarding where such feedback could originate from, a first possibility is proprioceptors in or around each chromatophore. Evidence for such proprioceptors around cephalopod chromatophores is lacking[5]. A second possibility is that cuttlefish use vision to assess the match between their immediate skin-patterning output and the background, for example, during each low-velocity segment in pattern-space motion. This could be tested by masking the animal's skin during camouflaging. A third possibility is efference copy of the motor command to the chromatophore array. This would require the existence of appropriate motor collaterals (not described to date), some calibration of the copy and some form of integrator, such that the copy accurately represents the true generated output. Our results will inform mechanistic studies required to understand this remarkable system.

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

## Methods

### Experimental animals

All research and animal care procedures were carried out in accordance with the institutional guidelines that are in compliance with national and international laws and policies (DIRECTIVE 2010/63/EU; German animal welfare act; FELASA guidelines). The study was approved by the appropriate animal welfare authority (E. Simon, Regierungspräsidium Darmstadt) under approval number V54-19c20/15-F126/1025. European cuttlefish *S. officinalis* were hatched from eggs collected in the English Channel and the North Atlantic and reared in a seawater system at 20 °C. The closed system contains 4,000 l of artificial seawater (ASW; Instant Ocean) with a salinity of 3.3% and pH of 8–8.5. Water quality was tested weekly and adjusted as required. Trace elements and amino acids were supplied weekly. Marine LED lights above each tank provided a 12 h–12 h light–dark cycle with gradual on- and off-sets at 07:00 and 19:00. The animals were fed live food (either *Hemimysis* spp. or small *Palaemonetes* spp.) ad libitum twice per day. Experimental animals of unknown sex, 4 to 10 months after hatching, ranging from 42 to 90 mm in mantle length, were selected for healthy appearance and calm behaviour. The animals were housed together in 120 l glass tanks with a constant water through-flow resulting in five complete water exchanges per hour. Enrichment consisted of natural fine-grained sand substrate, seaweed (*Caulerpa prolifera*), rocks of different sizes, and various natural and man-made 3D objects.

### Data acquisition

Experiments were performed in a 700 mm × 700 mm × 135 mm live-in filming tank in a separate 800 l system with its own water exchange, filtration and environmental enrichment (Extended Data Fig. 1a–c). At least 2 days before experiments, animals were moved from their home aquarium into the filming tank for acclimatization; they remained there throughout the days or weeks of filming. During experiments, a black frame was placed into the middle of the arena, restricting animals to a 400 mm × 400 mm area, keeping tank enrichment temporarily out of sight. During filming, an acrylic lid was placed onto the water surface to remove optical distortions caused by water ripples, and the arena was lit by four LED strip lights with diffusers, mounted 15 cm above the acrylic lid (SAW4 white, 698 cm length, Polytec), providing an illuminance of 3,400 lx measured at the lid centre). Background images were presented to the animal as prints on a 400-mm-wide fabric roll (210 g m$^{-2}$, 75 d.p.i.), moved over the arena floor gently with a manual crank. For experiments with natural backgrounds, a 2-mm-thick transparent acrylic sheet was placed on top of the fabric to provide extra stability for some of the animals. This increased the chance of capturing in-focus high-resolution frames during pattern transition.

We presented a set of 30 natural images with diverse visual statistics in at least five random orders (private collection; Extended Data Fig. 2a). Three background images were selected for further experiments (Extended Data Fig. 2b) on the basis of reliably eliciting distinct camouflage patterns in multiple animals. Checkerboard backgrounds were logarithmic series in three ranges of square sizes, one coarsely sampled from 0.04 to 20 cm, and two with denser sampling, from 0.63 to 2.5 cm and from 0.18 to 0.63 cm (Extended Data Fig. 2c). The coarse series was repeatedly presented in two random orders, and the finer series in one random order each. In three animals, these frozen random sequences were additionally broken in a portion of the trials by skipping through the fabric roll. The four sets of frozen random series (and their respective reverses) were not presented in a defined order. The evoked behaviours were comparable, and therefore combined for analysis.

For experiments with looming visual stimuli, the effective size of the arena was reduced to 150 × 400 mm by inserting a transparent plexiglass wall. An LCD monitor (Dell U2412M, size 52 × 32.5 cm, 50 × 22 cm visible to the animal, 60 Hz refresh rate) was suspended along the long arena edge at 40° from horizontal and maintained at a constant luminance (300 cd m$^{-2}$).

Visual stimuli were (1) manual presentations of the experimenter's hand approaching the animal at approximately 45°, stopping 20 cm away from the animal with fingers outstretched (hand looms); or (2) single presentations of a dark expanding circle on the monitor, subtending a visual angle of about 1.5° at onset, before expanding to simulate an object approaching at constant speed, according to the equation:

$$\frac{r(t)}{d} = \tan\left(\frac{\theta(t)}{2}\right) = \frac{l}{vt}$$

where $r(t)$ is the radius of the circle on the screen, $d$ is the distance between screen and animal, $\theta(t)$ is the angular size, $l$ is the half-width of an approaching object and $v$ is the approach velocity. Stimuli were generated using PsychoPy[36] and presented at six different values of $l/v$ corresponding to collision times of 2.3, 5.7, 9.2, 17.0, 25.0 and 34.4 s. The spot was located on the screen directly above the animal at a constant $x$-coordinate, with the $y$-coordinate varied to match the position of the animal's head, approximately 45° from its zenith. The minimum interstimulus interval was 2 min, and the background was changed after 1–5 stimuli. Sessions contained either one or both stimulus types; in sessions with hand looms only, the monitor was removed. In the first 30 min of session 1 for each animal, several stimuli of different $l/v$ values and hand looms were presented to find a stimulus that elicited vigorous blanching behaviour for a given animal; subsequently, this stimulus was over-represented in the stimulus order.

For high-resolution filming, 17 calibrated cameras (Basler ace acA4112-30uc) were arranged in a planar array, each recording a 3,000 × 3,000 pixel video at 25 fps. A camera's field of view was 52.4 mm × 52.4 mm (17.4 µm per pixel, 1 chromatophore occupying 54 pixels on average), with approximately 20% (20.1 ± 2.0) of pixels overlapping in neighbouring cameras. An additional low-magnification camera was mounted next to, and synchronized with, the high-resolution array, with a low-resolution field of view of 360 × 360 mm (119.8 µm per pixel, 1 pixel containing 2.4 chromatophores on average). All of the cameras were mounted onto a 2D rail system moved by stepper motors. To deal with high bandwidths, all video data were directly hardware-encoded to h264 format in real-time during the experiment. For this purpose, we used three computers running Ubuntu (v.18.04), each equipped with two graphics cards (NVIDIA Quadro M4000) providing a maximum number of eight encoding streams on each computer. We developed PylonRecorder2[37], a multi-threaded C++ acquisition software. Each instance of this software was used to retrieve the signal from one camera through USB3, encode it to h264 through libnvenc/FFmpeg and write it to one dedicated solid-state drive. A fourth computer equipped with a PCAN-USB interface (PEAK-System) running PylonRecorder2 with an additional plugin[38] was used to control and monitor the entire experiment. An Arduino Mega 2560 equipped with a CAN bus shield was used as a central hardware trigger source for all of the cameras[39]. A tracking camera was placed outside the array to view the entire experimental arena. After calibrating the tracking camera to rail positions, the experimenter could position the camera array over the animal as it moved by selecting it in the tracking view.

### Skin-pattern representation from low-resolution data

Low-resolution imaging data were processed to generate a representation of the skin pattern (Extended Data Fig. 1d). In this study, 'skin-pattern representation' refers to 2D visual textures[15,16,40]. Cuttlefish can produce different 2D textures through chromatophore activity, and also alter their 3D appearance through postural motion and contraction of papillae[17]. These 3D alterations have effects on camouflage and alter the 2D visual patterning of the cuttlefish skin. These were detected by and incorporated into our low-resolution analysis.

**Segmentation and alignment.** For each frame, the cuttlefish was segmented from the background with the Detectron2 platform[41] using a pretrained baseline model (COCO Instance Segmentation with Mask R-CNN, R50-FPN, 3x), fine-tuned with a cuttlefish training dataset. The cuttlefish images were then aligned by one of two ways: (1) aligning the long axis of an ellipse fitted to the cuttlefish segmentation mask, with the anterior-posterior orientation determined by a model similar to the one above, but trained from a different baseline (COCO Person Keypoint Detection Keypoint R-CNN, R50-FPN, 3x); or (2) maximizing image cross-correlation from one frame to another. Erroneously segmented frames were detected with a threshold on the area of the segmentation masks at 2 s.d. As a result, about 3% of all frames were removed from the subsequent analyses.

**Texture representation.** The texture representation used in our low-resolution imaging (Figs. 1–3) was the max-pooled fifth layer activations (conv5_1) of the VGG-19 neural network with weights pretrained with the ImageNet dataset in an object-recognition task, accessed through the Keras platform[42]. The choice of layer and model was informed by findings from psychophysics experiments on visual textures synthesized using Gram matrices of different layers of the model[43], and more broadly by the visual texture literature[44,45]. To our knowledge, this method has not been previously used to study cuttlefish camouflage.

The inputs to the neural network were preprocessed as follows: cuttlefish images were converted into 8-bit greyscale and histogram-equalized using OpenCV 4 (ref. 46). The background, as detected in the segmentation step, was replaced by middle grey. The images were cropped and/or padded into a square such that the cuttlefish body length was half of the image length. The cuttlefish body length was estimated for each video by taking the mean lengths of the fitted ellipses from 5–10 randomly selected frames. Finally, the images were downscaled to 224 × 224 px, and zero-centred using the VGG-19/ImageNet-compliant input preprocessing function in Keras.

The max-pooled representation used in this study is a vector of length 512, where each element is the maximum value of one of 512 feature maps (each of size 14 × 14). The Gram matrix representation mentioned above[43,47] is a vector of length 262,144, vectorized from the Gram matrix of size 512 × 512 (symmetric), where each element is the scalar product between a pair of the 512 feature maps (each vectorized to a vector of length 196). The pairwise Euclidean distances of a random sample of 300 data points computed in the max-pooled representation space showed high correlation with the same computed in the Gram matrix space, despite being summarized by relatively few parameters. (Extended Data Fig. 3a)

The 512-dimensional pattern representation was further compared using the Portilla–Simoncelli[16] visual texture model (Extended Data Figs. 3e and 4d,e). Inputs to the Portilla–Simoncelli model were preprocessed similarly, with the only differences being (1) the 224 × 224 images were padded up to 256 × 256 and (2) zero-centring was not performed. Using the standard configuration of 4 scales and orientations respectively and a neighbourhood size of 7 px, this representation consists of about 800 unique parameters.

This skin-pattern representation can be interpreted as a metric that captures textural information using 512 variables derived objectively from the visual world. It was used to construct the UMAP visualization, estimate the dimensionality of camouflage pattern space and study camouflage pattern dynamics.

**Data selection.** Full-length videos were subsampled every 10 frames to generate the entire skin-pattern space of an animal (Fig. 1c), and every 100 frames to identify the time windows of skin pattern transitions[46] (Figs. 3 and 4). Transition periods were identified at timepoints at which (1) there was a jump between the 2–4 clusters (*k*-means) in the estimated pattern space (see below); or (2) the speed of change in pattern representation exceeded 1 s.d. Before and after each selected timepoint of pattern transition, the period between the times when the speed of pattern change exceeded and then returned to the baseline (mean) was designated as a chunk of pattern transition. Two consecutive chunks were merged into one if the interval between them was less than 20 s and did not contain a background switch. After identification using subsampled data, transition periods were processed at the full frame rate (25 Hz). To study static camouflage matching (Fig. 2), the last 30–60 s (depending on the animal) of each stimulus trial (5–10 min each) were considered to be stabilized camouflage response (Extended Data Fig. 3d), and were processed for subsequent analyses.

**Visualization of skin-pattern space.** Skin-pattern space was visualized using a UMAP model (min_dist=0.8, n_neighbours=100)[48], which embeds the 512-dimensional pattern representation into two dimensions nonlinearly. The UMAP model was trained with a geometry-preserving sample of 20,000 data points selected using the geosketch algorithm[49] on the top PCs accounting for 80% of dataset variance[49]. Misoriented frames were identified with a preliminary round of clustering and withheld during the training, but later embedded (Fig. 1c). Visual inspection found the above processing to be robust against the occasional upstream misorientation. For visualization, 3 × 3 grid points were laid onto each of the selected regions in the 2D UMAP space, the nearest data point with a distance of ≤0.1 was selected and the corresponding skin pattern was plotted (Fig. 1d).

## Skin-pattern space analysis

**Dimensionality.** To estimate the dimensionality of skin-pattern space, we followed a previously proposed pipeline[27]. We first standardized features by removing mean and scaling to unit variance. We obtained an upper-bound dimensionality estimate using parallel analysis—a linear method that was found to be the most accurate among the tested linear methods for both linearly and nonlinearly embedded simulated data. We next fitted a linear (principal component analysis (PCA), 90% variance cut-off) and a nonlinear (Joint Autoencoder) model, respectively, to the data with the same number of latent dimensions as determined by parallel analysis in the previous step. For the Joint Autoencoder, we increased the size of the dense layer from 36 to 240, and the number of training epochs from 1,000 to 2,000 to reflect the increase in the number of input features (from 96 to 512). We found that a nonlinear model (variance explained, 60.0 ± 0.58) did not perform significantly better than a linear model (variance explained, 74.0 ± 0.65), suggesting that skin-pattern space in our data was largely linear (Extended Data Fig. 3a). We therefore chose parallel analysis, a linear method, to estimate the dimensionality of skin-pattern space. In brief, parallel analysis reports the number of PCs with statistically significant explanatory power compared with a null distribution defined by a parallel PCA in which the data points of each feature are independently shuffled. It should be noted that parallel analysis tends to underestimate the true dimensionality of a linear space above 20, although to a lesser extent than nonlinear methods. The above analysis was performed using 20,000 randomly sampled data points (frames) from each animal, as the estimation tends to stabilize beyond that sample size (Extended Data Fig. 3c).

**Pattern matching.** Pattern matching was studied using two stimulus sets: natural images and checkerboard series. To study visual features of natural image backgrounds (Fig. 2a and Extended Data Fig. 4d–f), the backgrounds were sampled by random selections of patches corresponding to animal size (>6 patches) near the animal from low-resolution imaging data. The background patches were then masked by the contour of the animal, processed through the same VGG-19 network for the pattern representation and further used to extract low-level statistical visual features.

Four parameters were derived from Fourier statistics[28]. The image was transformed to a power spectrum by fast Fourier transform (FFT). The 2D power spectrum was radially averaged and fitted with a line in

log–log scale. FFT-$\alpha$ and FFT-$\beta$ were the slope and intercept, respectively, of the fitted line. The third FFT parameter was the peak of the residual of the 1D power spectrum from the $1/f^{\alpha}$ fit. The fourth parameter, FFT-iso, was calculated as the ratio of the contour at 60% of the energy to a fitted isotropic ellipse in the 2D power spectrum. From the 2D power spectrum, the spatial autocorrelation was computed by inverse FFT (Wiener–Khinchin theorem). The Auto-freq parameter was the frequency at 50% of maximal auto-correlation. Two Weibull parameters, CE (contrast energy) and SC (spatial coherence), represent the width and the shape of the Weibull fits for the local contrast histogram, derived from multiple filters with different spatial scales[50]. The kurtosis and skewness of the contrast-value distribution were measured after using a first-order difference-of-Gaussians filter (size = 5) to extract contrast values.

To link visual statistics to an animal's camouflage pattern, we calculated the correlation between animal patterns and background images (Fig. 2a and Extended Data Fig. 4e). To enable the direct comparison between the body patterns and backgrounds, the 512-dimensional pattern representations of both body patterns and backgrounds (755 pairs from 3 animals) were first transformed by PCA. The first 50 PCs were then used for canonical correlation analysis to identify the linear combination of PCs best able to correlate body patterns and backgrounds. The Pearson correlation was calculated for each PC between body patterns and backgrounds, by animal. Second, different general linear models were trained to predict the camouflage patterns using individual or combinations of the visual statistics described above (Extended Data Fig. 4d). For each animal, we performed threefold cross validation (2/3 training, 1/3 test) on animal–background image pairs. For the training set, 13 general linear models were fitted separately on two visual texture representations (VGG-19 and Portilla & Simoncelli texture model), nine low-level image features, the combination of these nine features and downsized images. Model prediction residuals were calculated using the test dataset. The relative reduction of such residuals from the residual by the null model (fitted only using the intercept) were calculated as deviance reduction. The averaged deviance reduction, computed from 1,000 repetitions of fitting and cross-validation, was used to compare the performance of different visual features in predicting the animal's responses. Similarly, for the checkerboard dataset (Fig. 2b), the skin-pattern representation was first transformed by PCA on all animals collectively (50 components). Then linear regression was performed on each of the PCs per animal.

**Dynamics.** The speed of skin-pattern change was calculated as the time derivative ($dt = 0.04$ s (Figs. 3 and 4) and $dt = 0.4$ s (Extended Data Fig. 3d–f)) of the Euclidean distance of the first 200 PCs in skin-pattern space, smoothed with a 2 s window.

To compare the dynamics associated with animal locomotion and background transition (Extended Data Fig. 3d–f), the speed profiles were aligned ($t = 0$) to the peak in motion speed (where the background remained unchanged), or the trough in background correlation (corresponding to a background transition, which were occasionally followed by motion of the animal). The aligned speed profiles were resampled at 1 s intervals. Periods during which the background remained unchanged were identified as ones where the frame-to-frame image correlation remained above 0.9 for at least 10 s. Motion epochs were detected during these constant-background periods by thresholding the 2D speed of the centre-of-mass of the cuttlefish mask at 2 s.d. above the mean. A background transition is defined as a period between two constant-background periods of different background identities. The background identity of each constant-background period was determined by the following procedure: first, 4 patches of the first frame around (but not containing) the animal were combined into a composite. Then, the third-layer (conv3_1) activation of the VGG-19 model (see above) of each composite was max-pooled and then classified ($k$-means, 3 classes, with manual cluster sorting). The motion- and background transition-triggered speed profiles were built for each animal. We measured the duration of skin-pattern change starting at the time at which the motion speed (in pattern space) exceeded 10% of the peak motion speed above the baseline.

To characterize the dynamics of skin pattern change during camouflage transitions (Figs. 3 and 4), low-velocity regions of each trajectory were identified as local minima after 2 s window smoothing. Before and after each slow point, that is, during deceleration (from local speed maximum to local minimum) and acceleration (from speed local minimum to local maximum), the speed quartiles were used to separate fast (Fig. 4b (red)) from slow phases. The duration of each slow phase was defined as the dwell time at that slow point (Fig. 3d). Each step between the fast and slow phases along the trajectory was considered a step in camouflage refinement. In skin-pattern space, the distance from skin pattern to background pattern was measured in the top two PCs. For the histogram of steps per trial (Fig. 3d), the distance to the background at slow points was used as the distance for each step. The histogram was plotted for each trajectory and averaged across all trajectories ($n = 85$, from 3 animals). The dwell time was bin-averaged (bin = 55) along the distance (Fig. 3d (red curve)). For Fig. 3c, two motion-direction models were distinguished by measuring two angles, $\alpha$ and $\beta$, as an animal's skin pattern moved from a starting pattern (start), through intermediate slow points towards an eventual steady-state pattern (goal). $\alpha$ is the angle between the vector connecting point $n - 1$ to point $n$, and the vector connecting point $n - 1$ to the goal. $\beta$ is the angle between the vector connecting point $n - 1$ to point $n$, and connecting the start to the goal. In the memory model, the animal follows the initial direction from the start to the goal, resulting in both $\alpha$ and $\beta$ values of near 0. In the update model, the animal updates the direction that it must move to reach the goal in every step, resulting in $\alpha$ values of near 0, but not $\beta$. The angle was measured as the arctan of the cross product and dot product of the two vectors in the top two PCs. In Fig. 3e, we calculated after each step (that is, at each local minimum of pattern motion velocity) the correlation between the skin pattern at that time and the background, in the space defined by PCs 1–50. The difference between this instantaneous correlation and that measured at behaviour onset was then averaged across all of the trials analysed above.

### Chromatophore segmentation and tracking

High-resolution imaging data were processed to extract chromatophore population activity using a computational pipeline[14] that was modified to accommodate camera-array data, designed to film larger animals (Extended Data Fig. 1e).

**Data selection.** We filtered images over all the cameras with a difference-of-Gaussians (DoG) filter that was tuned to detect chromatophore-size features (2 and 1 s.d.). The sum of all pixels over all cameras was taken as a focus statistic. We placed a dataset-specific threshold on this statistic to select a series of in-focus time periods (chunks) for the different experiments:

Checkerboard datasets (Fig. 2): as described above, the last 30–60 s of each 5–10 min trial was selected as the stabilized camouflage response for subsequent analyses. All chunks were confirmed visually for lack of animal locomotion.

Pattern transition datasets (Figs. 3 and 4): analysis of low-resolution video (above) revealed pattern transition timepoints. The subset of these transitions that were also in focus of the high-resolution camera array (~50%,) were taken for chromatophore analysis.

Threatening stimulus datasets (Fig. 5): all trials in which animals displayed a decrease in mean chromatophore size to less than 90% of the mean starting size in the first 2 s of the trial and remained in focus, were used to calculate the Spearman $R$ for blanching time versus return speed. For all of the other analyses, we discarded low-vigour blanching responses in which the mean chromatophore size during blanching remained above 50% of the mean starting size.

**Panorama construction.** For the first timepoint in every chunk, we next determined which cameras in the array contained a view of the cuttlefish. We constructed a rough panorama view over all cameras in the array using our extrinsic camera calibration. This image was filtered using the same DoG filter as introduced above, and smoothed with a Gaussian filter (s.d., 25 pixels). We then thresholded this image, taking the largest contour as a cuttlefish mask. Images containing mask pixels were taken as the relevant cameras for that chunk.

Depending on the animal's size and position relative to the array, 1–7 cameras were typically relevant for a given chunk in our datasets. For these cameras, and taking the first image in every chunk, we next used parallax-tolerant nonlinear stitching[51] to form a single panorama view. Prominent greyscale image features were detected using SURF[52], and features were matched across cameras with overlapping field of views. An affine transform was estimated from these matched feature points, and outliers were removed using the M-estimator SAmple Consensus (MSAC) algorithm[53]. Noisy image pairs containing few (10–150 depending on the dataset) matched features were removed. We refined our initial camera extrinsic parameters using these matched features. We performed bundle adjustment using the Levenberg–Marquardt algorithm[54,55], optimizing the similarity transform between all sets of cameras. Finally, robust elastic warping[56] was performed to remove parallax effects. We saved the nonlinear transformations mapping each camera's image into the resulting panorama.

**Chromatophore segmentation.** In parallel with the above panorama construction, we segmented chromatophores on the relevant cameras (see above) over all of the images within usable chunks. In this study, we refer to the pigmented chromatophore proper as 'chromatophore', and 'chromatophore size' as the size of the pigment cell that we track. We trained convolutional neural networks (U-Net[57]) to perform semantic segmentation, classifying a cuttlefish's dark chromatophores. We used the prediction score as a probabilistic readout of the expansion state, allowing for sub-pixel resolution and improved signal-to-noise ratio. At our imaging resolution, light chromatophores[14] were not detected reliably enough for robust segmentation. Classifiers were trained on $64 \times 64$ cropped images of cuttlefish skin, manually labelled using a custom GUI (pyQt). To increase classifier robustness, we used image augmentation[58], randomly rotating, reflecting, scaling brightness, Gaussian blurring and applying piecewise affine transformations.

**Aligning segmented panorama within a chunk.** To track chromatophore expansion states, we modified our strategy[14] of fixing their pixel locations over the images in a dataset. We did this in two steps, removing animal and breathing movements to register all images within a chunk, followed by alignment over chunks, described below. For every frame in a chunk, we used our nonlinear transformations, calculated above (see the 'Panorama construction' section), to form panoramas of segmented images. During panorama construction, images were sequentially mapped into a unified reference frame[56]. Notably, during this process, we updated the panorama only at pixel locations where no image had yet been mapped to. Overlapping fields of view were thus not averaged together. This method helped us deal with errors in panorama mapping coming from slight animal movements. On the first panorama image of a chunk, we selected a random set of chromatophores distributed over the animal for tracking. For subsequent frames we used Lukas–Kanade optical flow and moving-least-squares interpolation[14] to track animal movements and align all images to the first image of a chunk.

**Stitching over chunks.** We mapped all chunks, separated in time by intervals as long as several days, into a common reference frame. We call this process 'stitching'. Coarse-to-fine grid alignment was performed as described previously[14], with four changes to increase accuracy. First, we stitched together the first segmented panorama image from each chunk, rather than the average segmented image over a chunk. Second,

we used a $128 \times 128$ pixel grid for coarse alignment, rather than $256 \times 256$ pixels. Third, we introduced a manual refinement step, in which poorly matched coarse grid points and images in cases in which registration failed were removed using a custom GUI (pyQt). Finally, grid alignment was followed by an additional alignment step: we used the SyN algorithm[59] (sigma_diff = 7, radius = 32) to register image pairs precisely, with a scale space of three levels (50, 25 and 5 pixels). The image with the lowest average reprojection error before manual refinement was selected as the dataset reference frame.

**Chromatophore extraction.** To extract chromatophore expansion states (areas) over time throughout a dataset, we mapped the first segmented panorama from all chunks into the dataset reference frame, and averaged the resulting image. We then applied the watershed transformation to this average aligned frame to determine chromatophore regions. The chromatophore expansion state was determined by mapping segmented panorama images from a chunk's reference frame into the dataset's common reference frame, and calculating the sum within every chromatophore region.

Imaging artefacts due to compression during video recording occurred about every 250 frames. Around such artefacts, detectable as periodic sharp peaks in PCA speed, 10 frames were removed and remained blank. For analyses concerning sequence of activation of individual chromatophores (Fig. 5), these artefacts were instead removed with a median filter with a 1 s window. A mask was constructed on the average aligned frame with DoG filtering to remove chromatophores with low probability of detection due to imprecise alignment. These chromatophores were generally located around papillae. This also removed regions at the curved edges of the mantle, resulting in the tracking of 76%, 75% and 70% of the pixels on the mantle in Fig. 2, and 78%, 75% and 79% in Fig. 4. In Fig. 5, a rectangular mask was used to restrict the analysis to the dorsal part of the mantle (Extended Data Fig. 10), of which 98%, 92% and 69% (in sepia218, sepia219, sepia221, respectively) of pixels remained after DoG filtering.

## Pipeline implementation

The chromatophore-tracking pipeline was implemented using OIST's Deigo and Saion HPC systems. Deigo performed all steps except for chromatophore segmentation, processing jobs in parallel on single nodes with up to 128 cores and 512 GB RAM per node. Chromatophore segmentation was performed on Saion GPU nodes using up to 32 GPUs (Nvidia V100 and P100s). Datasets for which the animal was small enough to fit in a single camera view were processed without panorama construction on CPU nodes of MPIBR's computing cluster (24–32 cores, 192–512 GB RAM per node). Data management and parallel computation was performed as described previously[14].

## Chromatophore space analysis

**Dynamics.** Speed in chromatophore space (Fig. 5 and Extended Data Figs. 5 and 8) was calculated as the time derivative ($dt = 0.04$ s) of Euclidean distance in 200 PCs, and was then smoothed by a 2 s window.

**Component analysis.** Tens of thousands of chromatophores (60,884; s.d., 679) were grouped into $32 \pm 3$ pattern components on the basis of their covariation during pattern changes. Chromatophore areas over time during pattern transitions were transformed by PCA. The top 50 PCs were then used to define co-varying chromatophores as connected nodes (n_neighbors = 10). The Leiden algorithm (resolution = 2 (Figs. 2 and 4) and 0.5 (Fig. 5)) was used to detect non-overlapping communities from the network of chromatophores[60] (scanpy package[61]). These communities of chromatophores were taken as pattern components. Components of which the mean expansion state was significantly changed (>1 s.d.) during the whole or specific phases of the pattern transition were considered to be active components (Fig. 4a).

To compare the degree of pairwise correlation of chromatophore activity between pattern components and their physical separation, multiple metrics were used to measure how two pattern components are interdigitated in space. We measured spatial overlap after spatial binning (Extended Data Fig. 7a,b), pairwise distance (Extended Data Fig. 7c) and Wasserstein distance (Fig. 4d and Extended Data Fig. 7d). To estimate how well community-based clustering could capture overall chromatophore activity (Fig. 4e), we substituted chromatophore activity for the mean activity of all chromatophores within a pattern component. This simplified chromatophore state matrix was then transformed by the same PCA model previously fit to the original chromatophore state matrix. Percentages of explained variance were compared between simplified and original trajectories in the same space defined by the first 200 PCs. To compare different component clusterings on the basis of the covariation in different pattern transitions (Fig. 4i), we paired components sharing the largest proportions of chromatophores among all possible pairs. For all transition pairs, the mean intersection over union of chromatophore groupings (also known as the Jaccard index[62]) was used as a similarity metric between different partitions. We tested the following additional metrics of clustering similarity: Wallace coefficient[63] and adjusted rand index[64] (Extended Data Fig. 7f,g). In all cases, the clustering similarity metrics were plotted against the distance between transitions pairs, which was defined by the mean pairwise distance between two transitions: for two trajectories of length $M$ and $N$, we calculated the average of the $M \times N$ matrix of distances. This distance was normalized by the s.d. of all the dataset of each animal.

For checkerboard datasets (Fig. 2c), log-linear regression was done on the mean area of the chromatophores in a given component over stimulus square sizes ranging from 0.1625 to 2.5 cm.

For threatening stimuli datasets (Fig. 5i and Extended Data Fig. 10), clustering was performed on the fast phase of the outwards and return trajectories. The fast phase was defined as the time when the instantaneous mean chromatophore speed (smoothed with a 4 s Hann time window) was above 10% of the peak speed in the respective outwards and return trajectories. Chromatophore area time-series were centred using only these fast phases before performing PCA and community detection (using the top 50 PCs). Such trimming was performed to isolate the behaviours of interest (pattern changes) in response to threatening stimuli, and thus exclude timepoints when the animal was set on a static pattern.

To compare chromatophore components between camouflage and blanching datasets (Fig. 5k–l), clustering was performed on all trials (all trials), and also for each individual trial (by trial). Explained variance for each trial and condition was computed as above, and the ratio obtained by dividing the by-trial-explained variance by the all-trial-explained variance. The shuffled dataset was generated by shuffling chromatophore-to-component assignment after by-trial clustering. We used all trials for blanching datasets (see the 'Data selection' section above) (sepia218, $n = 11$; sepia219, $n = 17$; sepia221, $n = 4$). For camouflage datasets, we selected trajectories that were close in PC space; they were selected by hierarchical clustering (Ward's linkage), performed on the pairwise Hausdorff distances (in the first two PCs) between all pairs of camouflage trajectories. We selected the largest trial cluster after cutting the hierarchy at a cophenetic distance of $d = 100$ (sepia213, $n = 7$; sepia218, $n = 8$; sepia219, $n = 10$). Analysis was performed across a range of resolution parameter values (0.25 to 4, with 0.25 steps) to check for the robustness of the results across different scales of component decompositions (Extended Data Fig. 9a). For Fig. 5k, the resolution parameter for each dataset was chosen to match the number of components extracted on individual trajectories.

**Experiments with threatening visual stimuli.** To compute correlations between start, blanched and end pattern, we first took a 10-frame (0.4 s) average of each chromatophore area around each event timepoint per trial. Hierarchical clustering was performed using correlation distance and complete linkage, and tanglegrams plotted with the R package dendextend, using the 'step2side' algorithm for untangling[65].

To identify component recruitment sequences, we first used median-filtered normalized (minimum–maximum) chromatophore area time series and, for each trial, selected chromatophores that had a minimum size increase of 0.15 during the return to camouflage. Times of chromatophore recruitment during the return trajectory were obtained after smoothing with a 1 s Hann time window and trial-wise minimum–maximum normalization. The time of recruitment was defined as the time of upwards crossing of a 0.1 threshold. Choosing the time of peak speed yielded similar results. Times of recruitments were then ranked using the average method to resolve identical ranks.

The density of recruitment ranks was analysed by histogram binning over trials (50 equally sized bins). The distribution over trials was plotted similarly by first computing, for each chromatophore, their mean rank over all of the trials. The s.d. of the bin density was computed as:

$$\sigma = \sqrt{\sum_i p_i (1 - p_i)}$$

for all chromatophores $i$, with $p_i$ the probability that a chromatophore with a normally distributed rank $N(\mu_i, \sigma_i)$ falls into that bin, where $\mu_i$ and $\sigma_i$ are the observed mean rank and s.d., respectively. A Kruskal–Wallis test was performed on component-wise chromatophore-averaged mean ranks. Post hoc multiple hierarchical permutation tests were performed with the Python package Hierarch[66] using 100 permutations, 10 bootstrap samples and Benjamini–Hochberg correction.

**Quantification of tortuosity.** To compute curvature along pattern trajectories, we reparameterized trajectories by their arclength. This enabled us to measure curvature homogeneously along the trajectories, independently of their speed. We first applied PCA (2–200 PCs tested; Extended Data Fig. 9b) on individual trials and then used the CubicSpline function (sciPy) to fit piecewise cubic polynomials to the trajectory coordinates along each principal component, using arclength as the independent variable. We then interpolated along these trajectories such that they were traversed at unit speed. The curvature at each point $n$ along these trajectories was computed as $||\mathbf{T}_n + 1 - \mathbf{T}_n||$, where $\mathbf{T}$ is the local tangent vector. For threatening visual stimulus datasets, reparameterization and curvature were computed only over the fast phases (see the 'Component analysis' section) to include only dynamics in response to the stimuli.

### Statistics and reproducibility

Unless stated otherwise, data are mean ± s.e.m. Box plots show the median and upper and lower quartiles, with whiskers extending to 1.5× the interquartile range and outliers are shown as individual points. Experiments were repeated independently several times with similar results. The numbers of repetitions were as follows:

Skin-pattern space analysis (Fig. 1 and Extended Data Fig. 3a–c) was carried out in 12 animals, 6 of which (each with at least 20 analysable trials of swift background change) were included in the analysis of background change (Extended Data Fig. 3d–f). Sample sizes were not predetermined, but chosen based on experience with similar experiments and on animal availability. Natural-image experiments (Fig. 2a and Extended Data Fig. 4d–f) were carried out in 3 animals with 8 to 12 repetitions each. Checkerboard experiments with dense sampling (Fig. 2b,c and Extended Data Fig. 4a–c) were carried out in 3 animals with 4 to 14 repetitions per stimulus in each animal. Three animals (14, 30 and 29 repetitions, respectively, for 6 types of background changes) with high-quality high-resolution data were included in the analyses of chromatophore space (Figs. 3–4 and Extended Data Figs. 5–7). For each animal, experiments were conducted in two to three experimental sessions on separate days. Threatening visual stimulation (moving

hand or looming image display) experiments (Fig. 5 and Extended Data Figs. 8–10) were carried out with 4 animals in 1 to 4 experimental sessions on separate days, yielding 11, 22, 19 and 9 trials with high-quality high-resolution data. All filming experiments were repeated by two to three experimenters, on different days, with the same animals, with comparable results.

## Reporting summary

Further information on research design is available in the Nature Portfolio Reporting Summary linked to this article.

## Data availability

Data are available from the corresponding authors on request. A small dataset is provided with the analysis code for demonstration purposes.

## Code availability

All analysis code are available at GitLab (https://doi.org/10.17617/1.93, https://doi.org/10.17617/1.94, https://doi.org/10.17617/1.95, https://doi.org/10.17617/1.96).

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

**Acknowledgements** We thank L. Jürgens, P. Dominiczak and S. Kranz for cephalopod care, aquarium design and upkeep; N. Golovyashkina, E. Northrup, G. Wexel and N. Vogt for veterinary care and assistance; F. Baier for mechanical engineering support; E. Papuschin and N. Heller electrical engineering support; F. Claudi for sharing PsychoPy scripts; and the members of the Laurent laboratory for feedback and suggestions throughout the project. We are grateful for the help and support provided by the Scientific Computing and Data Analysis section, Research Support Division at OIST. This research was funded by the Max Planck Society (to G.L.) and the Loewe-Schwerpunkt 2022 Center for Multiscale Modeling in the Life Sciences (to G.L.). S.R. was funded by the Okinawa Institute of Science and Technology and Kakenhi grants 60869155 and 20K15939; and D.A.E. by HFSP and EMBO long-term fellowships.

**Author contributions** G.L. and S.R. defined the initial project. S.R. and F.K. participated in the design of the experimental set-up. S.R., T.W., X.L., F.K., O.F. and D.A.E. wrote and adapted the code for data acquisition, processing and analysis. T.W., X.L. and D.A.E. designed and performed all experiments, with participation of O.F. for the blanching dataset. S.R. participated in the design, execution and supervision of all experiments and analyses and ran the primary image-data processing on the OIST servers. G.L. wrote the manuscript with participation of all authors, and supervised the project. T.W., X.L., D.A.E., O.F., F.K. and S.R. wrote the Methods. T.W., X.L., D.A.E. and O.F. prepared the figures with comments from all of the authors.

**Funding** Open access funding provided by Max Planck Society.

**Competing interests** The authors declare no competing interests.

**Additional information**
**Correspondence and requests for materials** should be addressed to Sam Reiter or Gilles Laurent.

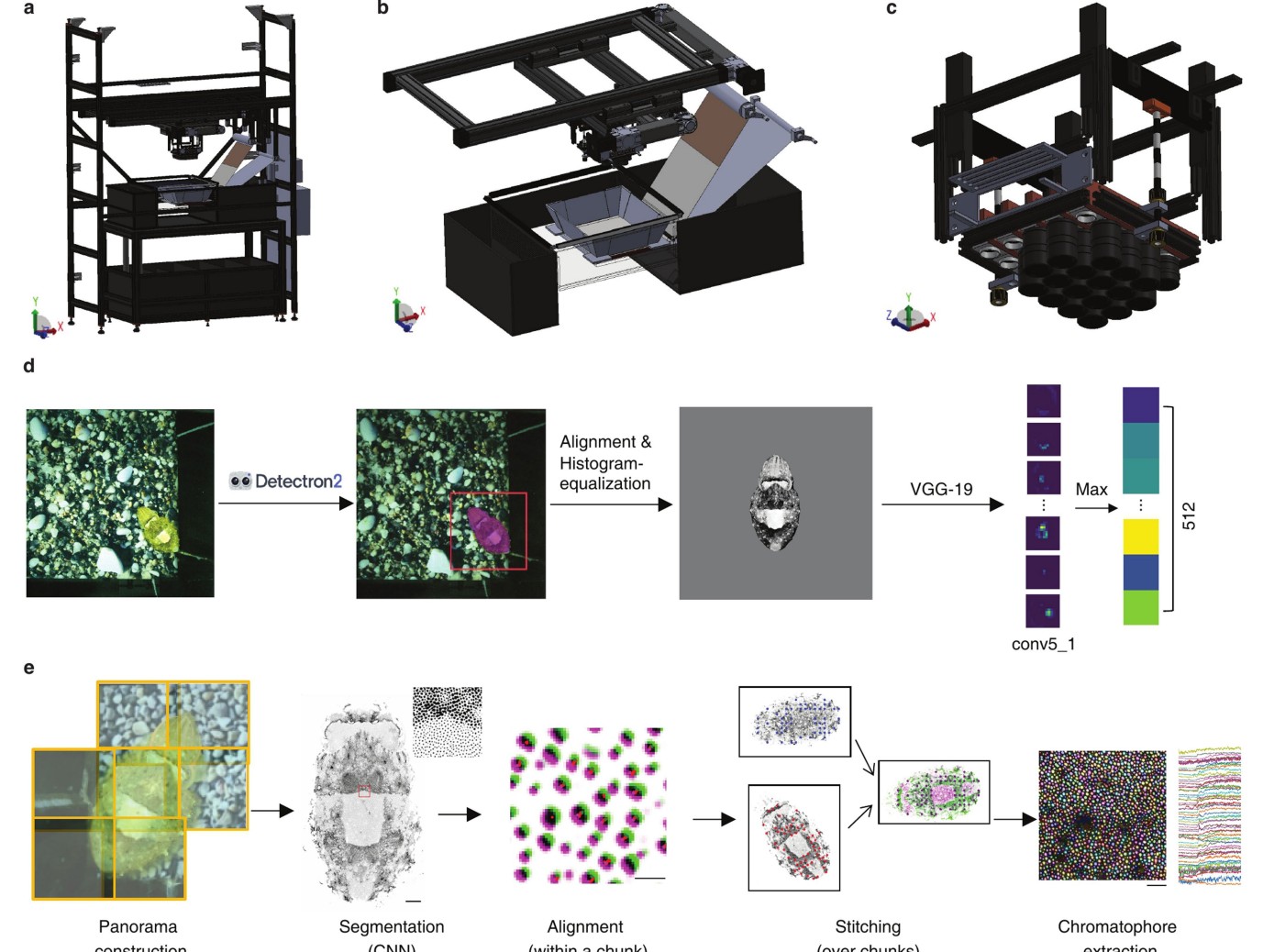

**Extended Data Fig. 1 | Schematic of experimental setup and analysis pipeline. a**. Full view of the live-in filming tank. **b**. Closer view of motorized camera array overlaying experimental arena and fabric roll with printed background images. **c**. Arrangement of 17 high-resolution filming cameras and one low-resolution camera (shorter lens, right corner). **d**. Analysis pipeline of low-resolution overview camera data (texture representation, see Methods). **e**. Analysis pipeline of high-resolution camera-array data to track single chromatophore activity (see Methods). Scale bars: Segmentation: 5,000 μm (816 μm for inset); Alignment: 200 μm; chromatophore extraction: 1,000 μm.

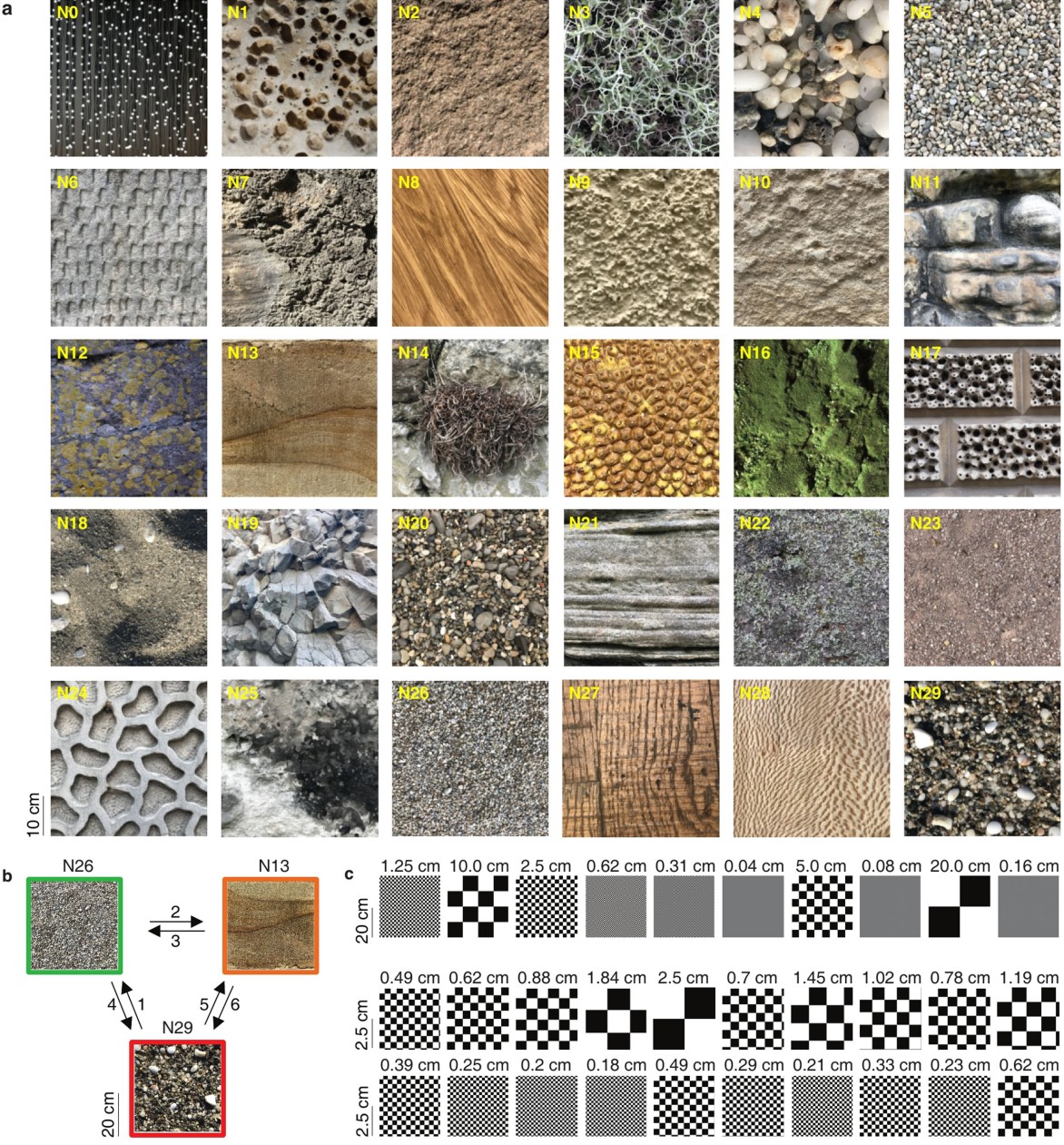

**Extended Data Fig. 2 | Camouflage-inducing background stimuli.**
**a**. 30 natural images used in Fig. 2a. **b**. Subset of 3 natural images tested on animals in Figs. 1, 3 and 4: large pebbles, small pebbles, limestone. Numbers besides the arrows denote the ordering of the stimulus presentation. **c**. Frozen random ordering of checkerboard stimuli used in Fig. 2. Top row: coarser sampling of spatial frequencies. Bottom two rows: denser sampling of spatial frequencies. Both shown in Fig. 2b.

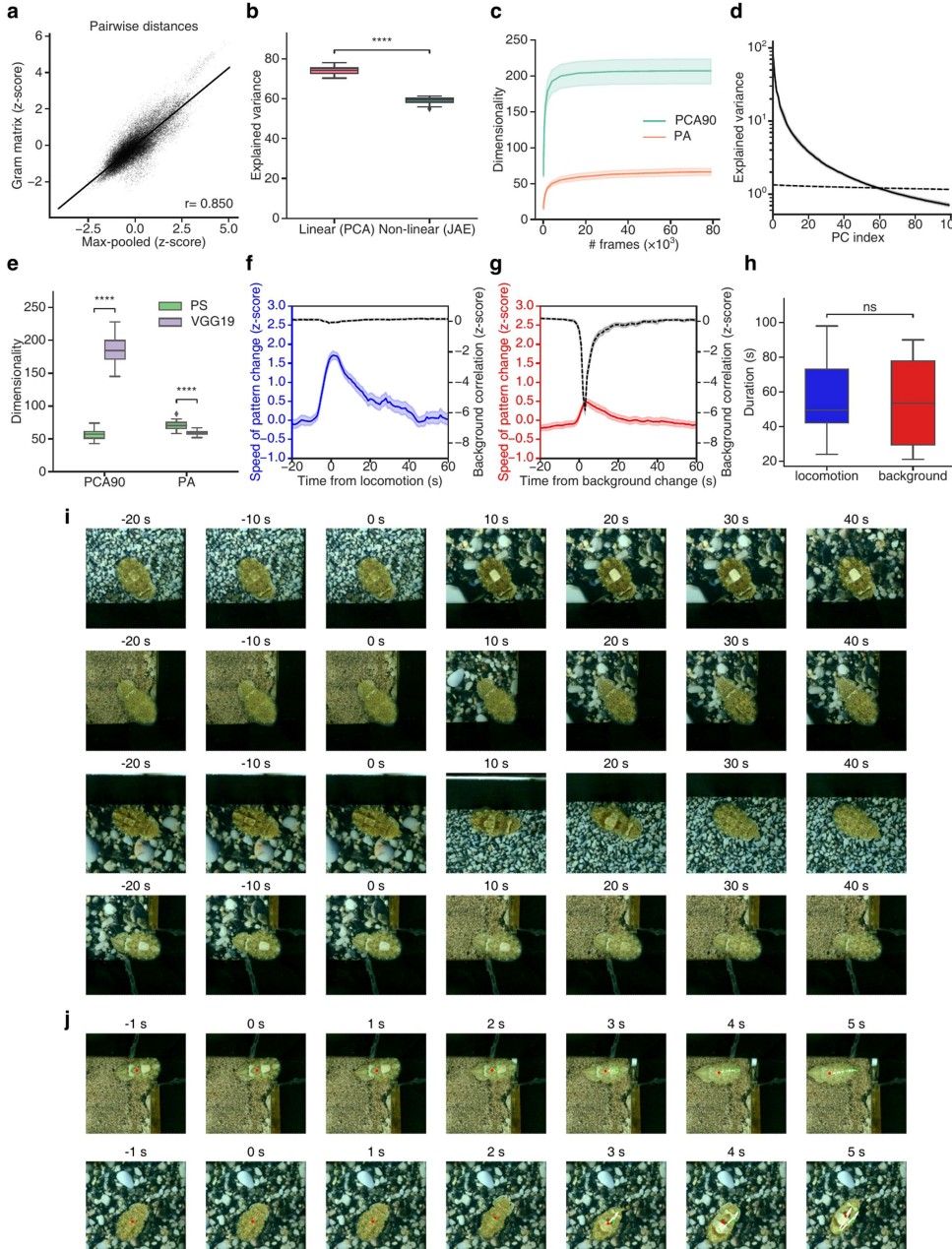

**Extended Data Fig. 3 | Changes in skin pattern associated with background change or animal movement. a**. Correlation between pairwise distances of a random subset ($n$ = 300) of skin patterns in the 512-D max-pooled texture space and in the Gram matrix space. Pearson's r(44,848) = 0.850, p < 4.94e-324, same animal as in Fig. 1. **b**. Skin pattern variance explained (mean + 95% confidence interval) by principal component analysis (PCA) and Joint Autoencoder (JAE), at 59.4 ± 1.2 latent dimensions. (**: p ≤ 0.01, two-sided paired t-test, 12 animals, Methods). **c**. Effect of number of frames on estimated dimensionality using PCA (90% variance threshold) and Parallel Analysis (PA), mean + 95% C.I., 4 animals. **d**. Dimensionality estimation using PA (see Methods). Solid line: data; dashed line: shuffled data; mean + 95% C.I., 12 animals. All datasets are downsampled to 20,000 frames. **e**. Dimensionality estimated using PCA (90% variance threshold) and PA, using the VGG19 or Portilla-Simoncelli (PS) texture model. (mean + 95% C.I.,****: p ≤ 0.0001, two-sided paired t-test, 12 animals). All datasets are downsampled to 20,000 frames. **f**. Speed of change (blue) of

skin-pattern aligned to onset of animal body motion (mean + 95% C.I.), during times when the background is unchanged (background image correlation in grey). 10 animals, 299 trials. **g**. Speed of change (red) of skin pattern aligned to onset of background-stimulus change (mean + 95% C.I.). Time of background change shown by background image correlation (grey). Y-axes standardized as in **d**. 10 animals, 474 trials. **h**. Time taken for skin pattern to return to steady state after self-initiated lomotion (**d**) and or background change (**e**). Difference is not statistically-significant (p > 0.05, two-sided Mann-Whitney-Wilcoxon test, 10 animals). **i**. Four trials (rows) illustrating clear changes in skin pattern (pattern displacement > 1.5 s.d.) after background change at t = 0. **j**. Two examples (rows) of clear changes in skin pattern (pattern displacement > 1.5 s.d.) during animal physical motion, starting at t = 0. Red dot: current position; green dots: previous positions (= red dots on earlier frames); white line: animal movement trajectory.

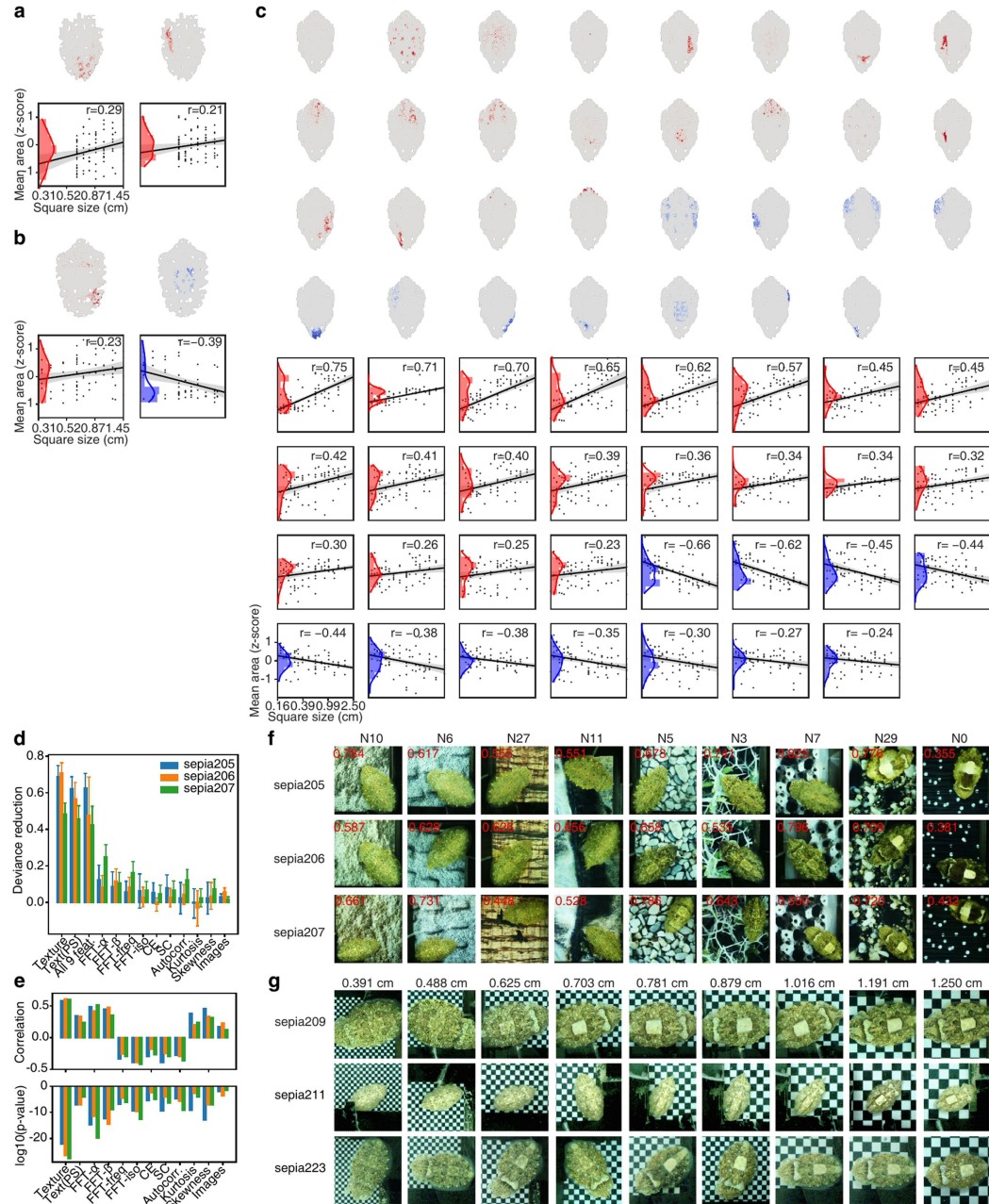

**Extended Data Fig. 4 | Camouflage pattern components and stimulus dependence. a-b**. Two illustrative clusters of co-varying chromatophores (components) demonstrating stimulus-dependent activity (4-8 trials per stimulus) from two animals not shown in Fig. 2c. Top: cluster locations; bottom: correlations between checkerboard period and mean total chromatophore area. Blue: negative, red: positive correlation (p ≤ 0.05). **c**. All clusters with p ≤ 0.05, for animal shown in Fig. 2c, plotted as in **a,b. d**. Separate GLMs to predict camouflage pattern based on: *Texture*: VGG-19 texture representation, as used in this study, *Text(PS)*: Portilla-Simoncelli (PS) texture representation, *All 9 feat.*: the combination of nine low-level visual features, individual features of background images (Methods), and *images*: the images themselves. Camouflage patterns are best predicted by the texture of backgrounds, with

the greatest reduction in deviance. The prediction performance was comparable but significantly lower when using PS texture (paired t-test, p = 2.3e-90, 6.5e-226, 8.2e-20; error bars denote s.d., see Methods) and using the combination of all nine visual features has comparable but significantly lower performance (paired t-test, p = 4.2e-285, 2.2e-195, 4.7e-127). **e.** Top: Correlation between animal's skin pattern and textures + ten image statistics of background images. Bottom: P-values (log scale) of these correlations (Methods). **f.** Representative frames along the diagonal of Fig. 2a showing responses to natural backgrounds. Numbers show correlation coefficient between background and skin patterns. **g.** Representative frames along the diagonal of Fig. 2b showing responses to checkerboard backgrounds.

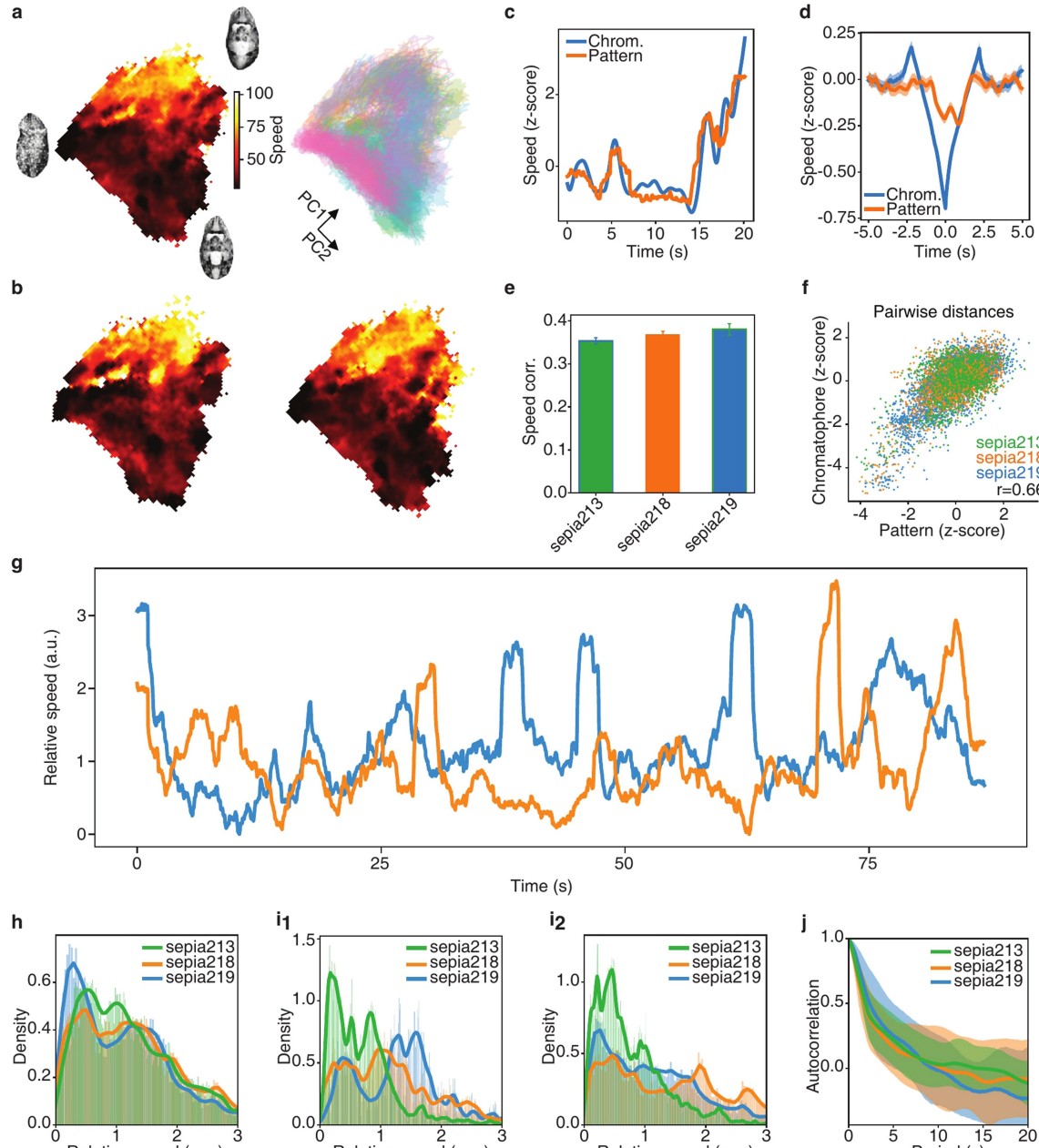

**Extended Data Fig. 5 | Dynamics of transitions between camouflage patterns. a**. Spatial distribution of pattern-change speed (left) plotted in PC1-2 of pattern space, averaged from 61 individual trajectories in one animal (right). Background images used in this figure are those in Extended Data Fig. 2b. **b**. Spatial distribution of pattern-change speed averaged from randomly selected half of trajectories (left) and the rest (right). Pearson correlation = 0.696 between two half-distributions. **c**. Pattern-change speed vs. time plot for one trajectory, illustrating the variations in speed of pattern change in pattern space (pattern, LR) and in chromatophore space (chrom., HR). **d**. Averaged speed of change in pattern space and in chromatophore space triggered from (at t = 0) speed troughs measured in chromatophore space (N = 60 trajectories, 3 animals, shading = s.e.m.). **e**. Correlation coefficient of speed measured in pattern vs. chromatophore space (N = 60 trajectories, 3 animals). **f**. Correlation between pairwise distances of subset (n = 523, 324, 261) of skin patterns in the pattern space and in the chromatophore space (Pearson's r(427,856) = 0.66, p = 0.0). **g**. Variations in time of speed of pattern change in chromatophore space (two different trajectories, same animal). To allow comparisons across trajectories, the minimal speed within each trajectory was subtracted. Note the large speed variations. **h**. Distributions of pattern-change speed in chromatophore space are multimodal (all trajectories, three animals). Three datasets (animals: sepia219, sepia218, sepia213) rejected the unimodal test with p = 0.016, 0.008, 0.005. **i**. Pattern-change speed distributions in **h** are split and plotted separately for 0 < t < 54s (**i1**) and t > 54s (**i2**) after background switch (t = 0). t = 54s chosen as the average duration of fast pattern changes (see Extended Data Fig. 3f). All distributions rejected the unimodal test with p < 0.05. **j**. Autocorrelation of speed vs. time (line: mean; shading: s.d.), from 72 trajectories in three animals. Note absence of periodicity in the autocorrelation indicating absence of regularity in speed variations.

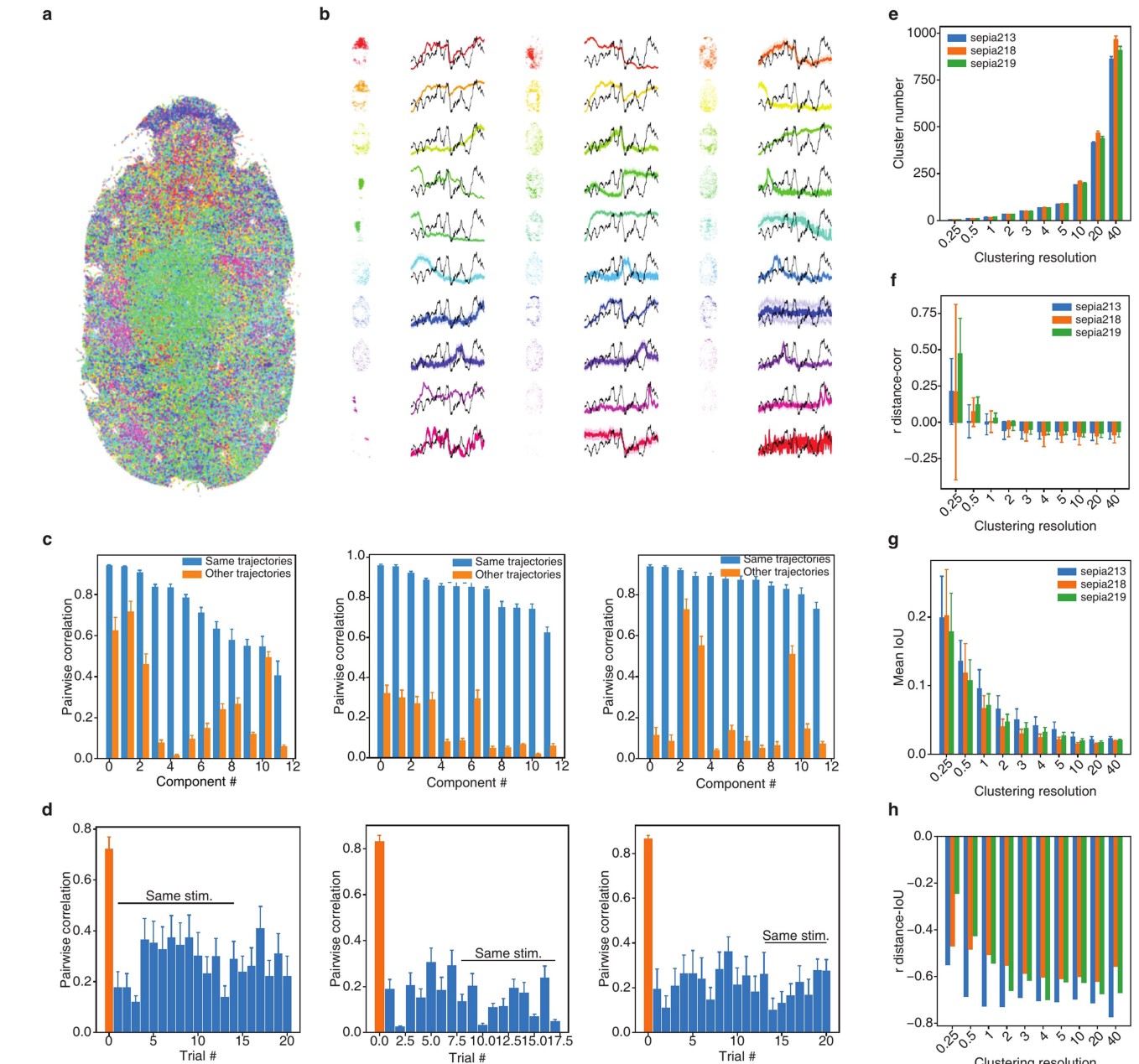

**Extended Data Fig. 6 | Chromatophore component clustering. a.** Clustering of chromatophores based on the activity in the trajectory in Fig. 4a (Leiden community detection, N = 30 clusters). **b.** Spatial distribution of chromatophores in each cluster (insets at left) and their average area change over time in this transition. Black trace common to all panels is the animal's speed of pattern change. **c.** Averaged pairwise correlation of chromatophore activity within pattern components. The top 12 most variable components are plotted. Blue, correlation of chromatophore activity during the trajectory in which pattern components are defined. Orange, average correlation of activity of the same chromatophores during other trajectories (N = 21, 18, 21 trials for sepia213, left; sepia218, middle; sepia219, right). **d.** Correlation as in **c**, averaged over components 1-12 instead of trials. Orange: average correlation in the selected

camouflage transition. Blue: correlation of same clustering during other trajectories. (grouped by stimulation types; N = 21, 18, 21 trials for sepia213, left; sepia218, middle; sepia219, right). **e.** Number of clusters at different clustering resolutions. Clustering was based on trial-specific activity. (N = 21, 18, 21 trials for sepia213, sepia218, sepia219). **f.** Absence of correlation between Wasserstein distance and group activity correlation (Mean ± s.e.m. of Pearson's r; as in Fig. 4d) at different clustering resolutions. **g.** Mean IoU (Intersection over Union of chromatophore groupings) at different clustering resolutions (Mean ± s.e.m.). Higher clustering resolution does not result in higher clustering stability. **h.** Correlation coefficient of the distance between trajectories and mean IoU (Mean ± s.e.m. of Pearson's r; as in Fig. 4i), at different clustering resolutions.

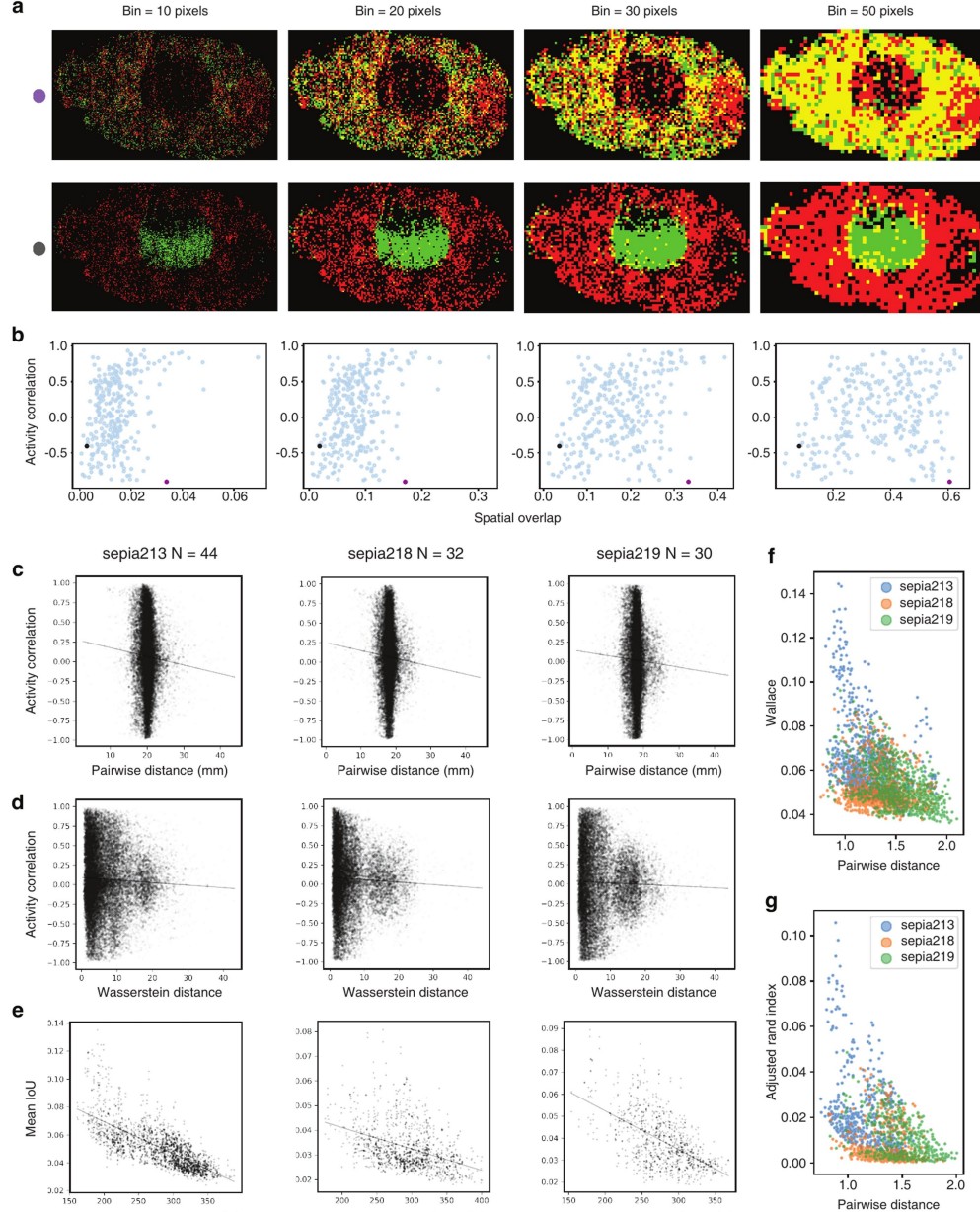

**Extended Data Fig. 7 | Spatial distribution and stability of pattern components. a**. Examples of measurement of spatial overlap between pattern components (green and red; overlap in yellow): each column stands for a correlation coefficient of spatial density using a different bin size. Upper row: components showing high spatial overlap; lower row: components showing low spatial overlap. **b**. For pairs of pattern components, correlation of activity and spatial overlap do not correlate (N = 435 component pairs defined in single trajectories between camouflage patterns). Purple dot: pair in upper row in a; black dot: pair in lower row in a. **c**. Absence of correlation between the pairwise distance between pattern components (measured as averaged physical distance between pairs of chromatophores) and the correlation of their mean activities during repeated transitions. N indicates the number of repeated

trials (trajectories) included in each analysis (Same N values for d-e). High dot density near 20 mm is explained by the half-width of the animals and the left-right symmetry of the pattern component pairs. **d**. Measurements as in **c**, using Wasserstein distance as a metric for distance between pattern components. **e**. Relationship between the dissimilarity of two transitions (measured as pairwise distance in 200 PCs, Methods) and the proportion of chromatophores that remain in the same component across those two transitions. **f**. Comparison as in **e**, using Wallace distance as a metric for clustering similarity. Distance was normalized by the s.d. of all dataset of each animal. Pearson's r = −0.382, p = 2.9e-100. **g**. Comparison as in **e**, using Adjusted-Rand-Index as a metric for clustering similarity. Pearson's r = −0.436, p = 4.3e-133.

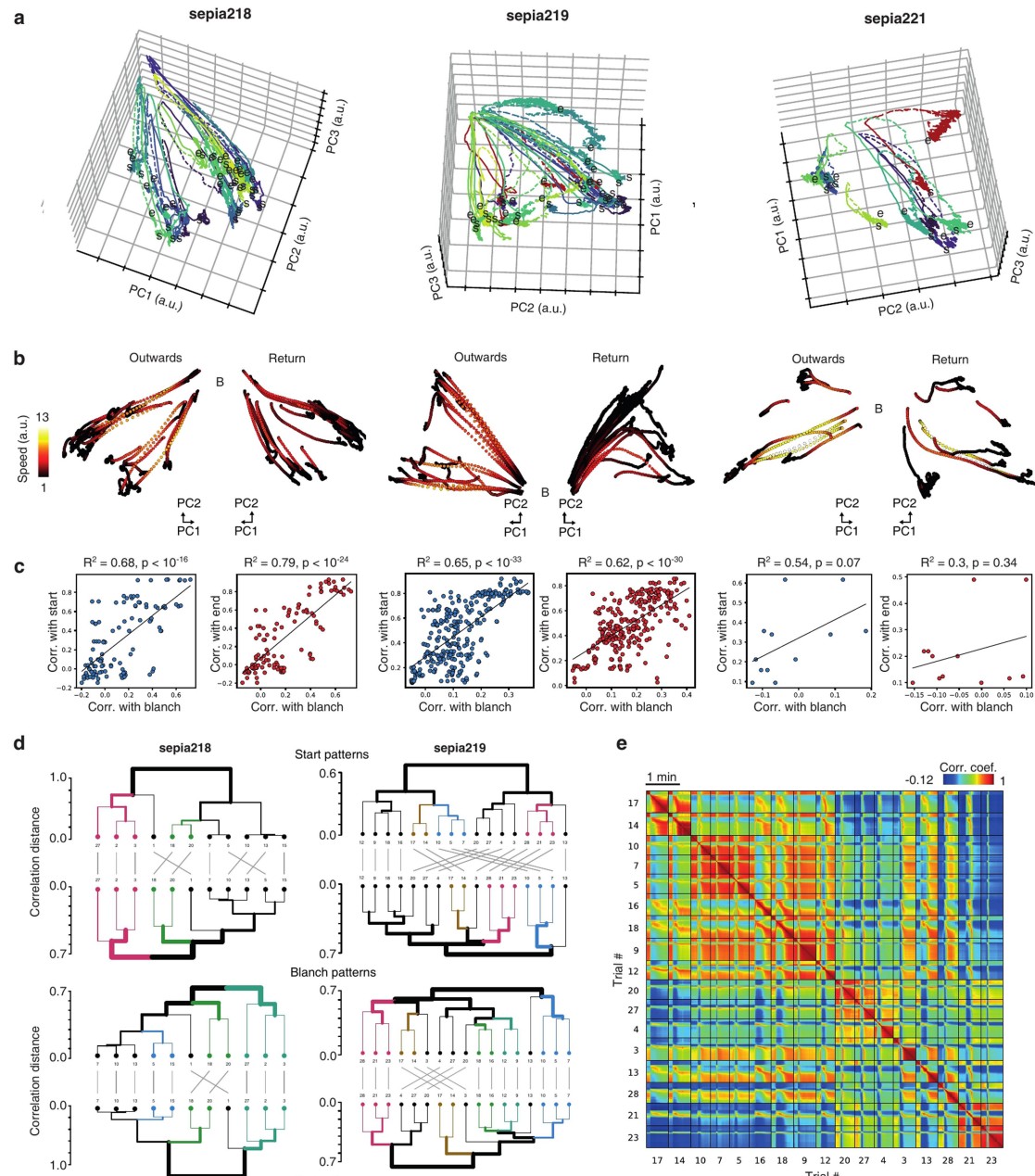

**Extended Data Fig. 8 | Transitions and correlations between camouflage and blanching patterns. a**. Trajectories corresponding to all blanching trials (evoked by threatening stimulus) in three animals (sepia218: n = 22, sepia219: n = 19, sepia221: n = 6), projected in the space defined by their first three principal components. Trajectories are coloured by trial number, and the rare trials showing different start and end patterns are highlighted in magenta. Solid lines: outward trajectories towards blanched state; dashed lines: return to camouflage patterns. **b**. Same as in **a** but shown in only their two first PCs and split at their peak blanching point to separate blanching and return trajectories. Colour represents the instantaneous speed in space defined by the first 200 PCs (scale as in Fig. 5b). **c**. The correlation between the starting (or ending) pattern of a trial and the blanched pattern reached in another trial predicts the correlation between the starting (or ending) patterns of both trials (both pattern correlations are positively correlated). This suggests that blanched patterns carry information about the starting (and ending) camouflage pattern of a same trial. In addition (data not shown), the mean correlation coefficient (z-scored) between starting and blanching patterns of the *same* trial is significantly higher than the mean correlation coefficient between starting

and blanching patterns of *different* trials, suggesting that blanching patterns depend on the camouflage pattern preceding blanching (mean ± s.e.m.: 1.29 ± 0.16 vs. −0.00 ± 0.06, P = 0.013, two-sided paired t-test, N = 3 animals). This is also true for blanching and end patterns (0.91 ± 0.19 vs −0.51 ± 0.06, P = 0.0081, two sided paired t-test, N = 3 animals). **d**. Aligned tanglegrams to visualize hierarchical clustering performed on start, blanching and end patterns at chromatophore resolution (mean of 10 frames per chunk) for two animals, showing that similarities that exist between patterns during camouflage are conserved during blanching (left; sepia218, start-to-blanch cophenetic corr. = 0.63, P = 0.0046, Mantel test, blanch-to-end cophenetic corr. = 0.80, P = 2e-04, Mantel test, right; sepia219, start-to-blanch cophenetic corr. = 0.26, P = 0.0047, Mantel test; blanch-to-end cophenetic corr. = 0.26, P = 0.015, Mantel test). Colours denote common subtrees in each dendrogram pair; numerical leaf labels denote trial ID. **e**. Heatmap of pairwise correlation coefficient between all frames of all strong blanching trials for sepia219 (n = 17 trials). Trials are sorted by pairwise correlation coefficient between their respective start and end patterns.

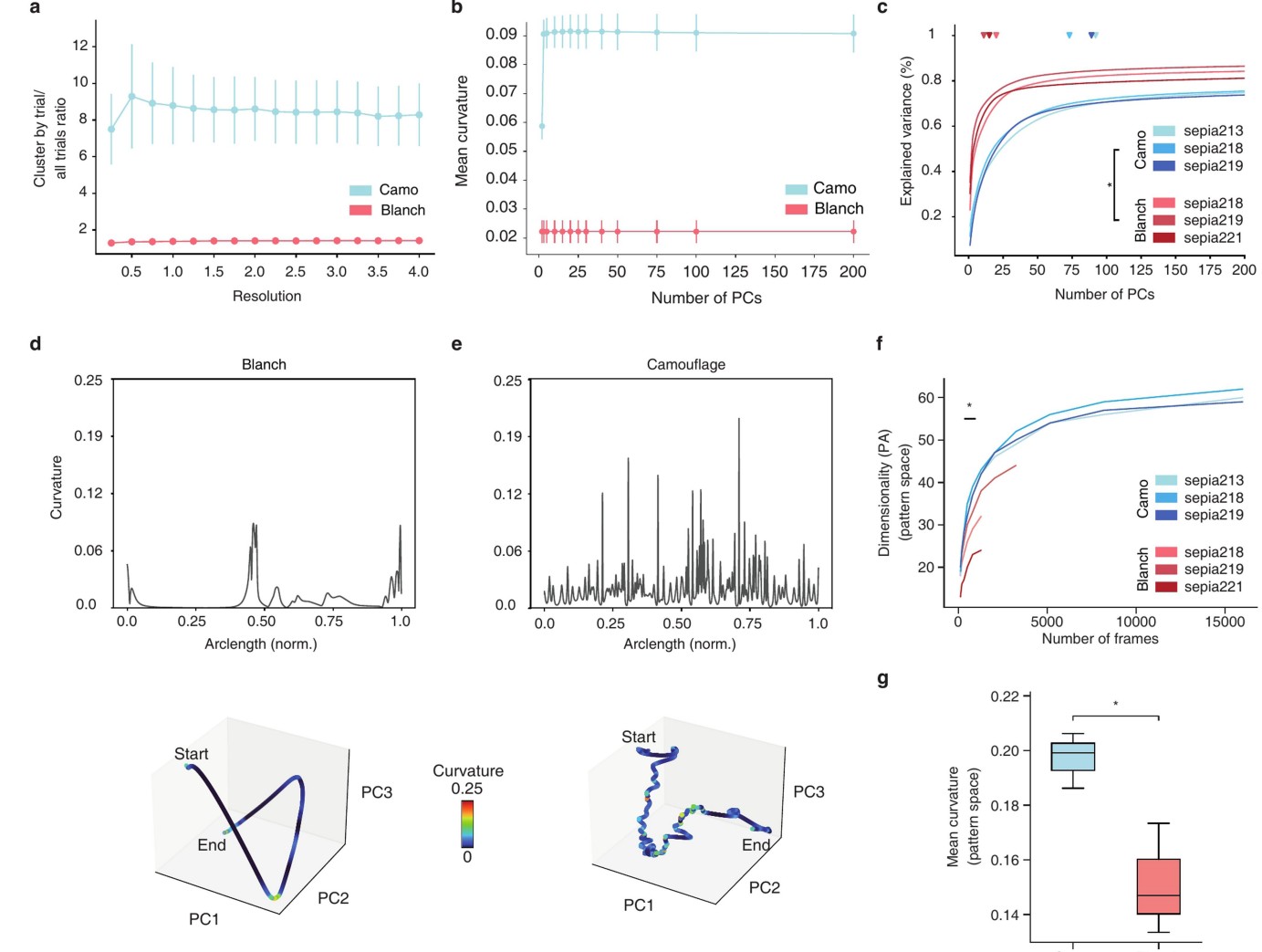

**Extended Data Fig. 9 | Pattern-change trajectories are more tortuous for camouflage than for blanching transitions. a**. Mean ratio (± s.e.m.) of explained variance when using sets of components obtained from Leiden community detection over a range of resolution parameter values. The ratio is between values obtained by using data from individual trials and those obtained from all trials. This ratio is computed over 6 different datasets (camo: mean of N = 3 camouflaging animals, blanch: mean of N = 3 blanching animals). **b**. Mean curvature (± s.e.m.) over all trials for all four datasets using 2 to 200 principal components (camo: mean of N = 3 animals, blanch: mean of N = 3 animals). **c**. Proportion of explained variance as a function of the number of principal components for looming and camouflage datasets. Markers point to the number of components to explain 70% of the variance of the datasets illustrating the simpler dynamics during blanching (two-sided t-test: camo (N = 3 animals) vs. blanch (N = 3 animals), p = 0.0004). All datasets were homogeneously downsampled to 15,000 frames. **d-e**. Top: example traces of curvature computed along arc-length reparameterized trajectories (see Methods). Bottom: example trajectories in PC1-3 space, colour-coded by curvature. **f**. Dimensionality estimated in pattern space using Parallel Analysis (PA). *: p ≤ 0.05, two-sided t-test, N = 3 and 3 animals. **g**. Mean curvature computed along arc length reparameterized trajectories in skin pattern space projected to the top 50 PCs. Two-sided t-test, camo vs. blanch (N = 3 and 3 animals), p = 0.0247.

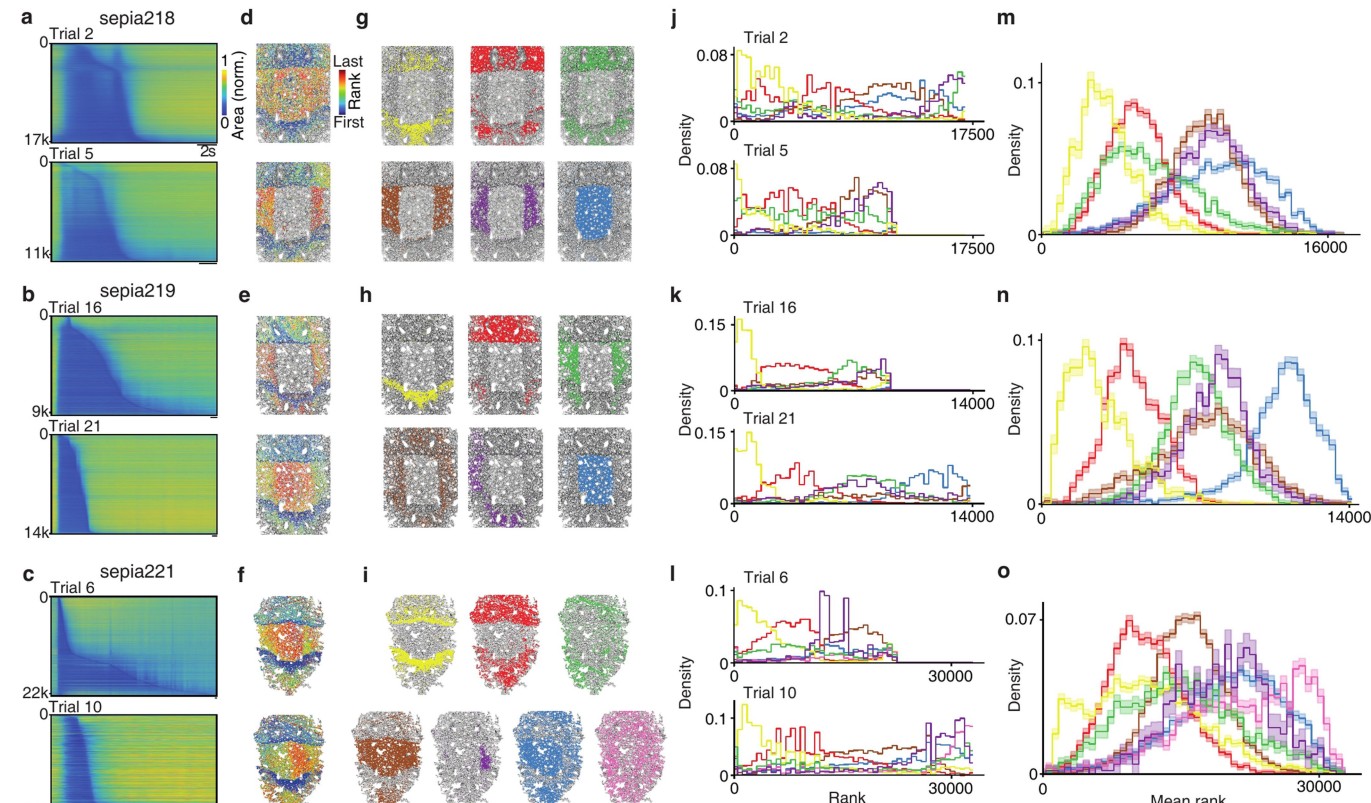

**Extended Data Fig. 10 | Return from blanching to camouflage: identification of pattern components. a-c**. Heatmaps of chromatophore size (normalized min-to-max expansion in colour scale) during blanching trials (looming stimulus) in three animals (**a-c**). Chromatophore size was min-max normalized for each chromatophore across all trials, and chromatophores are ranked by time of threshold-crossing during the return from blanching. Only chromatophores whose size change was significant (Methods) are displayed. Horizontal scale bars: 2 sec. **d-f**. Mantle of cuttlefish in **a-c**, with chromatophores colour-coded by rank of recruitment time during the return phase in the trials in a-c. **g-i**. Components identified from Leiden community detection plotted on the mean pattern of each animal. **g**: sepia218; 19,313 chromatophores; 6 components. **h**: sepia219; 15,468 chromatophores; 6 components. **i**: sepia221; 37,238 chromatophores; 7 components. **j-l**. Chromatophore rank distribution by component shown in **g-i** (for the two trials in **a-c**). **m-o**. Chromatophore mean-rank distribution by component from **g-i** (from all trials). Shading: bin s.d. (see Methods). **m**: sepia218; n = 11 trials. **n**: sepia219; n = 17 trials. **o**: sepia221; n = 4 trials. Kruskal-Wallis test on the component chromatophore-averaged mean rank indicates that at least one distribution is significantly different (sepia218: H = 38.1, P = 3.5x10$^{-7}$, sepia219: H = 67.3, P = 3.7x10$^{-13}$, sepia221: H = 18.5, P = 0.005). Post-hoc multiple hierarchical permutation tests on the full nested datasets for each pair of components indicate that most component distributions are significantly different (sepia218: all pairs P < 0.005; sepia219: all pairs P = 0.001; sepia221: P = 0.12, 0.16 for components 1 vs 6, 2 vs 3 respectively, other pairs P ≤ 0.005).

# Reporting Summary

## Statistics

For all statistical analyses, confirm that the following items are present in the figure legend, table legend, main text, or Methods section.

| n/a | Confirmed | |
|---|---|---|
| ☐ | ☒ | The exact sample size ($n$) for each experimental group/condition, given as a discrete number and unit of measurement |
| ☐ | ☒ | A statement on whether measurements were taken from distinct samples or whether the same sample was measured repeatedly |
| ☐ | ☒ | The statistical test(s) used AND whether they are one- or two-sided<br>*Only common tests should be described solely by name; describe more complex techniques in the Methods section.* |
| ☒ | ☐ | A description of all covariates tested |
| ☐ | ☒ | A description of any assumptions or corrections, such as tests of normality and adjustment for multiple comparisons |
| ☐ | ☒ | A full description of the statistical parameters including central tendency (e.g. means) or other basic estimates (e.g. regression coefficient) AND variation (e.g. standard deviation) or associated estimates of uncertainty (e.g. confidence intervals) |
| ☐ | ☒ | For null hypothesis testing, the test statistic (e.g. $F$, $t$, $r$) with confidence intervals, effect sizes, degrees of freedom and $P$ value noted<br>*Give P values as exact values whenever suitable.* |
| ☒ | ☐ | For Bayesian analysis, information on the choice of priors and Markov chain Monte Carlo settings |
| ☐ | ☒ | For hierarchical and complex designs, identification of the appropriate level for tests and full reporting of outcomes |
| ☒ | ☐ | Estimates of effect sizes (e.g. Cohen's $d$, Pearson's $r$), indicating how they were calculated |

*Our web collection on statistics for biologists contains articles on many of the points above.*

## Software and code

Policy information about availability of computer code

| Data collection | *Provide a description of all commercial, open source and custom code used to collect the data in this study, specifying the version used OR state that no software was used.* |
|---|---|
| Data analysis | The chromatophore tracking pipeline was an extension from the published code in Reiter et al. (2018), running in the Python (3.6-3.8) and MATLAB (2018a-2019a). Other analyses were performed in Python (3.6-3.8) and R (4.1.1), with specific packages detailed under Methods. |

For manuscripts utilizing custom algorithms or software that are central to the research but not yet described in published literature, software must be made available to editors and reviewers. We strongly encourage code deposition in a community repository (e.g. GitHub). See the Nature Portfolio guidelines for submitting code & software for further information.

## Data

Policy information about availability of data

All manuscripts must include a data availability statement. This statement should provide the following information, where applicable:

- Accession codes, unique identifiers, or web links for publicly available datasets
- A description of any restrictions on data availability
- For clinical datasets or third party data, please ensure that the statement adheres to our policy

Data will be available upon request.

# Field-specific reporting

Please select the one below that is the best fit for your research. If you are not sure, read the appropriate sections before making your selection.

☒ Life sciences ☐ Behavioural & social sciences ☐ Ecological, evolutionary & environmental sciences

For a reference copy of the document with all sections, see nature.com/documents/nr-reporting-summary-flat.pdf

# Life sciences study design

All studies must disclose on these points even when the disclosure is negative.

| | |
|---|---|
| Sample size | Sample sizes was not predetermined. Data were collected over days to weeks from each animal, from 15 animals. Sample sizes were chosen based on previous experience with these experiments, within the limit of available animals. |
| Data exclusions | Data selection is detailed under Texture representation from overview camera data > III. Data selection, and Chromatophore segmentation and tracking > I. Data selection. In brief, video segments relevant to individual behaviours studied were extracted from continuous recordings using a set of selection criteria, including a focus statistic that excludes data with motion blur. |
| Replication | All experiments were repeated by 2-3 experimenters independently on different days with the same animals, leading to comparable results. |
| Randomization | Randomization was not relevant to our study. Animals were allocated based on availability, healthy appearance, and calm behaviour. No comparisons were made between groups. |
| Blinding | Blinding was not relevant to our study. Animals were allocated based on availability, healthy appearance, and calm behaviour. No comparisons were made between groups. Texture representation and chromatophore tracking were automated without information from the stimulus (background masked, and in the case of threatening stimulus, not present). |

# Behavioural & social sciences study design

All studies must disclose on these points even when the disclosure is negative.

| | |
|---|---|
| Study description | *Briefly describe the study type including whether data are quantitative, qualitative, or mixed-methods (e.g. qualitative cross-sectional, quantitative experimental, mixed-methods case study).* |
| Research sample | *State the research sample (e.g. Harvard university undergraduates, villagers in rural India) and provide relevant demographic information (e.g. age, sex) and indicate whether the sample is representative. Provide a rationale for the study sample chosen. For studies involving existing datasets, please describe the dataset and source.* |
| Sampling strategy | *Describe the sampling procedure (e.g. random, snowball, stratified, convenience). Describe the statistical methods that were used to predetermine sample size OR if no sample-size calculation was performed, describe how sample sizes were chosen and provide a rationale for why these sample sizes are sufficient. For qualitative data, please indicate whether data saturation was considered, and what criteria were used to decide that no further sampling was needed.* |
| Data collection | *Provide details about the data collection procedure, including the instruments or devices used to record the data (e.g. pen and paper, computer, eye tracker, video or audio equipment) whether anyone was present besides the participant(s) and the researcher, and whether the researcher was blind to experimental condition and/or the study hypothesis during data collection.* |
| Timing | *Indicate the start and stop dates of data collection. If there is a gap between collection periods, state the dates for each sample cohort.* |
| Data exclusions | *If no data were excluded from the analyses, state so OR if data were excluded, provide the exact number of exclusions and the rationale behind them, indicating whether exclusion criteria were pre-established.* |
| Non-participation | *State how many participants dropped out/declined participation and the reason(s) given OR provide response rate OR state that no participants dropped out/declined participation.* |
| Randomization | *If participants were not allocated into experimental groups, state so OR describe how participants were allocated to groups, and if allocation was not random, describe how covariates were controlled.* |

# Ecological, evolutionary & environmental sciences study design

All studies must disclose on these points even when the disclosure is negative.

| | |
|---|---|
| Study description | *Briefly describe the study. For quantitative data include treatment factors and interactions, design structure (e.g. factorial, nested, hierarchical), nature and number of experimental units and replicates.* |

| Research sample | *Describe the research sample (e.g. a group of tagged Passer domesticus, all Stenocereus thurberi within Organ Pipe Cactus National Monument), and provide a rationale for the sample choice. When relevant, describe the organism taxa, source, sex, age range and any manipulations. State what population the sample is meant to represent when applicable. For studies involving existing datasets, describe the data and its source.* |
| Sampling strategy | *Note the sampling procedure. Describe the statistical methods that were used to predetermine sample size OR if no sample-size calculation was performed, describe how sample sizes were chosen and provide a rationale for why these sample sizes are sufficient.* |
| Data collection | *Describe the data collection procedure, including who recorded the data and how.* |
| Timing and spatial scale | *Indicate the start and stop dates of data collection, noting the frequency and periodicity of sampling and providing a rationale for these choices. If there is a gap between collection periods, state the dates for each sample cohort. Specify the spatial scale from which the data are taken* |
| Data exclusions | *If no data were excluded from the analyses, state so OR if data were excluded, describe the exclusions and the rationale behind them, indicating whether exclusion criteria were pre-established.* |
| Reproducibility | *Describe the measures taken to verify the reproducibility of experimental findings. For each experiment, note whether any attempts to repeat the experiment failed OR state that all attempts to repeat the experiment were successful.* |
| Randomization | *Describe how samples/organisms/participants were allocated into groups. If allocation was not random, describe how covariates were controlled. If this is not relevant to your study, explain why.* |
| Blinding | *Describe the extent of blinding used during data acquisition and analysis. If blinding was not possible, describe why OR explain why blinding was not relevant to your study.* |

Did the study involve field work? ☐ Yes ☐ No

# Field work, collection and transport

| Field conditions | *Describe the study conditions for field work, providing relevant parameters (e.g. temperature, rainfall).* |
| Location | *State the location of the sampling or experiment, providing relevant parameters (e.g. latitude and longitude, elevation, water depth).* |
| Access & import/export | *Describe the efforts you have made to access habitats and to collect and import/export your samples in a responsible manner and in compliance with local, national and international laws, noting any permits that were obtained (give the name of the issuing authority, the date of issue, and any identifying information).* |
| Disturbance | *Describe any disturbance caused by the study and how it was minimized.* |

# Reporting for specific materials, systems and methods

We require information from authors about some types of materials, experimental systems and methods used in many studies. Here, indicate whether each material, system or method listed is relevant to your study. If you are not sure if a list item applies to your research, read the appropriate section before selecting a response.

## Materials & experimental systems

| n/a | Involved in the study |
|---|---|
| ☒ | ☐ Antibodies |
| ☒ | ☐ Eukaryotic cell lines |
| ☒ | ☐ Palaeontology and archaeology |
| ☐ | ☒ Animals and other organisms |
| ☒ | ☐ Human research participants |
| ☒ | ☐ Clinical data |
| ☒ | ☐ Dual use research of concern |

## Methods

| n/a | Involved in the study |
|---|---|
| ☒ | ☐ ChIP-seq |
| ☒ | ☐ Flow cytometry |
| ☒ | ☐ MRI-based neuroimaging |

# Antibodies

| Antibodies used | *Describe all antibodies used in the study; as applicable, provide supplier name, catalog number, clone name, and lot number.* |
| Validation | *Describe the validation of each primary antibody for the species and application, noting any validation statements on the manufacturer's website, relevant citations, antibody profiles in online databases, or data provided in the manuscript.* |

# Eukaryotic cell lines

Policy information about cell lines

| | |
|---|---|
| Cell line source(s) | *State the source of each cell line used.* |
| Authentication | *Describe the authentication procedures for each cell line used OR declare that none of the cell lines used were authenticated.* |
| Mycoplasma contamination | *Confirm that all cell lines tested negative for mycoplasma contamination OR describe the results of the testing for mycoplasma contamination OR declare that the cell lines were not tested for mycoplasma contamination.* |
| Commonly misidentified lines (See ICLAC register) | *Name any commonly misidentified cell lines used in the study and provide a rationale for their use.* |

# Palaeontology and Archaeology

| | |
|---|---|
| Specimen provenance | *Provide provenance information for specimens and describe permits that were obtained for the work (including the name of the issuing authority, the date of issue, and any identifying information). Permits should encompass collection and, where applicable, export.* |
| Specimen deposition | *Indicate where the specimens have been deposited to permit free access by other researchers.* |
| Dating methods | *If new dates are provided, describe how they were obtained (e.g. collection, storage, sample pretreatment and measurement), where they were obtained (i.e. lab name), the calibration program and the protocol for quality assurance OR state that no new dates are provided.* |

☐ Tick this box to confirm that the raw and calibrated dates are available in the paper or in Supplementary Information.

| | |
|---|---|
| Ethics oversight | *Identify the organization(s) that approved or provided guidance on the study protocol, OR state that no ethical approval or guidance was required and explain why not.* |

Note that full information on the approval of the study protocol must also be provided in the manuscript.

# Animals and other organisms

Policy information about studies involving animals; ARRIVE guidelines recommended for reporting animal research

| | |
|---|---|
| Laboratory animals | European cuttlefish Sepia officinalis were hatched from eggs collected in the English Channel and the North Atlantic, and reared in a closed seawater system. Animals of both sexes, 3-8 months old, were used in this study. |
| Wild animals | This study did not involve wild animals. |
| Field-collected samples | This study did not involve samples collected from the field. |
| Ethics oversight | This study was approved by the animal welfare authority (Dr. Vet. Med. E. Simon. Regierungspräsidium Darmstadt, Germany) under approval number V54-19c20/15-F126/1025. |

Note that full information on the approval of the study protocol must also be provided in the manuscript.

# Human research participants

Policy information about studies involving human research participants

| | |
|---|---|
| Population characteristics | *Describe the covariate-relevant population characteristics of the human research participants (e.g. age, gender, genotypic information, past and current diagnosis and treatment categories). If you filled out the behavioural & social sciences study design questions and have nothing to add here, write "See above."* |
| Recruitment | *Describe how participants were recruited. Outline any potential self-selection bias or other biases that may be present and how these are likely to impact results.* |
| Ethics oversight | *Identify the organization(s) that approved the study protocol.* |

Note that full information on the approval of the study protocol must also be provided in the manuscript.

# Clinical data

Policy information about clinical studies

All manuscripts should comply with the ICMJE guidelines for publication of clinical research and a completed CONSORT checklist must be included with all submissions.

| | |
|---|---|
| Clinical trial registration | *Provide the trial registration number from ClinicalTrials.gov or an equivalent agency.* |

| Study protocol | *Note where the full trial protocol can be accessed OR if not available, explain why.* |
|---|---|
| Data collection | *Describe the settings and locales of data collection, noting the time periods of recruitment and data collection.* |
| Outcomes | *Describe how you pre-defined primary and secondary outcome measures and how you assessed these measures.* |

# Dual use research of concern

Policy information about [dual use research of concern](dual use research of concern)

## Hazards

Could the accidental, deliberate or reckless misuse of agents or technologies generated in the work, or the application of information presented in the manuscript, pose a threat to:

No | Yes
- ☐ ☐ Public health
- ☐ ☐ National security
- ☐ ☐ Crops and/or livestock
- ☐ ☐ Ecosystems
- ☐ ☐ Any other significant area

## Experiments of concern

Does the work involve any of these experiments of concern:

No | Yes
- ☐ ☐ Demonstrate how to render a vaccine ineffective
- ☐ ☐ Confer resistance to therapeutically useful antibiotics or antiviral agents
- ☐ ☐ Enhance the virulence of a pathogen or render a nonpathogen virulent
- ☐ ☐ Increase transmissibility of a pathogen
- ☐ ☐ Alter the host range of a pathogen
- ☐ ☐ Enable evasion of diagnostic/detection modalities
- ☐ ☐ Enable the weaponization of a biological agent or toxin
- ☐ ☐ Any other potentially harmful combination of experiments and agents

# ChIP-seq

## Data deposition

☐ Confirm that both raw and final processed data have been deposited in a public database such as [GEO](GEO).

☐ Confirm that you have deposited or provided access to graph files (e.g. BED files) for the called peaks.

| Data access links<br>*May remain private before publication.* | *For "Initial submission" or "Revised version" documents, provide reviewer access links. For your "Final submission" document, provide a link to the deposited data.* |
|---|---|
| Files in database submission | *Provide a list of all files available in the database submission.* |
| Genome browser session<br>(e.g. [UCSC](UCSC)) | *Provide a link to an anonymized genome browser session for "Initial submission" and "Revised version" documents only, to enable peer review. Write "no longer applicable" for "Final submission" documents.* |

## Methodology

| Replicates | *Describe the experimental replicates, specifying number, type and replicate agreement.* |
|---|---|
| Sequencing depth | *Describe the sequencing depth for each experiment, providing the total number of reads, uniquely mapped reads, length of reads and whether they were paired- or single-end.* |
| Antibodies | *Describe the antibodies used for the ChIP-seq experiments; as applicable, provide supplier name, catalog number, clone name, and lot number.* |
| Peak calling parameters | *Specify the command line program and parameters used for read mapping and peak calling, including the ChIP, control and index files used.* |

| Data quality | *Describe the methods used to ensure data quality in full detail, including how many peaks are at FDR 5% and above 5-fold enrichment.* |
|---|---|
| Software | *Describe the software used to collect and analyze the ChIP-seq data. For custom code that has been deposited into a community repository, provide accession details.* |

# Flow Cytometry

## Plots

Confirm that:

☐ The axis labels state the marker and fluorochrome used (e.g. CD4-FITC).

☐ The axis scales are clearly visible. Include numbers along axes only for bottom left plot of group (a 'group' is an analysis of identical markers).

☐ All plots are contour plots with outliers or pseudocolor plots.

☐ A numerical value for number of cells or percentage (with statistics) is provided.

## Methodology

| Sample preparation | *Describe the sample preparation, detailing the biological source of the cells and any tissue processing steps used.* |
|---|---|
| Instrument | *Identify the instrument used for data collection, specifying make and model number.* |
| Software | *Describe the software used to collect and analyze the flow cytometry data. For custom code that has been deposited into a community repository, provide accession details.* |
| Cell population abundance | *Describe the abundance of the relevant cell populations within post-sort fractions, providing details on the purity of the samples and how it was determined.* |
| Gating strategy | *Describe the gating strategy used for all relevant experiments, specifying the preliminary FSC/SSC gates of the starting cell population, indicating where boundaries between "positive" and "negative" staining cell populations are defined.* |

☐ Tick this box to confirm that a figure exemplifying the gating strategy is provided in the Supplementary Information.

# Magnetic resonance imaging

## Experimental design

| Design type | *Indicate task or resting state; event-related or block design.* |
|---|---|
| Design specifications | *Specify the number of blocks, trials or experimental units per session and/or subject, and specify the length of each trial or block (if trials are blocked) and interval between trials.* |
| Behavioral performance measures | *State number and/or type of variables recorded (e.g. correct button press, response time) and what statistics were used to establish that the subjects were performing the task as expected (e.g. mean, range, and/or standard deviation across subjects).* |

## Acquisition

| Imaging type(s) | *Specify: functional, structural, diffusion, perfusion.* |
|---|---|
| Field strength | *Specify in Tesla* |
| Sequence & imaging parameters | *Specify the pulse sequence type (gradient echo, spin echo, etc.), imaging type (EPI, spiral, etc.), field of view, matrix size, slice thickness, orientation and TE/TR/flip angle.* |
| Area of acquisition | *State whether a whole brain scan was used OR define the area of acquisition, describing how the region was determined.* |

Diffusion MRI    ☐ Used    ☐ Not used

## Preprocessing

| Preprocessing software | *Provide detail on software version and revision number and on specific parameters (model/functions, brain extraction, segmentation, smoothing kernel size, etc.).* |
|---|---|
| Normalization | *If data were normalized/standardized, describe the approach(es): specify linear or non-linear and define image types used for transformation OR indicate that data were not normalized and explain rationale for lack of normalization.* |
| Normalization template | *Describe the template used for normalization/transformation, specifying subject space or group standardized space (e.g.* |

| Normalization template | *original Talairach, MNI305, ICBM152) OR indicate that the data were not normalized.* |

| Noise and artifact removal | *Describe your procedure(s) for artifact and structured noise removal, specifying motion parameters, tissue signals and physiological signals (heart rate, respiration).* |

| Volume censoring | *Define your software and/or method and criteria for volume censoring, and state the extent of such censoring.* |

## Statistical modeling & inference

| Model type and settings | *Specify type (mass univariate, multivariate, RSA, predictive, etc.) and describe essential details of the model at the first and second levels (e.g. fixed, random or mixed effects; drift or auto-correlation).* |

| Effect(s) tested | *Define precise effect in terms of the task or stimulus conditions instead of psychological concepts and indicate whether ANOVA or factorial designs were used.* |

Specify type of analysis: ☐ Whole brain   ☐ ROI-based   ☐ Both

| Statistic type for inference (See Eklund et al. 2016) | *Specify voxel-wise or cluster-wise and report all relevant parameters for cluster-wise methods.* |

| Correction | *Describe the type of correction and how it is obtained for multiple comparisons (e.g. FWE, FDR, permutation or Monte Carlo).* |

## Models & analysis

n/a | Involved in the study
☐ | ☐ Functional and/or effective connectivity
☐ | ☐ Graph analysis
☐ | ☐ Multivariate modeling or predictive analysis

| Functional and/or effective connectivity | *Report the measures of dependence used and the model details (e.g. Pearson correlation, partial correlation, mutual information).* |

| Graph analysis | *Report the dependent variable and connectivity measure, specifying weighted graph or binarized graph, subject- or group-level, and the global and/or node summaries used (e.g. clustering coefficient, efficiency, etc.).* |

| Multivariate modeling and predictive analysis | *Specify independent variables, features extraction and dimension reduction, model, training and evaluation metrics.* |

