## [Peer Review File · Nature]

Manuscript Title: The Dynamics of Pattern Matching in Camouflaging Cuttlefish

Reviewer Comments & Author Rebuttals

Reviewer Reports on the Initial Version:

Referees' comments:

Referee #1 (Remarks to the Author):

In the article Texture-matchin search in camouflaging cuttlefish, the authors present studies of the space of response patterns to many backgrounds, the dynamics of transitions between patterns, the differential grouping and regrouping of chromatophores, the capturing of spatial frequencies of checkboard backgrounds and the preservation/dynamics of the camouflage pattern interrupted by blanching. The data generated for these studies seem amazing – great examples of the kind of rich, massive data sets that are changing fields that were traditionally observational.

I am not able to comment on the novelty of all of the results in this study, as I am in a neighboring field, but I did find them fun, interesting and have enjoyed reading the article. I would add, however, that the article was extremely dense with more findings than I could comfortably wrap my head around – seemingly all good pieces of the puzzle but they did not click together as I was hoping. I think I would have preferred to read the same material as two or three articles, preferably with more examples, videos, statistics, and ties between the different parts (e.g. transition patterns (in time) of clusters of chromatophores co-varying in checkerboards square-sizes). The density of the material was particularly evident in the figures and figure legends – just so, sooo much. Cool stuff but I'm exhausted.

Below is a list of comments, broken up by topic. Note, I am mixing major and minor points but will denote stronger concerns with * or ** depending on the level of concern.

SKIN PATTERN SPACE

Ultimately the discussion of the repertoire of possible patterns of the animals and how it is high dimensional centers around the number of PCs needed, I would suggest showing plots of the number of PCs on the x-axis and explained variance on the y-axis.

In figure 1(h), it would be good to see where the hot spots correspond to in the UMAP space – perhaps it could made as a contour plot and placed over a faint version of 1(e).

DYNAMICS OF TRANSITIONS / SHARED SLOW SPOTS

* It was unclear how a “trajectory” was defined (sorry if this was clearly stated somewhere and I

missed it). It would be nice if the definition would be put also in relation to the speed of change plot in figure 1(f).

The trajectories shown in figure 2(a1 and a2), were these during transitions between the same backgrounds (e.g. 1->4 with notation as in extended data fig 1(d))?

* The metric for describing the trajectories in figure 2(a) as having starting and ending points in the same neighborhood and forming noisy bundles, is this by eye in the space of the first 3 PCs? It is my understanding that the first 3 PCs would not explain much variability (e.g. need for 200 PCs, accounting for ~70% of the variability, to estimate speeds).

The points and trajectories in figures 2(a1, a2, c and e) are not easy to see, even when zoom in on the higher resolution images.

* It would be helpful to see the stability of the spatial distribution of pattern change speed in figure 2b, shown for even/odd trajectories or grouped by transitions between the three natural background images – in particular since as the mean speed metric is in arbitrary units, with the variability difficult to interpret.

* It is my understanding that the trajectories that were considered similar for further evaluation in figure 2 were determined “manually” from the 1D (LLE from 5D PCA) version of the pattern space shown in extended data figure 3(a). It is not clear to me in what way these trajectories are similar. Perhaps it would help to plot also the transitions that were deemed not similar to these 7.

* It is difficult to imagine that the location of these “shared slow spots” is really shared in terms of the full pattern space since this is a 1D version of the first 5 PCs that themselves only explain a small % of the variance of the data. Perhaps the authors could report the fraction of variance that this 1D LLE representation explains.

* Please explain further how the local minima of the average speed were determined as it is not clear why some local minima visible in the right panel of extended data figure 3(a) were used as “local minima” and other skipped.

** The bulk of the argument for the shared slow spots appears to come from the one example of 7 transitions built up in figures 2(e-f) and extended data figure 3. In the extended data figure there are two more examples presented from two other animals but missing are the cross-validations of the slow shared spots and in general it isn't clear how shared or slow the shared slow spots are in these additional examples. I would be more confident if these results were accompanied by some sort of permutation test showing how likely it is to find such results by chance or with an alternative approach altogether (e.g. suggestion below).

Perhaps another approach would be to search in the high dimensional space for locations along the transitions where there are many other transitions close by (measured in the 200 PC space used to estimate speeds). The speeds at these shared locations could then be compared with speeds in the surrounding volume to determine if there is a slowing of the transitions through each. The number

of slowing spots could then be compared with the number of spots without slowing, etc. It is not obvious that this would have to be limited to only those trajectories that were similar in some way. This is just one thought of an alternative analysis that could be done to check the claims related to the shared slow spots. I would welcome other alternatives. As it is, the current support is not that convincing as it is largely based on a single example with manual preprocessing steps.

The p-values shown in figure 2(f) and extended data figure 2(e) are not obviously valid. The shared slow spots were chosen as being local minima in the 1D pattern space and movement on the space should be continuous, so time and location should be related, making the test problematic. Also, please state the details of the paired t-test (e.g. H_0 : speed at $t=-2.5s$ is equal to speed at $t=0$).

* It would be nice to show stability of the plots in figure 2(h), perhaps on even versus odd trajectories. As an illustration it is cool but it is hard to determine how stable/meaningful it is. They could also be shown for the other slow spots.

* It would be nice if the pattern-change speed distributions in 2(j) could be plotted separately for the first and second halves (in time) of the trajectories to show that the bimodality is indeed different than the phenomenon in 1(f).

If 2(j), it is unclear if all three data sets rejected the unimodal test mentioned, perhaps the p-values could be listed.

It is also unclear the relationship between the slow spots in c-h and the bimodality in i,j, especially since the "slow spots" are local minima of the speeds with a number of local minima being faster than local maxima in other parts. This could be made apparent in the text.

(RE)ORGANIZATION OF CHROMATOPHORE GROUPINGS DURING PATTERN TRANSITIONS

The clusterings of figure 3 are thought-provoking and the idea that individual chromatophores are rearranged into different groups in different trajectories is cool.

** In figure 3(e) and extended data figures 4(f), if I understood correctly, the distance on the x-axis is the average distance between pairs sampled from 2 clusters. I would think a metric more related to the idea in figure 3(d), showing two clusters can be "interdigitated" and not grouped together, would make more sense. The approach that comes to mind is to treat the activation patterns as densities (e.g. by binning or smoothing spatially) and then consider their overlap or the Wasserstein distance between the two as the metric for the x-axis of this plot. As it is, it is unclear how well the results support or not the claim except for the very few points to the left and right of the blob centered around 18mm, which are probably from pairs of small clusters.

I am curious about the points with correlation coefficients remarkably close to 1 in figure 3(e) – wouldn't these end up in a common cluster?

Please explain the "n" values in extended data figure 4(f).

* The two examples in figure 3(a-d) and 3(g-i) are compelling and seem to support the conclusion that the chromatophores are re-organizing into different groupings during different transitions. I think, however, the argument would be strengthened if the comparison of the groupings could be made less dependent on the clustering algorithm (and hyperparameters). For example, it would be nice to see/compare the pairwise correlations matrices of the active chromatophores (for example, of those in 3(i)) across many transitions with all matrices sorted by the clustering from one transition. This would still be example-based but perhaps a few of these could be shown with some larger summaries.

** The decoding results in 3(f) come across as slightly misleading. If I have understood correctly, the plot shows a significant increase in accuracy using the clusters identified in the same pattern transition (decomp) versus using clusters identified from similar, all or other transition trajectories, etc. This comparison, however, appears problematic since, in the decomp case, the same data used for clustering is also used for decoding (i.e. in-sample accuracy), allowing for potential overfitting that is not possible in the comparisons. I would suggest removing the other comparisons unless a suitable way to account for this can be applied.

The names on the x-axis of 3(f) were challenging to decipher from figure text, main text and extended figure text (still not entirely sure what “nearest” refers to). It is also not clear how many / which transitions were used to create these.

It is not clear how the x-axis is defined in figure 3(j) and extended data figure 3(g).

* Please remark on values of the mean intersection of unions (IoU) in 3(j). The mean IoU values are on average low even for similar trajectories but it is not obvious how those numbers compare to chance or what to expect from noisy clusters. Perhaps some context could be given for these numbers along with some additional metrics for clustering similarity, like the adjusted rand index. It would be nice to show that these are not sensitive to the method (and hyperparameter) used for clustering.

CAMOUFLAGE PATTERNS ON CHECKERBOARDS

In figure 4(b), it seems odd to show the points in a UMAP embedding further projected onto the two axes. First, please confirm that the UMAP was originally 2D. Next, it would be nice to see the points plotted in the 2D UMAP. Finally, it is unclear what to understand from this plot, especially as UMAP is a nonlinear method, not necessarily preserving relative locations of the points. It appears to show more or less similar properties as in (a) but in a more confusing way and now with the additional data sets. Would it not be possible to simply add these points to figure 4(a)?

The x-axis in figure 4(c) appears to be mistaken as it goes into negative values for the skin-pattern periods.

In the text for figure 4(d), to make it easier for the reader, I would recommend relating the values in

the matrix to the insets in 4(c).

In figure 4(e), it is not stated what is being shown in the bottom two rows.

In figure 4(f), are the points in each cluster “frames sampled at 25 Hertz over 46 seconds”? Also, why 46 seconds?

The example clusters identified in figure 4(g) and ext data figure 5(i) appear compelling but are not easily tied into the results from figure 3. It would be great if this could be expanded on / discussed. Following the methods described for pattern transitions, perhaps similar pattern trajectories could be identified during transitions through the checkboard backgrounds and then the trajectories visualized (as in 3h) using the clusters in 4(g).

TRANSITION INTO AND OUT OF BLANCHED STATE

In figure 5(c), it is not obvious that the data from the four animals can be combined to include a single fit. The y-axis is in arbitrary units that will vary from data set to data set. Given that a normalization would be tricky, perhaps it would be best to show the fit and corresponding r-squared values for each animal separately.

The brown and purple colors in figure 5(g) are difficult to see.

As with the chromatophore clusters in figure 4(g), the nice clusters in 5(g) coming from multiple trials appear to contrast the unstable clusterings during similar trajectories in figure 3. It would be nice to have these come together.

While the animals in extended data figure 8 were not as compelling as the one in figure 5, I still found they supported the claim well.

OTHER COMMENTS:

The split of vertebrates and cephalopods being 600M years ago was mentioned in three locations, maybe one less would be OK.

The definition of skin pattern space varied depending on the part of the analysis, making it difficult to follow. I made a list:

- UMAP (min_dist=0.8, n_neighbours=100) projection of the max pool layer of a pre-trained deep network responding to the images corresponding to a geometry preserving downsampling of the PCA (at 80%) of the images. <- for figure 1
- PCA to the first 200 dimensions <- for velocities throughout
- PCA to 2- or 3 dimensions <- for visualizations, evaluation of the trajectories, estimation of energy landscapes, etc.
- 1D locally linear embedding (LLE) of the top 5 principal components <- to identify slow spots

- PCA to 95% variance <- to study correlations in the checkboard section
- UMAP (min_dist=0.3, n_neighbours=20) projection of the "data". It appears that the data was first reduced with PCA to PCs preserving 95% of variance but this is not clear. <- for figure 4(b)

For these, it was not always clear if the pattern space referred to the space of all data collected for the animal or just a small subset. I would suggest finding a better way to summarize all of this.

It would be nice to have plots of the explained variances as a function of the number of dimensions.

It was not clear why the three clusterings of the chromatophores in figures 3, 4 and 5 were each done differently (50 principal components versus first 80%, different values for the number of neighbors and resolution) and in figure 5 the trajectories were first downsampled to just regions of high (top 10%) normalized speed.

That is it. I am sorry for the lengthy review. I hope the comments are helpful. Thanks again for the interesting article!

Referee #2 (Remarks to the Author):

Summary:

This manuscript by Woo et al. uses sophisticated methods to film and track individual chromatophores on the skin of cuttlefish over days-to-weeks of the animal's life. In contrast to the authors' previous study in Nature that used similar tracking and statistical methods (Reiter et al. 2018), the animals in this study exhibited camouflage behavior in response to different visual textures printed on fabric. Therefore, this study provided an exciting opportunity to dissect the relationship between the cuttlefish's visual stimulus and its camouflage output on its skin.

The major strength of this manuscript is the immense dataset, which required a highly impressive feat of engineering and computation to track the skin at chromatophore-level resolution during the fascinating behavior of camouflage. The field of cephalopod camouflage is in desperate need of more quantitative and rigorous studies. Therefore, this manuscript (with revisions) would provide an important contribution to the field. With that said, I am not convinced that the analyses the authors chose to perform on these datasets revealed much about the camouflage or visual system that we didn't know before. Many of the conclusions are in line with previous observations (inc Reiter et al. 2018), and the most interesting conclusions didn't have enough evidence to make a strong statement (e.g. the suggestion that the system does, in fact, have feedback, contrary to previous studies). Finally, the big advancement in this study is the addition of camouflage behavior (as opposed to the non-camouflage pattern changes seen in Reiter et al. 2018). The authors dedicated surprisingly little of this study to the investigation of camouflage - thus, this paper doesn't appear to be a major advancement over the previous study. For these reasons I do not recommend this manuscript for publication in Nature.

General feedback:

- I believe the subject of cuttlefish pattern change/camouflage is of interest to people inside and

outside of the field, but the manuscript is written in a way that makes it very difficult for scientists outside of computational/statistical disciplines to understand it. A significant rewrite is required to make the manuscript understandable to a broader audience. For instance, each section of the results should begin with some motivation for the particular analysis being performed (i.e. it should not begin with a description of the figure, e.g. "Figure 3a,b illustrates the evolution of camouflage..." - L183).

- There is a general sense that the authors created an impressive dataset but then weren't sure what to do with the data, so tried a number of analyses (with unclear biological motivation), and included all of those figures in the manuscript. Many of these figures are not necessary and should be moved to the supplement.

- A lot more care is needed in generating figures - it took far too long to dissect most figures. There should be fewer figures, they should be larger, have legends in the figure itself, and all axes should be labeled. The text legend needs to be clearer, and the main text should explain what analysis was performed to create each figure.

- I recommend that the authors be careful in their use of the word "texture" to mean "pattern". Given that cuttlefish can change 3D texture to match their surroundings, the title could be misinterpreted (as well as many other instances of the word). Perhaps use "visual texture"?

I will address each of the major conclusions the authors made.

1. "Camouflage-texture space is high-dimensional" (L41)

Figure 1 shows substantial complexity in the skin patterning system, which suggests camouflage does not consist of the many-to-few mapping that has been indicated by other researchers in the field. However, neither the main text nor figure explicitly addresses whether camouflage is high-dimensional. If this is a major conclusion of the paper, I suggest the authors explain this in the main text.

For the entire figure, the presentation of data could be greatly improved. For instance:

- After showing Fig 1A, I would show 1E to make it clear what the UMAP plot is indicating. Then I would provide the analysis (1B-D). However, I find plots 1B-D difficult to decipher, especially because the reader has to make constant reference to the 1E plot, and the colors superimposed on each other are difficult to see. I would find an alternative method to represent this data (or provide a magnified view of one cluster). The color codes should be included as a legend in the figure itself.

- Figure 1E: the authors state that this represents a great diversity of camouflage patterns, but at the resolution provided, they don't look very different (other than 4 & 6). Perhaps the authors should show a single (larger) image for each pattern, and include the 3x3 panel in the supplement.

- Figure 1H: the authors mentioned that there was sometimes image blur when the cuttlefish swam during a fabric change, and they said "we could usually track skin patterning at low resolution when the animal swam during these transitions". This implies that the authors could be missing patterns in other regions of texture space, so I don't think they can support the conclusion made for Figure 1H.

2. "The search for a pattern ultimately accepted by the animal was not stereotyped. Rather, patterns meandered through texture space, decelerating and accelerating repeatedly during each search, before stabilizing around a chosen, noisy pattern." (L42)

There are a few issues with this statement:

i. I agree that the data indicates animals used differing paths to reach a given pattern. But it's not clear that the data shows stabilization (especially since we can't see how long each trajectory is, or where each specific trajectory begins and ends). It would be useful to see a plot of the animal's skin change in response to each stimulus over time, so the reader can see the stabilization as a function of time (in seconds).

ii. It's not clear what the biological relevance of the accelerations and decelerations are, and the authors don't provide any motivation for this analysis. The authors even say "it is too early to say how illuminating this framework might be" (L322). Furthermore, the pauses are very small dips in speed (e.g. Fig 2F, see below).

iii. The authors suggest the stabilization of the system indicates there is feedback via sensory or via corollary discharge (L318 and L350). However, while this is an interesting suggestion, this statement requires evidence of such feedback.

Additional figure comments:

- The conclusion that cuttlefish used differing paths to reach a given pattern is in contrast to Reiter et al. 2018 (Fig 4), where the authors tracked the cuttlefish's response to motion. They state "upon each stimulus, the animal not only generated the same target patterns but also moved with chromatophore-level repeatability through very similar, low-dimensional sequences of intermediate states". The authors should supply an interpretation of this difference.

- Many of the trajectories are not visible because they're all superimposed on each other. If the authors want us to assess the similarity of trajectories, we should be able to see them.

- Figure 2F: the data underlying these small changes in speed (colored lines behind the black line) have massive variance. I don't think the authors can make a strong case for their conclusion with this data.

- It's not clear what value 2H adds to the paper. I would put this in the supplement unless the motivation for looking at this system as an energy landscape is provided. Also, a direct reference to this figure is missing in the text.

3. "During pattern motion, chromatophores could be grouped into "pattern components", based on their covariation. These components could take many shapes and sizes, and overlay one another or not" (L47)

This was established in Reiter et al. 2018 Figure 3. It's not clear what significant new information this figure offers, especially since the authors don't relate this analysis to the major difference between this study and the previous - the animal is now camouflaging.

4. "Two transitions between similar starting and ending skin patterns typically invoked very different components, suggesting neural flexibility in the realization of pattern search" (L50). "Identifying the smallest, consistent components of camouflage patterns will require even larger datasets" (L332).

It is an interesting result that two transitions between similar patterns involved very different components, and this adds to the conclusion that camouflage is very complex (and much more so than suggested by the majority of previous studies in the field).

It seems to me that identifying the smallest component (the effective pixel size) is one of the most interesting questions that should actually be identifiable using this dataset. The authors could focus their analysis on a region with a small pattern component (e.g. a spot).

Figure comments:

- There are more issues with interpretability: the colors aren't all visible on the animal's skin in Figures 3A and B, which leads to a lot of confusion about what the reader is looking at.
- Fig 3E: It's not clear why the distance between chromatophores is concentrated at 20 mm. This implies that there is a relationship between the physical distance on the skin. A clear explanation of how this plot was generated is missing in the text.
- The conclusion of Fig 3E - that pattern components can be physically clustered or loose and distributed - is very similar to a conclusion of Reiter et al. 2018 (Fig 3D) so I found it to be an expected result.
- Fig 3F is not clear at all. The x-axis abbreviations and descriptions are not provided.
- Fig 3G: the images of this pattern change may look similar, but the square appears at a different time, and the resolution is sufficiently low that I can't really see the change. So I'm not surprised the trajectories are somewhat different.
- By the end of Figure 3 it seems the authors are just making the same point multiple times. I would put some of this data in the supplement as it doesn't add to the paper.

5. "Using checkerboards spanning a wide range of spatial frequencies, we identified clusters of chromatophores whose state co-varied with input spatial frequency" (L333).

It is useful that the authors quantified the response of cuttlefish to changing spatial frequencies, but this result is in keeping with previous studies using checkerboards as visual input. Thus, this result is not surprising.

Figure comments:

- Figure 4B: what is the y-axis? There should be an explanation/legend of what red and white are. Is there a different way of showing this instead of UMAP_0 and UMAP-1?
- Fig 4C is underwhelming and hard to see an effect. Is there a way to plot this another way that makes the effect more clear?
- Fig 4D: It isn't clear what is being shown.
- Fig 4E: what's the y-axis? It's unclear if the mean power values are significant.

6. "Analysis of patterns evoked by threats of camouflaging animals revealed that a memory of the initial camouflage remains upon withdrawal of the threat" (L339).

This is an interesting result but there is the caveat that the animal may be returning to the same state after blanching because its environment hasn't changed and it's choosing the appropriate response (as opposed to retention of the previous state). The authors should include a control. For

instance, they might change the background while the animal is experiencing the threatening stimulus to see if the animal still returns to its original state before updating its skin pattern. The authors could also alternate two backgrounds, and plot the variance in skin pattern when the animal returns to the same pattern each time. Then, compare that to the animal returning to its skin pattern after blanching. Presumably the accuracy of the pattern reproduction should be much higher after blanching compared to post-camouflage change. It is too strong a statement to say that the animal has a memory of the initial camouflage.

Methods comments:

- What size of animals did you use for the experiments?
- What type of lighting did you use for the behavior system?

Conclusion:

The authors created an important dataset and developed highly sophisticated methods to study the skin of cuttlefish during camouflage. However, the analyses chosen by the authors fall short of the promise of this dataset. Too many of the conclusions either confirm previous studies, are not of notable significance, or require additional experimental support. For those reasons I think this study is not suited for publication in Nature.

Referee #3 (Remarks to the Author):

A. Summary of the key results

Woo et al. present a dazzling array of data to convince the reader that chromatophore control in cuttlefish in response to visual stimuli is extremely complex (not just three patterns) and from my interpretation controlled by a non-linear system at times. They were able to complete this study due to a previous publication demonstrating the method of tracking of 10,000s of chromatophores simultaneously.

They present several findings as new from their study including (1) patterns expressed are in a vast array of forms, (2) patterns are not stereotypical (though they contradict this later), (3) pattern expression changed over time and the speed of changes altered over time, and (4) different extracted patterns resulted when different stimuli were used, such as natural, vs checkerboard, vs looming, which they hypothesized was due to different control systems.

B. Originality and significance: if not novel, please include reference

- The method of tracking chromatophores is not novel and was published in Nature 2018 (<https://doi.org/10.1038/s41586-018-0591-3>).
- The relationship between visual stimuli and expression of skin patterns is not novel, as evidenced by many papers from the labs of Roger Hanlon and Daniel Osorio (See the massive number in Hanlon & Messenger 2018. *Cephalopod Behaviour*. Cambridge University Press).
- However, the novelty of this study would have been the utilization of a major increase in the spatial and temporal resolution in their tracking method to precisely quantify the relationships between visual stimuli and chromatophore activity at the level of single chromatophores (or small patches of chromatophores).

- The significance is they have demonstrated that chromatophore control systems in cuttlefish are very sophisticated (more than previously published) and their analysis seems to suggest they are producing highly tailored camouflage patterning across the dorsal surface with respect to specific visual stimuli.

C. Data & methodology: validity of approach, quality of data, quality of presentation

The tracking method is accurate from what was reported in Nature and this manuscript. It is producing some very detailed information about the control of cuttlefish skin patterns. But there are some major problems in the presentation of the research in this manuscript and potentially how it has been analyzed.

- The visual stimuli, particularly the natural types, are not presented or even described and related to each figure. Without this information, I cannot properly review the paper. I thank the authors for providing the stimuli, later on. They really should be included with the paper for publication and each data analysis and figure be referred to the given stimuli. Also, given their diversity of the stimuli, there should be some relationship made between the patterns expressed on the skin and the visual stimuli.
- Cuttlefish express papillae dynamically which alter skin texture and as the authors noted, the extraction of chromatophore data. The word papillae was only mentioned once, in the methods, and due to their presence, 19-21% of the chromatophore data was disregarded. This seems like a major limitation and it should have been highlighted throughout the introduction, results and methods sections.
- The title of the paper and throughout uses the word texture, but given that cuttlefish can alter the texture of their skin then texture should be defined very clearly early in the abstract and introduction to avoid confusion. Texture of the stimuli should also be defined as purely 2D as cuttlefish can resolve depth and use this information to express papillae. Finally some of the naturalistic stimuli contain visual depth cues, like shading or perspective, that the cuttlefish will perceive as depth information, and this would then add more complexity to the skin pattern and texture expressed, and the analysis and discussion.
- The manuscript discusses patterns, but most of these instances of the word pattern are not referring to specific arrangements or configurations of expressed chromatophore patterns, but rather they are implied patterns that were extracted from the UMAP analysis. The analysis of chromatophore activity without providing the ground truth that UMAP changes and variances are linked to specific patterns is not clearly shown (only Fig 1e, and there is no analysis of difference with and between the 8 selected clusters) and so this makes understanding the context of the UMAP and related data very hard. Pattern should be very clearly defined and differentiated from those extracted from correlations in data analysis and those that statistically differ on the skin of the animal (which relates back to error rate in segmentation, analysis, significant change in chromatophore diameter, etc.).
- I was also expecting that the chromatophore patterns from each trial and each animal would be warped to a common cuttlefish ground plan (shape) allowing analysis of different skin neural control 'units'. It would also allow different levels of neighborhood analysis that could be increasingly pixelated to compare to visual resolution of predators at different distances (the main driving force for camouflage). From my reading I did not see this type of approach or discussion.
- I was also expecting this type of common ground plan or framework analysis to link back to the stimuli presented to understand what image statistics link to which patterns expressed. It was really

surprising to see such low r^2 correlations between checkerboard patterns and chromatophore patterns (fig4).

- The authors do not cite, nor do I know of, papers that show sensory feedback from the skin to the brain. In this case it is hard to see how changing the chromatophores from one state to another over time constitutes a 'search'. It is much more parsimonious to consider they are taking a long time (relatively to the time to change a chromatophore state) to decide what pattern they finally decide to express. Given this possibility, it should be clearly defined what they mean by search. Given we also do not know what cuttlefish are thinking about, I suggest that search be replaced with "matching phase" or something similar that is linked to the action rather than implying an unknown mechanism.
- There seems to be a mismatch with the link between stimuli and chromatophore expression patterns. How much did the view of the visual stimuli differ (between trials and between animals)? Could any of these differences be averaged out and so be ruled out experimentally as not being important?
- Stereotypical responses query: Line 41 "...we found that camouflage-texture space is high-dimensional and that the search for a pattern ultimately accepted by the animal was not stereotyped." Yet on line 111, "Fig. 1c is in fact composed of several smaller clusters, each corresponding to the response of the animal to a different presentation of the same natural image. Variations within a cluster represent skin-pattern noise: patterns are never static, because chromatophores often flicker and undergo small local fluctuations." These two statements seems contradictory to me (along with others in the manuscript), that perhaps the UMAP method is too detailed (or at too fine a scale) due to the vast amount of data and the interactions between data points that are non-linear, that perhaps the method and the conclusions are hard to parse. I just don't know without more clarity in the writing and the presentation of the variance relative to the biology and the visual stimuli provided.
- The paper focuses a lot of capabilities of the cuttlefish but does not put it into context whether what is analyzed and plotted is normal or atypical. The link between capability and repeated tests from different animals should be clear. In many cases the figure legends do not describe the number of animals tested, the number of trials tested, which is the stimulus used (linking back to its picture), whether the stimulus is novel or repeated, etc.

D. Appropriate use of statistics and treatment of uncertainties

There is a lot of data analysis in this paper but without a better understanding of the variance and its link to the biology it is very hard to comment on the statistics and uncertainties.

E. Conclusions: robustness, validity, reliability

See above, without the stimuli linked to the figures, and the data analysis performed that highlights the relationship between stimuli and the expressed chromatophore patterns it is hard to understand if the conclusions are warranted.

There is also no easy way to parse whether temporal fluctuation variance in chromatophore expression is due to (1) encountering a new environment or visual scene, (2) still deciding what pattern is most suitable to express across their body, (3) variance within a stable expression pattern that has been chosen, and (4) refinement of a suitable match (expressed for some time), due to noticing new details in the scene presented. Many cephalopods are constantly vigilant and scanning their surroundings with slow head and eye rotations. While the authors have made some analyses

which may cover these options, they are not clear to me, particularly relating to variance in the dataset.

Line 315: "How then does an animal know that it has reached a suitable pattern?" This question raises another question, Does the animal need to know it has reached a suitable pattern? In the wild if it got it wrong the animal may be eaten by predators very quickly. Predators swimming past without noticing its presence would be a very good indicator and so Line 318 suggesting feedback does not even have to exist. It is also unclear which data suggests feedback exists.

Line 327: "apparent "pattern components", whose composition could be small or large, clustered or loose,..." This text apart from suggestion multi-scale control, also suggests the possibility of low dimensional outputs given the right conditions or experience (learning from predators). Thus it seems linking this back to previous work as parallel theory should also be posed.

Line 340: "revealed that a memory of the initial camouflage remains upon withdrawal of the threat." This was the most exciting line of the manuscript for me. By suggesting that once a visual inspection finds a match and expresses it, it can then be reproduced later, but also doesn't this also suggest that potentially all the chromatophore pattern fluctuations while they are in the matching phase (decision making most likely, rather than an actual search) are unimportant to the animal as they can be skipped over.

F. Suggested improvements: experiments, data for possible revision

See above.

G. References: appropriate credit to previous work?

The method is new, so no previous comparisons. The older work seems to be referenced well.

H. Clarity and context: lucidity of abstract/summary, appropriateness of abstract, introduction and conclusions

See above.

Author Rebuttals to Initial Comments:

Responses to Reviewers

We thank the reviewers for their detailed comments and criticisms. Before addressing them point by point, we summarise the major changes applied to our original submission.

The reviewers' criticisms could be split in 3 categories.

1. Issues of analysis: the reviewers' requests were many, precise and very useful; we responded to all of them, carrying out many new experiments, tests and new analyses on >130 hrs of video recording. Accordingly, figures 2 and 3 have been greatly modified to better illustrate the link between chromatophore patterns and background images. (A few peripheral points in the original version—e.g., hypothetical attractor landscape—have been eliminated, providing space for these new results.)

2. Issue of narrative: the reviewers found the paper overwhelming and descriptive; one referee reported that the data were presented without underlying rationale for our experiments. This is fair, and we could—and in hindsight, should—have used guiding questions or hypotheses at the outset. For example, a driving reason for this study is that past work suggested a low dimensional skin pattern space and a many-backgrounds to few-camouflages mapping—which we show to be incorrect. We have thus changed the narrative structure of the manuscript to motivate our experiments. We believe that this has greatly improved the manuscript.

3. Issues of novelty: one or two of the reviewers thought that our results were not novel. This can probably be linked to the issue of narrative: without precise reminders of, and introduction to prior results, the significance and novelty of our results was missed. We hope that this new version will correct this. Indeed, we trust that all the results of our study are new: the quantitative approach to camouflage (our previous paper did not address camouflage); the high-dimensionality of skin-texture space and the fine sensitivity of camouflage to background texture and spatial frequency (both contradicting published results from other labs); the complex dynamics of camouflaging, suggesting the role of feedback (entirely new); the complex and variable decomposition of pattern formation (new); the evidence for the separation of control systems for camouflage and another pattern production (new).

We believe that our extensive revisions and additions to our original manuscript address the reviewers' pointed criticisms.

We also believe that this new version has been improved tremendously by these criticisms and revisions, and thus thank the reviewers for their very constructive feedback.

Referee #1 (Remarks to the Author):

In the article *Texture-matching search in camouflaging cuttlefish*, the authors present studies of the space of response patterns to many backgrounds, the dynamics of transitions between patterns, the differential grouping and regrouping of chromatophores, the capturing of spatial frequencies of checkboard backgrounds and the preservation/dynamics of the camouflage pattern interrupted by blanching. The data generated for these studies seem amazing – great examples of the kind of rich, massive data sets that are changing fields that were traditionally observational.

I am not able to comment on the novelty of all of the results in this study, as I am in a neighboring field, but I did find them fun, interesting and have enjoyed reading the article. I would add, however, that the article was extremely dense with more findings than I could comfortably wrap my head around – seemingly all good pieces of the puzzle but they did not click together as I was hoping. I think I would have preferred to read the same material as two or three articles, preferably with more examples, videos, statistics, and ties between the different parts (e.g. transition patterns (in time) of clusters of chromatophores co-varying in checkerboards square-sizes). The density of the material was particularly evident in the figures and figure legends – just so, sooo much. Cool stuff but I'm exhausted.

Below is a list of comments, broken up by topic. Note, I am mixing major and minor points but will denote stronger concerns with * or ** depending on the level of concern.

We thank the reviewer for his/her helpful comments. We have restructured our manuscript to incorporate our results in the context of previous work (see general revisions summary above). We also provide new analyses, as requested by all reviewers.

SKIN PATTERN SPACE

1. Ultimately the discussion of the repertoire of possible patterns of the animals and how it is high dimensional centers around the number of PCs needed, I would suggest showing plots of the number of PCs on the x-axis and explained variance on the y-axis.

These analyses are now provided (EDF 2).

2. In figure 1(h), it would be good to see where the hot spots correspond to in the UMAP space – perhaps it could be made as a contour plot and placed over a faint version of 1(e).

This analysis has been removed, not being central.

DYNAMICS OF TRANSITIONS / SHARED SLOW SPOTS

3. * It was unclear how a “trajectory” was defined (sorry if this was clearly stated somewhere and I missed it). It would be nice if the definition would be put also in relation to the speed of change plot in figure 1(f).

By trajectory we mean ‘path’. Definitions and clarifications are now provided in Methods, e.g., in connection with speed, and when discussing Figs. 3, 4 and EDF. 3d-f (previously Fig. 1f).

4. The trajectories shown in figure 2(a1 and a2), were these during transitions between the same backgrounds (e.g. 1->4 with notation as in extended data fig 1(d))?

Yes. We now show the corresponding background images (Fig. 3a,b, shown below).

We also compute the texture metric for the background (as done for the animal's camouflage pattern), so as to measure the distance between animal and background textures as camouflaging unfolds (Fig. 3d).

5. * The metric for describing the trajectories in figure 2(a) as having starting and ending points in the same neighborhood and forming noisy bundles, is this by eye in the space of the first 3 PCs? It is my understanding that the first 3 PCs would not explain much variability (e.g. need for 200 PCs, accounting for ~70% of the variability, to estimate speeds).

This analysis has been removed, because it is not central.

5. The points and trajectories in figures 2(a1, a2, c and e) are not easy to see, even when zoom in on the higher resolution images.

We apologise. We have replaced these plots with new dynamic plots (Fig. 3a,b, see point 4 above).

6. * It would be helpful to see the stability of the spatial distribution of pattern change speed in figure 2b, shown for even/odd trajectories or grouped by transitions between the three natural background images – in particular since as the mean speed metric is in arbitrary units, with the variability difficult to interpret.

We now show the partial stability of pattern change speed in EDF. 5a. We have also now split the trials randomly into two sets, as suggested (EDF. 5b) (see below).

7. * It is my understanding that the trajectories that were considered similar for further evaluation in figure 2 were determined “manually” from the 1D (LLE from 5D PCA) version of the pattern space shown in extended data figure 3(a). It is not clear to me in what way these trajectories are similar. Perhaps it would help to plot also the transitions that were deemed not similar to these 7.

* It is difficult to imagine that the location of these “shared slow spots” is really shared in terms of the full pattern space since this is a 1D version of the first 5 PCs that themselves only explain a small % of the variance of the data. Perhaps the authors could report the fraction of variance that this 1D LLE representation explains.

* Please explain further how the local minima of the average speed were determined as it is not clear why some local minima visible in the right panel of extended data figure 3(a) were used as “local minima” and other skipped.

** The bulk of the argument for the shared slow spots appears to come from the one example of 7 transitions built up in figures 2(e-f) and extended data figure 3. In the

extended data figure there are two more examples presented from two other animals but missing are the cross-validations of the slow shared spots and in general it isn't clear how shared or slow the shared slow spots are in these additional examples. I would be more confident if these results were accompanied by some sort of permutation test showing how likely it is to find such results by chance or with an alternative approach altogether (e.g. suggestion below).

Perhaps another approach would be to search in the high dimensional space for locations along the transitions where there are many other transitions close by (measured in the 200 PC space used to estimate speeds). The speeds at these shared locations could then be compared with speeds in the surrounding volume to determine if there is a slowing of the transitions through each. The number of slowing spots could then be compared with the number of spots without slowing, etc. It is not obvious that this would have to be limited to only those trajectories that were similar in some way. This is just one thought of an alternative analysis that could be done to check the claims related to the shared slow spots. I would welcome other alternatives. As it is, the current support is not that convincing as it is largely based on a single example with manual preprocessing steps.

The p-values shown in figure 2(f) and extended data figure 2(e) are not obviously valid. The shared slow spots were chosen as being local minima in the 1D pattern space and movement on the space should be continuous, so time and location should be related, making the test problematic. Also, please state the details of the paired t-test (e.g. H0: speed at $t=-2.5s$ is equal to speed at $t=0$).

All points above: we removed all descriptions and discussion of "shared slow spots", hypothetical unstable attractors and energy landscape plots, in favour of providing more detail about background matching.

8. * It would be nice to show stability of the plots in figure 2(h), perhaps on even versus odd trajectories. As an illustration it is cool but it is hard to determine how stable/meaningful it is. They could also be shown for the other slow spots.

See point 7. We removed this analysis, in favour of deeper analysis of background matching.

9. * It would be nice if the pattern-change speed distributions in 2(j) could be plotted separately for the first and second halves (in time) of the trajectories to show that the bimodality is indeed different than the phenomenon in 1(f).

This analysis (multimodality and lack of periodicity in trajectory speed distributions) is now shown in EDF. 5h-j, including the suggested split of trial blocks in separate plots (first and second halves, using the statistics in 1(f) (now EDF. 3f, see plots below)) to guide the trial block separation).

10. If 2(j), It is unclear if all three data sets rejected the unimodal test mentioned, perhaps the p-values could be listed.

Three datasets (animals: sepia219, sepia218, sepia213) rejected the unimodal test with $p = 0.016, 0.008, 0.005$. This is now stated in EDF. 5h: “All distributions rejected the unimodal test with $p < 0.05$.”

11. It is also unclear the relationship between the slow spots in c-h and the bimodality in i,j, especially since the “slow spots” are local minima of the speeds with a number of local minima being faster than local maxima in other parts. This could be made apparent in the text.

We thank the reviewer for identifying this potential issue. We have now normalised all trials by zeroing to their individual min. speed.

(RE)ORGANIZATION OF CHROMATOPHORE GROUPINGS DURING PATTERN TRANSITIONS

12. The clusterings of figure 3 are thought-provoking and the idea that individual chromatophores are rearranged into different groups in different trajectories is cool.

This result was indeed not expected.

13. ** In figure 3(e) and extended data figures 4(f), if I understood correctly, the distance on the x-axis is the average distance between pairs sampled from 2 clusters. I would think a metric more related to the idea in figure 3(d), showing two clusters can be “interdigitated” and not grouped together, would make more sense. The approach that comes to mind is to treat the activation patterns as densities (e.g. by binning or smoothing spatially) and then consider their overlap or the Wasserstein distance between the two as the metric for the x-axis of this plot. As it is, it is unclear how well the results support or not the claim except for the very few points to the left and right of the blob centered around 18mm, which are probably from pairs of small clusters.

We thank the reviewer for this suggestion. We now plot the Wasserstein distance on the X axis for 3 animals, demonstrating the lack of correlation between cluster spatial overlap and activity correlation (Figure 4d, shown below).

We now provide the data for each animal individually in EDF 7, together with the other pairwise distance metrics. The half-width of these animals was ca. 18 mm. Thus for left-right symmetric pairs of clusters, distances should be centred on 18 mm, as observed. We now provide this explanation in the legend of EDF 7c: “High dot density near 20 mm is explained by the half-width of the animals and the left-right symmetry of the pattern component pairs.”

14. I am curious about the points with correlation coefficients remarkably close to 1 in figure 3(e) – wouldn't these end up in a common cluster?

Chromatophores were clustered using community detection, which is based on relative local density of activity profiles. The correlation between two clusters in (old) Fig. 3e (now Fig. 4d)

was computed between the mean activity profiles of the two clusters. Because community detection incorporates the notion of inter- and intra-cluster density, it is possible to observe multiple clusters with mean activity profiles that have high correlation.

15. Please explain the “n” values in extended data figure 4(f).

“N” refers to the number of trials/trajectories. We have added this to the legend of current EDF 7c: “N indicates the number of repeated trials (trajectories) included in each analysis (Same Ns apply to the rest of plots).”

16. * The two examples in figure 3(a-d) and 3(g-i) are compelling and seem to support the conclusion that the chromatophores are re-organizing into different groupings during different transitions. I think, however, the argument would be strengthened if the comparison of the groupings could be made less dependent on the clustering algorithm (and hyperparameters). For example, it would be nice to see/compare the pairwise correlations matrices of the active chromatophores (for example, of those in 3(i)) across many transitions with all matrices sorted by the clustering from one transition. This would still be example-based but perhaps a few of these could be shown with some larger summaries.

We thank the reviewer for this suggestion. In EDF 6c we now plot the average pairwise correlations between same/other trajectories for individual components. In EDF 6d we now plot pairwise correlations, averaged over components, for individual trajectories. (New plots shown below.)

17. ** The decoding results in 3(f) come across as slightly misleading. If I have understood correctly, the plot shows a significant increase in accuracy using the clusters identified in the same pattern transition (decomp) versus using clusters identified from similar, all or other transition trajectories, etc. This comparison, however, appears problematic since, in the decomp case, the same data used for clustering is also used for decoding (i.e. in-sample accuracy), allowing for potential overfitting that is not possible in the comparisons. I would suggest removing the other comparisons unless a suitable way to account for this can be applied.

We apologise for the confusion. The point we wished to make was not to emphasise how much variance is explained by (over)fitting the decomposition, but rather how little variance is explained when we apply that (over)fitted model to other trials. Our intention is to provide additional evidence for the inter-trial variability and the lack of stability of the pattern components across trials. We have now clarified the results of this analysis (now in Fig. 4e, see below) in the text and modified the figure to illustrate this point only. In addition, we also contrast this result (variability of the decomposition of camouflage-pattern changes) with that obtained with blanching responses to visual threats (reliability of decomposition, Fig. 5k), and discuss the implications of this difference for possible motor control strategies.

18. The names on the x-axis of 3(f) were challenging to decipher from figure text, main text and extended figure text (still not entirely sure what “nearest” refers to). It is also not clear how many / which transitions were used to create these.

It is not clear how the x-axis is defined in figure 3(j) and extended data figure 3(g).

We apologise for the confusion. “Nearest” refers to the trajectory that has the lowest mean pairwise distance. This is now defined in Methods as ‘...for 2 trajectories of length M and N, we calculated the average of the M*N matrix of distances.’ To clarify the plot, we now write ‘nearest trajectory’, and list N = 21, 18, 21 trajectories in the figure legend (now Fig. 4e).

19. * Please remark on values of the mean intersection of unions (IoU) in 3(j). The mean IoU values are on average low even for similar trajectories but it is not obvious how those numbers compare to chance or what to expect from noisy clusters. Perhaps some context could be given for these numbers along with some additional metrics for clustering similarity, like the adjusted rand index. It would be nice to show that these are not sensitive to the method (and hyperparameter) used for clustering.

We thank the reviewer for the suggestion. In addition to reporting the results of a shuffling procedure in Fig. 4i, we analysed the data using the suggested adjusted-rand-index and Wallace distance metrics. Our results are robust to the choice of metric (EDF. 7f, g, see below).

We have also validated this analysis using several clustering resolutions (EDF. 6 g,h, see below).

CAMOUFLAGE PATTERNS ON CHECKERBOARDS

20. In figure 4(b), it seems odd to show the points in a UMAP embedding further projected onto the two axes. First, please confirm that the UMAP was originally 2D. Next, it would be nice to see the points plotted in the 2D UMAP. Finally, it is unclear what to understand from this plot, especially as UMAP is a nonlinear method, not necessarily preserving relative locations of the points. It appears to show more or less similar properties as in (a) but in a more confusing way and now with the additional data sets. Would it not be possible to simply add these points to figure 4(a)?

To better illustrate the monotonic gradient of observed skin patterns, we now plot the first principal component of the skin texture representation as a function of the checkerboard size, for three animals (Fig 2b, see below). Additional quantitative details are provided in the figure 2 legend, which reads:

“b. Skin patterns evoked by checkerboards of different spatial frequencies (half-periods 0.04-20 cm, only 0.08-10 cm shown) reveal a monotonic gradient of intermediate responses.

PC1 shows statistically significant stimulus-response relationship within the shaded region (0.31-1.25 cm, linear regression $r^2 = 0.50 \pm 0.04$, $p \leq 0.0001$, 3 animals, 4-8 trials per stimulus). In the three analysed animals, the first 4 (animal 1), 2 (animal 2) and 4 (animal 3) of the top 50 PCs are statistically significant ($r^2 = 0.40 \pm 0.03$, $p \leq 0.0001$)."

21. The x-axis in figure 4(c) appears to be mistaken as it goes into negative values for the skin-pattern periods.

In the text for figure 4(d), to make it easier for the reader, I would recommend relating the values in the matrix to the insets in 4(c).

In figure 4(e), it is not stated what is being shown in the bottom two rows.

In figure 4(f), are the points in each cluster “frames sampled at 25 Hertz over 46 seconds”? Also, why 46 seconds?

We have removed these plots.

22. The example clusters identified in figure 4(g) and ext data figure 5(i) appear compelling but are not easily tied into the results from figure 3. It would be great if this could be expanded on / discussed. Following the methods described for pattern transitions, perhaps similar pattern trajectories could be identified during transitions through the checkerboard backgrounds and then the trajectories visualized (as in 3h) using the clusters in 4(g).

We agree that the links between different experiments should be improved. Our checkerboard experiments focused on static patterns after the animal settled, rather than the dynamics of camouflage pattern changes. So we chose to tie these results to Fig. 1 (static) rather than Fig. 3 (dynamics). We thus reordered the figures, and clarified the relationship between the experiments in the text. We also report the results of our experiments on pattern matching with the full set of our 30 natural images. Note that this chromatophore-resolution analysis documents, contrary to previous results, the smooth tracking of stimulus spatial frequencies (in the range matched to the animal’s display apparatus, and to its size relative to the checkerboard’s frequencies).

TRANSITION INTO AND OUT OF BLANCHED STATE

23. In figure 5(c), it is not obvious that the data from the four animals can be combined to include a single fit. The y-axis is in arbitrary units that will vary from data set to data set. Given that a normalization would be tricky, perhaps it would be best to show the fit and corresponding r-squared values for each animal separately.

We removed this analysis from Fig 5.

24. The brown and purple colors in figure 5(g) are difficult to see.

We have changed the colours (see below).

25. As with the chromatophore clusters in figure 4(g), the nice clusters in 5(g) coming from multiple trials appear to contrast the unstable clusterings during similar trajectories in figure 3. It would be nice to have these come together.

We agree. We emphasise this point in the new manuscript (Fig 5k-l, EDF 9a-c).

26. While the animals in extended data figure 8 were not as compelling as the one in figure 5, I still found they supported the claim well.

We thank the reviewer.

OTHER COMMENTS:

27. The split of vertebrates and cephalopods being 600M years ago was mentioned in three locations, maybe one less would be OK.

We agree, and now mention this only once.

28. The definition of skin pattern space varied depending on the part of the analysis, making it difficult to follow. I made a list:

- UMAP (min_dist=0.8, n_neighbours=100) projection of the max pool layer of a pre-trained deep network responding to the images corresponding to a geometry preserving downsampling of the PCA (at 80%) of the images. <- for figure 1
- PCA to the first 200 dimensions <- for velocities throughout
- PCA to 2- or 3 dimensions <- for visualizations, evaluation of the trajectories, estimation of energy landscapes, etc.
- 1D locally linear embedding (LLE) of the top 5 principal components <- to identify slow spots
- PCA to 95% variance <- to study correlations in the checkboard section
- UMAP (min_dist=0.3, n_neighbours=20) projection of the "data". It appears that the data was first reduced with PCA to PCs preserving 95% of variance but this is not clear. <- for figure 4(b)

For these, it was not always clear if the pattern space referred to the space of all data collected for the animal or just a small subset. I would suggest finding a better way to summarize all of this.

We apologise for the lack of consistency.

We have now homogenised the methods. We use

- UMAP for visualisation as previously (fig.1)
- PCA: up to 200 dimensions to compute velocities throughout
- PCA: 2-3 dimensions for visualisations of the trajectories

The other analysis has been removed.

The robustness of the analysis comparing camouflage and blanching displays is tested over 2-200 PCs and shown in EDF 9b.

The numbers of trials/experiments performed and animals used are provided in the text/legends.

29. It would be nice to have plots of the explained variances as a function of the number of dimensions.

We added plots of explained variance vs. texture-space dimensions/PC# (EDF 3b, see a,b,c below), and chromatophore dynamics in camouflage vs. blanching (EDF 9c, see bottom row below). We also performed additional analyses of dimensionality (EDF 3); the approach is described in Methods and the results in the text.

30. It was not clear why the three clusterings of the chromatophores in figures 3, 4 and 5 were each done differently (50 principal components versus first 80%, different values for the number of neighbors and resolution) and in figure 5 the trajectories were first downsampled to just regions of high (top 10%) normalized speed.

We now use uniform parameter values for Leiden clustering. We display different clustering resolutions in figures 4 and 5, and provide evidence for the robustness of our results to this parameter in EDFs 6 and 9.

In Fig. 5, we restricted analysis to regions above 1/10 peak speed (not top 10%) to extract the behaviours of interest (pattern changes) in response to threatening stimuli, and exclude timepoints when the animal was set on a static pattern. We now explain this in Methods: *“Such trimming was performed to isolate the behaviours of interest (pattern changes) in response to threatening stimuli, and thus exclude timepoints when the animal was set on a static pattern.”*

31. That is it. I am sorry for the lengthy review. I hope the comments are helpful. Thanks again for the interesting article!

We thank the reviewer very much for his/her many useful suggestions and requests.

Referee #2 (Remarks to the Author):

Summary:

This manuscript by Woo et al. uses sophisticated methods to film and track individual chromatophores on the skin of cuttlefish over days-to-weeks of the animal's life. In contrast to the authors' previous study in Nature that used similar tracking and statistical methods (Reiter et al. 2018), the animals in this study exhibited camouflage behavior in response to different visual textures printed on fabric. Therefore, this study provided an exciting opportunity to dissect the relationship between the cuttlefish's visual stimulus and its camouflage output on its skin.

The major strength of this manuscript is the immense dataset, which required a highly impressive feat of engineering and computation to track the skin at chromatophore-level resolution during the fascinating behavior of camouflage. The field of cephalopod camouflage is in desperate need of more quantitative and rigorous studies. Therefore, this manuscript (with revisions) would provide an important contribution to the field. With that said, I am not convinced that the analyses the authors chose to perform on these datasets revealed much about the camouflage or visual system that we didn't know before.

1. Many of the conclusions are in line with previous observations (inc Reiter et al. 2018), and the most interesting conclusions didn't have enough evidence to make a strong statement (e.g. the suggestion that the system does, in fact, have feedback, contrary to previous studies). Finally, the big advancement in this study is the addition of camouflage behavior (as opposed to the non-camouflage pattern changes seen in Reiter et al. 2018). The authors dedicated surprisingly little of this study to the investigation of camouflage - thus, this paper doesn't appear to be a major advancement over the previous study.

For these reasons I do not recommend this manuscript for publication in Nature.

We apologise and regret the lack of focus in our initial manuscript's narrative. As indicated in the introduction to this "Response to Reviewers", we have reorganised the manuscript to now place our results explicitly in the context of previous work.

We respectfully disagree, however, with the reviewer's comment that our results provide no new advances over our previous study—in which we reported new methods to study cuttlefish patterning, but did not study camouflage.

Our present manuscript, based on the quantitative methods mentioned above, provides entirely new results:

First, our results are inconsistent with published results reporting the low dimensionality of skin pattern space or the coarse graining of spatial frequencies. These two new results are, on their own, very significant because they suggest a much greater complexity and flexibility of the camouflage system than previously proposed.

Second, we provide entirely new results on camouflage pattern formation by cephalopods, exploiting our unique high-resolution analysis techniques (in time and in space). These results concern in particular:

1. the dynamics of camouflage pattern matching, suggesting a progressive operation requiring feedback; this has never been shown;
2. the flexibility and variability of pattern decomposition, suggesting the absence of pre-determined motor strategies; this is entirely new;
3. the evidence for a separation (and possible superposition) of control systems for camouflage and blanching displays; this is also entirely new.

We hope that this revised (and more detailed) version of our manuscript will convince the reviewer of the novelty of our results.

General feedback:

2. - I believe the subject of cuttlefish pattern change/camouflage is of interest to people inside and outside of the field, but the manuscript is written in a way that makes it very difficult for scientists outside of computational/statistical disciplines to understand it. A significant rewrite is required to make the manuscript understandable to a broader audience. For instance, each section of the results should begin with some motivation for the particular analysis being performed (i.e. it should not begin with a description of the figure, e.g. "Figure 3a,b illustrates the evolution of camouflage..." - L183).

- There is a general sense that the authors created an impressive dataset but then weren't sure what to do with the data, so tried a number of analyses (with unclear biological motivation), and included all of those figures in the manuscript. Many of these figures are not necessary and should be moved to the supplement.

We agree with the referee. We could and should have a better job of explaining the rationale for our experiments and should have done more than simply reporting our observations. Our new manuscript has thus been reorganised, omitting some sections and expanding others.

3. - A lot more care is needed in generating figures - it took far too long to dissect most figures. There should be fewer figures, they should be larger, have legends in the figure itself, and all axes should be labeled. The text legend needs to be clearer, and the main text should explain what analysis was performed to create each figure.

As part of our rewrite, we removed some analyses and figure panels, to simplify and clarify the narrative.

4. - I recommend that the authors be careful in their use of the word "texture" to mean "pattern". Given that cuttlefish can change 3D texture to match their surroundings, the title could be misinterpreted (as well as many other instances of the word). Perhaps use "visual texture"?

We agree with the reviewer. We now use 'visual texture' in the title, and specify that we are referring to a visual signal when discussing texture representation. Additionally, we now write in Methods:

‘Texture representation’ in this study refers to 2-D visual textures. Cuttlefish can produce different 2-D textures through chromatophore activity, and also alter their 3-D appearance through postural motion and contraction of papillae¹. These 3-D alterations have effects on camouflage and alter the 2-D visual texture of the cuttlefish. These are detected by and incorporated into our LR analysis.

I will address each of the major conclusions the authors made.

5. 1. “Camouflage-texture space is high-dimensional” (L41)

Figure 1 shows substantial complexity in the skin patterning system, which suggests camouflage does not consist of the many-to-few mapping that has been indicated by other researchers in the field. However, neither the main text nor figure explicitly addresses whether camouflage is high-dimensional. If this is a major conclusion of the paper, I suggest the authors explain this in the main text.

We thank the referee for this suggestion. In addition to reporting variance explained vs. principal components (EDF 3b), we have now performed the best-practice analysis suggested in ref (2), establishing the linearity of texture space and estimating dimensionality with “Parallel Analysis” (see Methods). We now discuss this in greater depth in relation to Fig. 1 and in the methods section.

6. For the entire figure, the presentation of data could be greatly improved. For instance:

- After showing Fig 1A, I would show 1E to make it clear what the UMAP plot is indicating. Then I would provide the analysis (1B-D). However, I find plots 1B-D difficult to decipher, especially because the reader has to make constant reference to the 1E plot, and the colors superimposed on each other are difficult to see. I would find an alternative method to represent this data (or provide a magnified view of one cluster). The color codes should be included as a legend in the figure itself.

- Figure 1E: the authors state that this represents a great diversity of camouflage patterns, but at the resolution provided, they don't look very different (other than 4 & 6). Perhaps the authors should show a single (larger) image for each pattern, and include the 3x3 panel in the supplement.

We have simplified Fig.1. Parts 1b-d have been removed. The images of skin patterns in different regions of pattern space have been enlarged. We show an array of 8 3x3 panels of

skin textures to better illustrate the significance of the UMAP embedding plot (different skin patterns in different regions of visual texture space; similar skin patterns in nearby regions of the space).

7. - Figure 1H: the authors mentioned that there was sometimes image blur when the cuttlefish swam during a fabric change, and they said “we could usually track skin patterning at low resolution when the animal swam during these transitions”. This implies that the authors could be missing patterns in other regions of texture space, so I don’t think they can support the conclusion made for Figure 1H.

Figure 1H has been removed. The previous text was misleading: animal movement affected our ability to track at high spatial resolution, but not at the low resolution we use for visual-texture-space analysis.

8. 2. “The search for a pattern ultimately accepted by the animal was not stereotyped. Rather, patterns meandered through texture space, decelerating and accelerating repeatedly during each search, before stabilizing around a chosen, noisy pattern.” (L42)

There are a few issues with this statement:

i. I agree that the data indicates animals used differing paths to reach a given pattern. But it’s not clear that the data shows stabilization (especially since we can’t see how long each trajectory is, or where each specific trajectory begins and ends). It would be useful to see a plot of the animal’s skin change in response to each stimulus over time, so the reader can see the stabilization as a function of time (in seconds).

We have greatly expanded our analysis of the meandering nature of camouflage dynamics. We plot the overall time course of skin pattern changes in EDF 3d-f (below left). Figure 3 now shows example trajectories, and analyses variable path direction and increased dwell time or number of pauses as the final camouflage is approached (below, right).

9. ii. It’s not clear what the biological relevance of the accelerations and decelerations are, and the authors don’t provide any motivation for this analysis.

We now motivate this analysis with a schematic (Fig. 1a), additional quantitative analysis (Fig. 3), and discussion:

“...the paths (in pattern space) taken during a camouflage change were tortuous, intermittent—consisting of alternating pattern motion and relative stability—and not stereotyped. The number of pauses and their duration increased as convergence neared. At each intermittent motion onset, pattern motion tended to aim towards the target camouflage, reflecting the animal’s instantaneous state rather than its initial camouflage at the onset of the behavior. Together these results suggest that camouflage relies on feedback during the approach to an adaptive pattern, more akin to correction of hand reaching movements in primates^{3,4} or of tongue reaching in rodents⁵, than to ballistic motion to a memorized target.”

10. The authors even say “it is too early to say how illuminating this framework might be” (L322). Furthermore, the pauses are very small dips in speed (e.g. Fig 2F, see below).

We have removed hypothetical mentions of attractor dynamics.

11. iii. The authors suggest the stabilization of the system indicates there is feedback via sensory or via corollary discharge (L318 and L350). However, while this is an interesting suggestion, this statement requires evidence of such feedback.

We have deepened our analysis of pattern dynamics (see 2i and 2ii above) and added new results. We reason that if feedback were not necessary—i.e., if an animal adopted a target skin pattern conditioned uniquely on a “target visual input”—then the animal should have an internal strategy to reach that target. We should therefore expect reliability in the dynamics of skin pattern change. We observed precisely this when an animal responded to visual threats (blanching behaviour) (Figure 5). In the context of camouflage, however, we observed dynamics more consistent with the reliance of feedback: meandering paths with pauses in pattern space; changes in direction of motion in pattern space; gradual convergence to an increasingly better match between skin pattern and background visual texture; longer dwell time as the animal’s texture approached its goal. This new evidence has been added.

12. Additional figure comments:

- The conclusion that cuttlefish used differing paths to reach a given pattern is in contrast to Reiter et al. 2018 (Fig 4), where the authors tracked the cuttlefish’s response to motion. They state “upon each stimulus, the animal not only generated the same target patterns but also moved with chromatophore-level repeatability through very similar, low-dimensional sequences of intermediate states”. The authors should supply an interpretation of this difference.

We agree and now devote more text and figures (Fig 5k-l, below; EDF 9a-c) to describing this contrast in repeatability and discussing its implications for chromatophore control strategies. The strategies used in the context of camouflage (Figs. 1-4) differ substantially from those used in response to threat displays (blanching, Fig. 5).

13. - Many of the trajectories are not visible because they're all superimposed on each other. If the authors want us to assess the similarity of trajectories, we should be able to see them.

We agree and now plot fewer trajectories in Figs. 3 and 5. Additional plots (including all trajectories) are provided in Extended Data Figures.

14. - Figure 2F: the data underlying these small changes in speed (colored lines behind the black line) have massive variance. I don't think the authors can make a strong case for their conclusion with this data.

This analysis has been removed.

15. - It's not clear what value 2H adds to the paper. I would put this in the supplement unless the motivation for looking at this system as an energy landscape is provided. Also, a direct reference to this figure is missing in the text.

We have removed Figure 2H and mentions of this energy landscape.

16. 3. "During pattern motion, chromatophores could be grouped into "pattern components", based on their covariation. These components could take many shapes and sizes, and overlay one another or not" (L47)

This was established in Reiter et al. 2018 Figure 3. It's not clear what significant new information this figure offers, especially since the authors don't relate this analysis to the major difference between this study and the previous - the animal is now camouflaging.

The conclusion of Fig 3E - that pattern components can be physically clustered or loose and distributed - is very similar to a conclusion of Reiter et al. 2018 (Fig 3D) so I found it to be an expected result.

We did not study camouflage in Reiter et al., 2018. In our current study, focused on camouflage, we provide evidence that pattern components are unstable in the context of camouflage, and get reconfigured in each trial or instantiation of a camouflage transition. We now provide more data to substantiate the process of background matching, and explicitly contrast camouflage behaviour (variable) with blanching responses to visual threats (highly repeatable).

While Fig. 3d in Reiter et al. 2018 described a distribution of physical spread of motor elements, Fig. 3e of the present manuscript (Fig. 4d in the revised version, with a new distance metric) shows that “the degree of pairwise correlation between components” is “independent of their spatial overlap”. The existence of interspersed but independently-controlled large pattern components serves as the basis for reorganisation, which we further detail in the remainder of figure 4.

17. 4. “Two transitions between similar starting and ending skin patterns typically invoked very different components, suggesting neural flexibility in the realization of pattern search” (L50). “Identifying the smallest, consistent components of camouflage patterns will require even larger datasets” (L332).

It is an interesting result that two transitions between similar patterns involved very different components, and this adds to the conclusion that camouflage is very complex (and much more so than suggested by the majority of previous studies in the field).

We agree with the reviewer.

18. It seems to me that identifying the smallest component (the effective pixel size) is one of the most interesting questions that should actually be identifiable using this dataset. The authors could focus their analysis on a region with a small pattern component (e.g. a spot).

We agree. We examined this question in Reiter et al. 2018: analysis of chromatophore covariation revealed small sets of spatially localised chromatophores (3-10 typically) that co-fluctuate and therefore are most likely driven by one or a few common motor neurons, as suggested by prior anatomical work.

In the current manuscript we describe, in the context of camouflage, the flexible co-fluctuation of much larger groups of chromatophores (“components”), each presumably composed of many of these motor units. (A separate on-going project in this laboratory is focused on describing the micro-control of chromatophores, but this is beyond the scope of the present manuscript.)

19. Figure comments:

- There are more issues with interpretability: the colors aren’t all visible on the animal’s skin in Figures 3A and B, which leads to a lot of confusion about what the reader is looking at.

We now show a different and less complex trajectory for illustration, enabling a better discrimination of the colours on the animal’s skin.

20. - Fig 3E: It’s not clear why the distance between chromatophores is concentrated at 20 mm. This implies that there is a relationship between the physical distance on the skin. A clear explanation of how this plot was generated is missing in the text.

We now provide an explanation for this clustering in Methods, and write, in the legend of EDF 7d, ‘High dot density near 20 mm is explained by the half-width of the animals and the left-right symmetry of the pattern component pairs.’

21. - Fig 3F is not clear at all. The x-axis abbreviations and descriptions are not provided.

Now provided in figure legend (now Fig. 4e), and in Methods.

22. - Fig 3G: the images of this pattern change may look similar, but the square appears at a different time, and the resolution is sufficiently low that I can't really see the change. So I'm not surprised the trajectories are somewhat different.

We now clarify that this analysis was motivated by the proximity of their starting and ending points, as well as the shared occupancy in chromatophore space, in addition to the visual similarity of the pattern transition. We agree that the trajectories (in this projection) are not identical. But here our analysis is concerned with the level of chromatophore-component re-organization, something that is not a simple consequence of having different trajectories (different trajectories could obtain, for example, with a different recruitment order of the same components).

23. - By the end of Figure 3 it seems the authors are just making the same point multiple times. I would put some of this data in the supplement as it doesn't add to the paper.

Given its unexpectedness, the surprising re-organization of components requires, we believe, a convincing demonstration. We therefore provide examples of the phenomenon in Figs. 4f,g and broader support for the repeatability of the phenomenon (in Figs. 4 h,i).

24. 5. "Using checkerboards spanning a wide range of spatial frequencies, we identified clusters of chromatophores whose state co-varied with input spatial frequency" (L333).

It is useful that the authors quantified the response of cuttlefish to changing spatial frequencies, but this result is in keeping with previous studies using checkerboards as visual input. Thus, this result is not surprising.

We have rewritten the manuscript to better explain the novelty and significance of these results. Previous work showed that animals modulate their skin pattern in response to checkerboard spatial frequency, moving in a few discrete steps as the spatial frequency of the stimulus was varied.

In light of the high-dimensionality of pattern space (Fig. 1), we reasoned that the published coarse-grained responses to checkerboards could reflect the low sampling intervals of the stimulus set used in these published experiments, rather than the limits of the animal's camouflage abilities.

Using a finer sampling of background spatial frequencies (larger stimulus set), we could reveal a smooth gradient of skin-pattern spatial frequencies (new), and the differential sensitivity of individual components (themselves labile) to spatial frequency.

25. Figure comments:

- Figure 4B: what is the y-axis? There should be an explanation/legend of what red and white are. Is there a different way of showing this instead of UMAP_0 and UMAP-1?
- Fig 4C is underwhelming and hard to see an effect. Is there a way to plot this another way that makes the effect more clear?
- Fig 4D: It isn't clear what is being shown.
- Fig 4E: what's the y-axis? It's unclear if the mean power values are significant.

These analyses have all been removed.

26. 6. "Analysis of patterns evoked by threats of camouflaging animals revealed that a memory of the initial camouflage remains upon withdrawal of the threat" (L339).

This is an interesting result but there is the caveat that the animal may be returning to the same state after blanching because its environment hasn't changed and it's choosing the appropriate response (as opposed to retention of the previous state). The authors should include a control. For instance, they might change the background while the animal is experiencing the threatening stimulus to see if the animal still returns to its original state before updating its skin pattern.

We agree that this is an interesting result, and have removed the loaded term 'memory', which confused our argument. We argue, based on the experiments described in Fig. 5d-f and Extended Data Fig 8d, that the blanched pattern contains information about previous (and in most cases, future) camouflage patterns. This suggests that a camouflage pattern typically persists throughout a trial. A simple explanation is that blanching is superimposed on camouflage by a separate control system. The different dynamics of blanching and camouflaging support this interpretation.

We agree that the persistent trace of camouflage was probably explained by the fact that the background had not changed; we did not mean to suggest that the animal returned to a previous state, stored in memory. Rather, we suggest that chromatophore patterns must be under the simultaneous control of more than one neural system. We propose a possible site of convergence of the two in the final paragraph of the discussion:

"Blanching represents the shrinking of chromatophores, caused by the relaxation of the chromatophore muscles. By contrast, the return to a camouflage pattern requires the differential expansion of chromatophores, by contraction of those same muscles. Blanching could therefore be generated by a transient and general inhibition of the chromatophore motor drive, downstream of the camouflage control level; but because recovery from blanching reveals components with different dynamics, this putative inhibition probably acts upstream of the motoneurons (at an intermediate level of chromatophore control) rather than directly on them. In conclusion, camouflage control in Sepia appears to be extremely flexible and non stereotypical when analysed at cellular resolution. The dynamics of its output suggest the use of feedback to converge onto a chosen camouflage. These results will guide mechanistic studies of this remarkable texture-matching system."

27. The authors could also alternate two backgrounds, and plot the variance in skin pattern when the animal returns to the same pattern each time. Then, compare that to the animal returning to its skin pattern after blanching. Presumably the accuracy of the pattern reproduction should be much higher after blanching compared to post-camouflage change. It is too strong a statement to say that the animal has a memory of the initial camouflage.

As mentioned above, we did not mean to suggest that memory guides the return path, but that the non-stereotyped blanch patterns contain traces of the past camouflage states. We apologise for the confusion caused by our use of the term “memory”.

28. Methods comments:

- What size of animals did you use for the experiments?

We use animals with mantle lengths ranging from 42 to 90 mm. We now state this in the Methods section.

29. - What type of lighting did you use for the behavior system?

We used four LED strip lights with diffusers, mounted 15 cm above the arena (SAW4 white, 698cm length, Polytec GmbH), providing an illuminance of 3400 lx at the centre of the arena lid. We now state this in the Methods section.

30. Conclusion:

The authors created an important dataset and developed highly sophisticated methods to study the skin of cuttlefish during camouflage. However, the analyses chosen by the authors fall short of the promise of this dataset. Too many of the conclusions either confirm previous studies, are not of notable significance, or require additional experimental support. For those reasons I think this study is not suited for publication in Nature.

We hope that our explanations, rewrite, new experiments and new analyses, will convince the reviewer of the novelty of this study.

Referee #3 (Remarks to the Author):

A. Summary of the key results

Woo et al. present a dazzling array of data to convince the reader that chromatophore control in cuttlefish in response to visual stimuli is extremely complex (not just three patterns) and from my interpretation controlled by a non-linear system at times. They were able to complete this study due to a previous publication demonstrating the method of tracking of 10,000s of chromatophores simultaneously.

They present several findings as new from their study including (1) patterns expressed are in a vast array of forms, (2) patterns are not stereotypical (though they contradict this later), (3) pattern expression changed over time and the speed of changes altered over time, and

(4) different extracted patterns resulted when different stimuli were used, such as natural, vs checkerboard, vs looming, which they hypothesized was due to different control systems.

1. B. Originality and significance: if not novel, please include reference

- The method of tracking chromatophores is not novel and was published in Nature 2018 (<https://doi.org/10.1038/s41586-018-0591-3>).
- The relationship between visual stimuli and expression of skin patterns is not novel, as evidenced by many papers from the labs of Roger Hanlon and Daniel Osorio (See the massive number in Hanlon & Messenger 2018. Cephalopod Behaviour. Cambridge University Press).
- However, the novelty of this study would have been the utilization of a major increase in the spatial and temporal resolution in their tracking method to precisely quantify the relationships between visual stimuli and chromatophore activity at the level of single chromatophores (or small patches of chromatophores).
- The significance is they have demonstrated that chromatophore control systems in cuttlefish are very sophisticated (more than previously published) and their analysis seems to suggest they are producing highly tailored camouflage patterning across the dorsal surface with respect to specific visual stimuli.

We thank the referee for appreciating the advantages conferred by the methods which we developed and now use to study this remarkable behaviour.

We respectfully disagree, however, with the statement that our analysis of the relationship between visual stimuli and expression of skin patterns is not novel.

Our present manuscript, based on the quantitative methods mentioned above, provides entirely new results:

First, our results are inconsistent with published results reporting the low dimensionality of skin pattern space or the coarse graining of spatial frequencies. These two new results are, on their own, very significant because they suggest a much greater complexity and flexibility of the camouflage system than previously understood.

Second, we provide entirely new results on camouflage pattern formation by cephalopods, exploiting our unique high-resolution analysis techniques (in time and in space). These results concern in particular:

1. the dynamics of camouflage pattern matching, suggesting a progressive operation requiring feedback; this has never been done;
2. the flexibility and variability of pattern decomposition, suggesting the absence of pre-determined motor strategies; this is entirely new;
3. the evidence for a separation (and possible superposition) of control systems for camouflage and blanching displays; this is also entirely new.

We hope that this revised (augmented in parts, and more detailed) version of our manuscript will convince the reviewer of its novelty.

2. C. Data & methodology: validity of approach, quality of data, quality of presentation

The tracking method is accurate from what was reported in Nature and this manuscript. It is producing some very detailed information about the control of cuttlefish skin patterns. But there are some major problems in the presentation of the research in this manuscript and potentially how it has been analyzed.

- The visual stimuli, particularly the natural types, are not presented or even described and related to each figure. Without this information, I cannot properly review the paper. I thank the authors for providing the stimuli, later on. They really should be included with the paper for publication and each data analysis and figure be referred to the given stimuli. Also, given their diversity of the stimuli, there should be some relationship made between the patterns expressed on the skin and the visual stimuli.

All the visual stimuli are shown in EDF 2. We refer to these stimuli in the revised text.

We performed many new analyses of the relationship between skin patterns and visual stimuli. For example, we now added evidence for the tight correlation between stimulus and response with natural textures in Fig 2a and with reduced stimuli (checkerboards) in Fig. 2b.

In EDF 4e-g, we motivate our choice of visual texture metrics, showing how they are more predictive of animal-background matching than even optimal combinations of lower-order image statistics.

In Fig. 3, we study the dynamics of animal-background matching over time.

- 3. • Cuttlefish express papillae dynamically which alter skin texture and as the authors noted, the extraction of chromatophore data. The word papillae was only mentioned once, in the methods, and due to their presence, 19-21% of the chromatophore data was disregarded. This seems like a major limitation and it should have been highlighted throughout the introduction, results and methods sections.

We apologise for the misleading statistic. We now discuss papillae in the main text as part of our discussion of (2D) visual textures, and in the methods. The complex 3D motion of papillae is indeed problematic for our chromatophore tracking algorithm, which relies on locally affine image transformations. This was one motivation for combining HR approaches

(excluding papillae) with lower-resolution visual texture representations, allowing us to incorporate papillae expression.

Our results at chromatophore resolution do not depend on tracking all chromatophores which, as we did and still state, we do not do. (Tracking this high a fraction of participating cells is, we believe, unparalleled.)

The 19-21% rejected data refer to “putative chromatophores”, and included mostly regions at the edge of the mantle and on the animal’s head—which we remove through masking. Note also that due to initial setting in our segmentation and stitching pipeline, the initial number of putative chromatophores is often overestimated.

The new methods section better describes our masking procedure and reports, for all the figures, the fraction of the animal’s dorsal mantle that was not trackable at chromatophore resolution due to papillae or other non-affine motion.

4. • The title of the paper and throughout uses the word texture, but given that cuttlefish can alter the texture of their skin then texture should be defined very clearly early in the abstract and introduction to avoid confusion. Texture of the stimuli should also be defined as purely 2D as cuttlefish can resolved depth and use this information to express papillae. Finally some of the naturalistic stimuli contain visual depth cues, like shading or perspective, that the cuttlefish will perceive as depth information, and this would then add more complexity to the skin pattern and texture expressed, and the analysis and discussion.

We agree with the reviewer. We now use ‘visual texture’, and specify that we are referring to a visual signal when discussing texture representation. In addition, we added in the Methods:

‘Texture representation’ in this study refers to 2-D visual textures. Cuttlefish can produce different 2-D textures through chromatophore activity, and also alter their 3-D appearance through postural motion and contraction of papillae¹. These 3-D alterations have effects on camouflage and alter the 2-D visual texture of the cuttlefish. These are detected by and incorporated into our LR analysis.’

5. • The manuscript discusses patterns, but most of these instances of the word pattern are not referring to specific arrangements or configurations of expressed chromatophore patterns, but rather they are implied patterns that were extracted from the UMAP analysis.

The analysis of chromatophore activity without providing the ground truth that UMAP changes and variances are linked to specific patterns is not clearly shown (only Fig 1e, and there is no analysis of difference with and between the 8 selected clusters) and so this makes understanding the context of the UMAP and related data very hard.

Our visual texture representation of an animal’s skin pattern is independent of our UMAP embedding. UMAP was used only for visualisation (currently Fig. 1c) because it allowed us to embed a high dimensional space into 2D. Using a neural network to generate a

representation of visual texture is a novel analysis in the context of cephalopod camouflage. It lets us analyse animal skin patterns and background images within a common space, one that has been shown to be perceptually relevant through psychophysical studies⁶. We believe that this type of analysis is critical to understand camouflage matching, even if future technological developments allow for all chromatophores to be constantly recorded.

To address the reviewer's concern that our visual texture representation reflects differences in chromatophore patterns poorly, we now show that skin pattern dynamics described using this texture representation correlate well with chromatophore population dynamics (EDF 5c-f).

6. Pattern should be very clearly defined and differentiated from those extracted from correlations in data analysis and those that statistically differ on the skin of the animal (which relates back to error rate in segmentation, analysis, significant change in chromatophore diameter, etc.).

We agree with the reviewer, and clarified our terminology. In the abstract we define the process of camouflage pattern matching in general:

'This behaviour relies on a visual assessment of the animal's surroundings, on an interpretation of visual-texture statistics⁷⁻⁹, and on matching these statistics using millions of cellular pixels called chromatophores, located in the skin and controlled by motoneurons located in the brain¹⁰⁻¹².'

We analyse these skin patterns at 2 levels: visual textures and chromatophore population. We are careful to state which level is being analysed throughout the text, referring to 'texture space' and 'chromatophore space'.

7. • I was also expecting that the chromatophore patterns from each trial and each animal would be warped to a common cuttlefish ground plan (shape) allowing analysis of different skin neural control. Also allow different levels of neighborhood analysis that could be increasingly pixelated to compare to visual resolution of predators at different distances (the main driving force for camouflage). From my reading I did not see this type of approach or discussion.

Sadly, the stochastic nature of chromatophore addition into a cuttlefish's skin means that chromatophores can not be mapped precisely from one animal to another. Further, the spatial overlap and reconfiguration of skin pattern components we describe in Fig. 4 precludes this interesting suggested analysis.

8. • I was also expecting this type of common ground plan or framework analysis to link back to the stimuli presented to understand what image statistics link to which patterns expressed. It was really surprising to see such low r^2 correlations between checkerboard patterns and chromatophore patterns (fig4).

We performed additional analysis of camouflage matching (Figs. 2, 3) to address the reviewer's comment. The r^2 values in Fig. 4 (now Fig. 2c, inserted below) and in EDF. 4a-c concern the activity of particular groups of chromatophores (components defined by community detection), not the entire chromatophore population. Following previous work (by others) on this issue, we expected to find a discrete and small number of skin patterns and a step function relationship between chromatophore activity and checkerboard size. We therefore consider the reported r^2 values to be high. We clarified the text to better place this result in context.

9. • The authors do not cite, nor do I know of, papers that show sensory feedback from the skin to the brain. In this case it is hard to see how changing the chromatophores from one state to another over time constitutes a 'search'. It is much more parsimonious to consider they are taking a long time (relatively to the time to change a chromatophore state) to decide what pattern they finally decide to express. Given this possibility, it should be clearly defined what they mean by search. Given we also do not know what cuttlefish are thinking about, I suggest that search be replaced with "matching phase" or something similar that is linked to the action rather than implying an unknown mechanism.

We have added a definition of 'search' as the 'process of pattern matching' in the text.

Our new analysis (Fig. 3) describes how this search is characterised by a series of meandering fast movements and pauses, with movement direction gradually converging on a target pattern and pause frequency increasing as the target pattern is approached. We discuss how these dynamics are more consistent with a strategy involving visual feedback (see also responses to reviewer 2). We agree that this has not yet been explored mechanistically, nor has it been described phenomenologically before.

10. • There seems to be a mismatch with the link between stimuli and chromatophore expression patterns. How much did the view of the visual stimuli differ (between trials and between animals)? Could any of these differences be averaged out and so be ruled out experimentally as not being important?

We now provide analysis showing the strong relationship between background visual texture and animal camouflage (Fig. 2a,b). One of the motivations to use checkerboard stimuli is that they provide homogeneous visual information independent of the direction of view. For the natural-image experiments (Fig. 2a), which involved stimuli that were less regular or homogeneous, we defined the background visual texture by averaging from a random sample of >6 animal-sized patches around the animal. This is now documented in the Methods section.

11. • Stereotypical responses query: Line 41 “...we found that camouflage-texture space is high-dimensional and that the search for a pattern ultimately accepted by the animal was not stereotyped.” Yet on line 111, “Fig. 1c is in fact composed of several smaller clusters, each corresponding to the response of the animal to a different presentation of the same natural image. Variations within a cluster represent skin-pattern noise: patterns are never static, because chromatophores often flicker and undergo small local fluctuations.” These two statements seems contradictory to me (along with others in the manuscript), that perhaps the UMAP method is too detailed (or at too fine a scale) due to the vast amount of data and the interactions between data points that are non-linear, that perhaps the method and the conclusions are hard to parse. I just don’t know without more clarity in the writing and the presentation of the variance relative to the biology and the visual stimuli Provided.

We apologise for our initial lack of clarity and hope to have remedied this in this revision (in particular in EDF. 3a-c).

These results are not contradictory but the consequence of two observations: first, camouflage patterns depend on background, as we develop in new analyses in Figs 2 (static) and 3 (dynamic, search in pattern space); second, animals typically adopt different camouflage patterns, even when they are repeatedly exposed to the same background. These differences are not due to chromatophore flickering (which exists but contributes little to our measurements in texture space, which are not sensitive to individual chromatophore variations). These two facts together contribute to the high dimensionality of skin pattern space.

Our dimensionality estimates are not merely a consequence of the vast amounts of data, as we show by contrasting the dimensionality of camouflage pattern search (not stereotyped) with that of responses to visual threats (blanching, Fig. 5c,k,l, and EDF. 9a-c) (stereotyped).

Note finally that UMAP (Fig 1) is used only for display purposes, and not for the analysis of dimensionality or response stereotypy (see Methods).

12. • The paper focuses a lot of capabilities of the cuttlefish but does not put it into context whether what is analyzed and plotted is normal or atypical. The link between capability and repeated tests from different animals should be clear. In many cases the figure legends do not describe the number of animals tested, the number of trials tested, which is the stimulus used (linking back to its picture), whether the stimulus is novel or repeated, etc.

We apologise for the lack of clarity. We now report these values throughout the manuscript where applicable—see for example Statistics and Reproducibility section in Methods, which reads:

Unless stated otherwise, data are mean \pm s.e.m. Experiments were repeated independently several times with similar results, with numbers of repetitions as follows:

Skin pattern texture space analysis (Fig. 1, Extended Data Fig. 3a-c) was carried out in ten animals, six of which (each with at least 20 analyzable trials of swift background change) were included in the analysis of background change (Extended Data Fig. 3d-f). Natural-image experiments (Fig. 2a, Extended Data Fig. 4d-f) were carried out in three animals with 8 to 12 repetitions each. Checkerboard experiments with dense sampling (Fig. 2b-c, Extended Data Fig. 4a-c) were carried out in three animals with 4 to 14 repetitions per stimulus in each animal. Three animals (14, 30, and 29 repetitions respectively for 6 types of background changes) with high-quality HR data were included in the analyses of chromatophore space (Fig. 3-4, Extended Data Fig. 5-7). For each animal, experiments were conducted in 2 to 3 experimental sessions on separate days. Threatening visual stimulation (moving hand or looming image display) experiments (Fig. 5, Extended Data Fig. 8-10) were carried out with four animals in 1 to 4 experimental sessions on separate days, yielding 11, 22, 19 and 9 trials with high-quality HR data.

13. D. Appropriate use of statistics and treatment of uncertainties

There is a lot of data analysis in this paper but without a better understanding of the variance and its link to the biology it is very hard to comment on the statistics and uncertainties.

We have added new analyses of variance, across animals and across experimental conditions (e.g., Figs. 4e, 5k, EDFs 4d-f, 5, 9).

14. E. Conclusions: robustness, validity, reliability

See above, without the stimuli linked to the figures, and the data analysis performed that highlights the relationship between stimuli and the expressed chromatophore patterns it is hard to understand if the conclusions are warranted.

We now plot the relationship between the visual texture of natural images and that of skin pattern in Fig 2a. Fig. 2b reports the relationship between spatial frequency and skin pattern using checkerboard stimuli. EDF. 4d-f examines and details the methods tested and motivates our choice of visual-texture metrics. We show how these metrics are more predictive of animal-background matching than even optimal combinations of lower-order image statistics. Finally, in Fig. 3, we deepened our analysis of the dynamics of animal-background matching over time.

15. There is also no easy way to parse whether temporal fluctuation variance in chromatophore expression is due to (1) encountering a new environment or visual scene, (2) still deciding what pattern is most suitable to express across their body, (3) variance within a stable expression pattern that has been chosen, and (4) refinement of a suitable match (expressed for some time), due to noticing new details in the scene presented. Many cephalopods are constantly vigilant and scanning their surroundings with slow head and eye rotations. While the authors have made some analyses which may cover these options, they are not clear to me, particularly relating to variance in the dataset.

We thank the reviewer for highlighting this issue. While there is indeed a level of constant chromatophore variation, we find much larger variation in the context of camouflage pattern changes induced by background change, or following locomotion. We now analyse the time course of these changes in EDF. 3d-f. In both cases, animals demonstrate ~20-100 seconds of dynamic skin pattern change before settling into a new stable pattern. Refinement of background matching happens during this transient phase, which we now study in Fig. 3. We have modified our text to clarify which skin pattern changes happen in what contexts.

16. Line 315: “How then does an animal know that it has reached a suitable pattern?” This question raises another question, Does the animal need to know it has reached a suitable pattern? In the wild if it got it wrong the animal may be eaten by predators very quickly. Predators swimming past without noticing its presence would be a very good indicator and so Line 318 suggesting feedback does not even have to exist. It is also unclear which data suggests feedback exists.

We have modified our text to better motivate our discussion of feedback. We reason that if an animal chose, during its initial visual exploration of its environment, a target skin pattern

that best matches that environment among a small set of possible skin patterns—one of the three classes of previously proposed patterns: disruptive, mottled and uniform, as has been argued—then the animal could be expected to use an open loop strategy to reach that selected target skin pattern. (As the reviewer suggests, these targets might indeed be the result of natural selection.) One expectation of this hypothesis is that dynamics of pattern change should be direct and repeatable. This is what we observed with blanching, a stereotyped response to visual threats.

In the context of camouflage, however, we observed dynamics more consistent with the existence of repeated comparisons between input and output: meandering paths in pattern space with increasingly frequent pauses, directed adjustments, gradually converging to a match between skin pattern and background visual texture. The results therefore suggest that, during camouflage, but not during blanching, the animals use feedback. We discuss possible mechanisms in the discussion.

17. Line 327: “apparent “pattern components”, whose composition could be small or large, clustered or loose,...” This text apart from suggestion multi-scale control, also suggests the possibility of low dimensional outputs given the right conditions or experience (learning from predators). Thus it seems linking this back to previous work as parallel theory should also be posed.

We agree, and now devote a larger fraction of the manuscript (Fig 5c,k,l, EDF. 9) to explicitly contrasting the flexibility of camouflage behaviour with the repeatability of responses to visual threat.

18. Line 340: “revealed that a memory of the initial camouflage remains upon withdrawal of the threat.” This was the most exciting line of the manuscript for me. By suggesting that once a visual inspection finds a match and expresses it, it can then be reproduced later, but also doesn’t this also suggest that potentially all the chromatophore pattern fluctuations while they are in the matching phase (decision making most likely, rather than an actual search) are unimportant to the animal as they can be skipped over.

Our earlier use of the word ‘memory’ was inappropriate and detracted from an important conclusion of these experiments, namely that traces of the camouflage pattern are always present during blanching, indicating the superposition of patterns. We have taken out the word memory and clarified our mechanistic inferences to read (discussion):

“Blanching represents the shrinking of chromatophores, caused by the relaxation of the chromatophore muscles. By contrast, the return to a camouflage pattern requires the differential expansion of chromatophores, by contraction of those same muscles. Blanching could therefore be generated by a transient and general inhibition of the chromatophore motor drive, downstream of the camouflage control level; but because recovery from blanching reveals components with different dynamics, this putative inhibition probably acts upstream of the motoneurons (at an intermediate level of chromatophore control) rather than directly on them. In conclusion, camouflage control in Sepia appears to be extremely flexible and non stereotypical when analysed at cellular resolution. The dynamics of its

output suggest the use of feedback to converge onto a chosen camouflage. These results will guide mechanistic studies of this remarkable texture-matching system.”

F. Suggested improvements: experiments, data for possible revision

See above.

19. G. References: appropriate credit to previous work?

The method is new, so no previous comparisons. The older work seems to be referenced well.

Thank you. We have attempted to better place our results in the context of previous work.

H. Clarity and context: lucidity of abstract/summary, appropriateness of abstract, introduction and conclusions

See above.

References:

1. ___ Gonzalez-Bellido, P. T., Scaros, A. T., Hanlon, R. T. & Wardill, T. J. Neural Control of Dynamic 3-Dimensional Skin Papillae for Cuttlefish Camouflage. *iScience* **1**, 24–34 (2018).
2. ___ Altan, E., Solla, S. A., Miller, L. E. & Perreault, E. J. Estimating the dimensionality of the manifold underlying multi-electrode neural recordings. *PLoS Comput. Biol.* **17**, e1008591 (2021).
3. ___ Meyer, D. E., Abrams, R. A., Kornblum, S., Wright, C. E. & Keith Smith, J. E. Optimality in human motor performance: Ideal control of rapid aimed movements. *Psychological Review* vol. 95 340–370 (1988).
4. ___ Spijkers, W. A. & Lochner, P. Partial visual feedback and spatial end-point accuracy of discrete aiming movements. *J. Mot. Behav.* **26**, 283–295 (1994).
5. ___ Bollu, T. *et al.* Cortex-dependent corrections as the tongue reaches for and misses targets. *Nature* **594**, 82–87 (2021).
6. ___ Wallis, T. S. A. *et al.* A parametric texture model based on deep convolutional features closely matches texture appearance for humans. *J. Vis.* **17**, 5 (2017).
7. ___ Reiter, S. & Laurent, G. Visual perception and cuttlefish camouflage. *Curr. Opin. Neurobiol.* **60**, 47–54 (2020).
8. ___ Kelman, E. J., Osorio, D. & Baddeley, R. J. A review of cuttlefish camouflage and object recognition and evidence for depth perception. *J. Exp. Biol.* **211**, 1757–1763 (2008).
9. ___ Victor, J. D., Conte, M. M. & Chubb, C. F. Textures as Probes of Visual Processing. *Annu Rev Vis Sci* **3**, 275–296 (2017).
10. ___ Messenger, J. B. Cephalopod chromatophores: neurobiology and natural history. *Biol. Rev. Camb. Philos. Soc.* **76**, 473–528 (2001).
11. ___ Dubas, F., Hanlon, R. T., Ferguson, G. P. & Pinsker, H. M. Localization and stimulation of chromatophore motoneurons in the brain of the squid, *Lolliguncula brevis*. *J. Exp. Biol.* **121**, 1–25 (1986).
12. ___ Florey, E. & Kriebel, M. E. Electrical and mechanical responses of chromatophore

muscle fibers of the squid, *Loligo opalescens*, to nerve stimulation and drugs. *Z. Vgl. Physiol.* **65**, 98–130 (1969).

Reviewer Reports on the First Revision:

Referees' comments:

Referee #1 (Remarks to the Author):

I appreciate the authors putting in the effort to respond to my comments. I think the new draft is an improvement over the previous version. I found the text to be easier to follow and much more coherent. In particular, the stronger emphasis on the dynamics of the pattern formation in the text – the potential role of the slow spots in updating the trajectory, their frequency/duration as it converges, the re-grouping of chromatophores in different but similar transitions, and how all of this is different to what happens during blanching. It's a fun story and the observations seem to provide interesting constraints for theoretical work on the dynamics of cuttlefish camouflage pattern formation. I am particularly interested in the reorganization of the chromatophores. It does still feel a bit like a list of observations but much less than before.

The arguments and discussion about the apparent high dimensionality of the pattern space seem reasonable but perhaps not bulletproof since 1) dimensionality estimation is challenging/poorly understood in high-dimensional, noisy data sets and 2) there are a lot of processing steps taken in this analysis. The authors mentioned in the rebuttal that this finding goes against previous work suggesting it should be low-dimensional. I think it would be helpful to the reader (and me) to discuss what was different in the previous studies that led them to conclude low dimensionality, i.e. is it the resolution? the variety/number of input patterns? the analysis methods? etc.

I did not find any other major concerns.

I have a few minor comments:

- The error bars in figure 2(b) are not defined
- In the gray part of figure 3(d) the error bars are not defined and the text of the y-axis doesn't match the wording in the caption.
- It could be stated more clearly in the methods that the texture space used to construct the UMAP visualization, the dimensionality estimation and pattern dynamics (the one that starts with PCA to 200PCs) was the 512D representation of the max-pooled fifth layer activations of the VGG-19 neural network.

This is interesting work and I enjoyed learning from it. Thank you!

Referee #2 (Remarks to the Author):

Major concern:

Reading the revised manuscript has highlighted a major concern about the machine-learning analysis performed on the low resolution data, which calls into question the conclusions of Figures 1, 2 and

3. The problem of visual texture is well studied in the fields of computer vision and visual neuroscience, and there are well known parameterizations of textures (e.g. the works of Eero Simoncelli). Rather than use these existing methods, or reference them, the authors use the activations of a late layer of a standard deep-neural network to create a 'texture representation' of the cuttlefish body pattern. This deep-neural network was trained on an image classification task to discriminate objects and to ignore backgrounds, which doesn't seem well suited to use as a representation of texture. They motivate the use of this representation based on "findings from psychophysics experiments" (referring to Wallis et al.) However, Wallis et al. represent texture using the Gram matrix, which creates a set of summary statistics that are spatially invariant, a defining feature of visual texture (which is in line with previous texture synthesis work, Portilla and Simoncelli, 2000). By contrast, in this manuscript, Woo et al. simply used the activations of pool 5 - not the Gram matrix of correlations between feature maps of pool 5. By not computing a summary statistic on the network's features, the representation the authors are using is not spatially invariant, which calls into question its accuracy as a texture representation, and therefore the fidelity of any conclusions reached using this data. To convincingly show that they are using a good representation of texture I would want to see that they could generate perceptually indistinguishable textures from this representation. This is the standard in the field, however, doing so seems outside the scope of this work. Thus, I would advise using a standard texture representation.

Aside from this, It's not clear why the authors didn't use the images of the cuttlefish patterning as the representation of texture. Why was it necessary to transform the patterning itself? If there is a convincing reason to not use the image representation, I suggest the authors use the Portilla and Simoncelli model to parameterize the textures. The Gram matrices of VGG would highly over-parameterize the space.

Furthermore, the dimensions of the layers of deep neural networks vary from layer to layer and can be either larger or smaller than the dimensionality of the original images. In practice, there is a bias towards higher dimensional representations in later layers (this can be seen by looking at the cumulative eigenspectrum across different layers of a deep neural network). The fact that the authors find that cuttlefish camouflage is high dimensional based on the fact that the deep neural network representation is high dimensional is confounded by the fact that later layers in a DNN are intrinsically high dimensional. Thus, this result could just be a result of how they parameterized the texture space.

Finally, the authors' texture representation makes interpreting the result of smooth versus unsmooth trajectories in this texture space difficult. The roles of upper layers of a deep neural network are to make the representations of objects separable, and therefore it is likely to attempt to shatter the space into regions. This is not well suited for asking whether trajectories are smooth or not because these transitions are going to be between these regions and likely not smooth. Thus, the conclusion that camouflage pattern trajectories are torturous and intermittent may again be the result of how the author's chose to parameterize the texture space.

Other feedback:

The authors have made some improvements to the description and motivation of experiments, but

further improvements are needed. For instance, Line 85, the authors state “We first describe camouflage-pattern space”. The authors should describe the goal and experiment first, e.g. “To quantitatively assess cuttlefish pattern space, we presented a series of natural images to a cuttlefish using printed fabric, and filmed the cuttlefish skin at both high and low resolution (Figure 1A)” - Figure 1A should be a diagram of the experimental setup, combined with 1B. It doesn’t make sense to show a diagram of the dynamics when Figure 1 is not about dynamics.

Similarly, the authors should begin the section at line 150 with an introduction to the goals and methods. I also suggest highlighting the transition to using the high resolution data. E.g. “To identify pattern components on the cuttlefish skin, we used the high resolution imaging data to track covarying chromatophores (Figure 1A)”

Some figures have been improved (although, see Major Concern, which pertains to all conclusions from the low resolution analysis, for instance, the trajectories in Figure 3A could be an artifact of the analysis). However, axes must always be labeled. For instance, 2A is a quantitative plot without numbers. Figure 5A should provide a quantification of how similar the starting and ending patterns are for all animals tested. The Figure 2 legend should describe what the colors are.

Use of “texture” - the authors use visual texture in the abstract, but immediately shift to just “texture” in the main text. I think a definition is required. For instance, they say “The generation of skin textures relies on a motor system that controls...” - the generation of 3D texture also relies on a motor system, so this will confuse some readers.

The authors use the blanching experiments as an example of a separate control system. Blanching could be the result of uniform inhibition of the entire skin pattern, in which case it would not be surprising that there is a fast and direct change, returning to the previous state. A more informative test of a different control system would be to quantify the skin change for an innate behavior (like the zebra skin pattern).

Line 34: “This suggests that motion within camouflage space should be stereotyped (being limited to few possible outputs) and direct (from lacking the possibility of correction).”

That may or may not be the case. But in these experiments, the cuttlefish is in a dynamic environment, moving freely, so each time it is exposed to one of the fabric textures, the cuttlefish is in a different location and therefore will see a different visual scene. Furthermore, since cuttlefish skin is so dynamic, the starting point for each transition will be different, so it’s not surprising that no two trajectories are the same.

Line 65: Chromatophores are technically organs not cells

Line 91: "We then used machine learning to parameterize skin-patterns".

The texture representation is critical for all subsequent analyses of the LR images and thus, should be described in brief in the main text. Machine learning is too broad of a description.

Line 98: "Having tested the linearity of texture space we opted for a linear method"

It is not clear from the main text how the linearity of texture space was tested. Is the fact that PCA was compared to JAE what is being referred to here? If so, that doesn't follow from the discussion in the main text.

Line 433: "Therefore, we chose PA, a linear method, to estimate the dimensionality of visual texture space."

I don't see how this logically follows from the analysis conducted. My understanding is that the authors determined the dimensionality using PA and then computed how much variance was explained using this number of latent dimensions for a linear dimensionality reduction method (PCA) and non-linear dimensionality reduction method (JAE). This analysis does not justify the use of PA. It seems from the methods that PA was being used independently of whether they used a linear or non-linear method to determine the latent dimensions. Do the authors mean they chose to use PCA, a linear method rather than a joint autoencoder?

Line 107: "Because natural visual textures themselves are high-dimensional and thus difficult to parameterize simply."

This statement is in conflict with the visual texture literature. What makes visual texture interesting is that, unlike other natural images, they can be parameterized simply with a low-dimensional representation. A recent work that highlights this is Parthasarathy & Simoncelli, 2020.

Line 125: "None (spatial frequency included) matched the predictive power of a high-dimensional texture parameterization"

This is likely because of the methods chosen by the authors. Again, I suggest reading the visual texture literature (especially Eero Simoncelli).

Line 153-154: "Note that cluster numbers and sizes depend on arbitrary decisions about resolution."

If this is in fact an issue with the clustering algorithm the authors should discuss how they chose the cluster numbers and sizes or explain how their arbitrary decisions don't change the inferences taken from the analysis.

Lines 164-168: "Our pattern decomposition had high explanatory power only if the components had been derived from ..."

The author's interpretation of a lack of generalization is very optimistic. It feels more likely that the lack of generalization could mean that they are not finding an appropriate trajectory space with their methods or there is noise in the data. Ideally, they would define one set of components from the data and then look at all trajectories within this space – which is not currently my understanding of how they went about the analysis.

References:

Gatys, L., Ecker, A. S., & Bethge, M. (2015). Texture Synthesis Using Convolutional Neural Networks. *Advances in Neural Information Processing Systems*, 28.

<https://papers.nips.cc/paper/2015/hash/a5e00132373a7031000fd987a3c9f87b-Abstract.html>

Geirhos, R., Rubisch, P., Michaelis, C., Bethge, M., Wichmann, F. A., & Brendel, W. (2018, September 27). ImageNet-trained CNNs are biased towards texture; increasing shape bias improves accuracy and robustness. *International Conference on Learning Representations*.

<https://openreview.net/forum?id=Bygh9j09KX>

Parthasarathy, N., & Simoncelli, E. P. (2020). Self-Supervised Learning of a Biologically-Inspired Visual Texture Model (arXiv:2006.16976). *arXiv*. <http://arxiv.org/abs/2006.16976>

Portilla, J., & Simoncelli, E. P. (2000). A Parametric Texture Model Based on Joint Statistics of Complex Wavelet Coefficients. *International Journal of Computer Vision*, 40(1), 49–70.

<https://doi.org/10.1023/A:1026553619983>

Wallis, T. S. A., Funke, C. M., Ecker, A. S., Gatys, L. A., Wichmann, F. A., & Bethge, M. (2017). A parametric texture model based on deep convolutional features closely matches texture appearance for humans. *Journal of Vision*, 17(12), 5. <https://doi.org/10.1167/17.12.5>

Referee #3 (Remarks to the Author):

A. Summary of the key results

Woo et al. present an updated but still dazzling array of data and figures to convince the reader that chromatophore control in cuttlefish in response to visual stimuli is extremely complex (not just three patterns). They were able to complete this study due to a previous publication demonstrating the method of tracking of 10,000s of chromatophores simultaneously. This new manuscript version is much better in terms of how things were done, justification they provided and improved clarity in figures.

They present several findings as new from their study including (1) chromatophore activations are expressed in a vast array of forms, (2) chromatophore activations are not stereotypical (though they contradict this later), (3) chromatophore activations changed over time and the speed of changes altered over time, (4) analyses showed subcomponents of camouflage activation regions, and (5) different extracted chromatophore activations resulted when different stimuli were used, such as natural, vs checkerboard, vs looming, which they hypothesized was due to different control systems.

B. Originality and significance: if not novel, please include reference

- The method of tracking chromatophores is not novel and was published in *Nature* 2018

(<https://doi.org/10.1038/s41586-018-0591-3>).

- While the analysis methods in this manuscript are novel (sorry this was not clear previously), the relationship between visual stimuli driving expression of chromatophore activation patterns is not

novel, as evidenced by many papers from the labs of Roger Hanlon and Daniel Osorio (See the massive number of papers cited in Hanlon & Messenger 2018. *Cephalopod Behaviour*. Cambridge University Press) and particularly the new paper from Osorio et al. 2022 *Current Biology*.

- The abstract and throughout talks about feedback, which I assume the authors to mean visual feedback as a closed loop visuomotor processing step that alters chromatophore expression continuously. Nearly all publications have an adaptation phase when cuttlefish are either shown, or put in a tank with, a specific background. So clearly it is closed loop rather than a reflex response that is open-loop. On a bigger scale, clearly without functional eyes the cuttlefish cannot produce appropriate skin patterns, which has been shown on nearly every publication. Refs 5, 12 & 13 do not disable the visual system during pattern expression nor does these studies deactivate vision reversibly to demonstrate a lack of feedback is required for proper chromatophore activation patterns. So there has been no way to demonstrate precisely that closed loop exists. Certainly not enough is known about the brain and vision circuitry to implicate feed-forward and feedback neural connections related to specific chromatophore pattern expression on the skin and this study does not look at the brain connectivity. So it is not novel that incoming visual information (which could be termed visual feedback) alters the state of the expanded pattern of activated chromatophores. Given the diversity in the results between animals and stimuli (and they are not clear) it remains unclear when visual feedback determines the final expressed chromatophore pattern. This is especially true for cephalopods because in addition to a decision taking time to select a pattern, there are also skin responses to visual stimuli they experience, both static and from depth cues as they or other objects move in their environment. Fig 4 attempts to get at this, by looking at dynamic changes, but the stimulus is not reported. Reported results are not clear for example the PC1 & PC2 may have high variance indicating the squiggly lines are not important, as they overlap and do not show interrelationships.

- The novelty of this study utilizes a major increase in the spatial and temporal resolution in their tracking method to precisely quantify and show the relationships between visual stimuli and correlated chromatophore activity at the level of single chromatophores (or small patches of chromatophores). However they do not show specific full body patterns except for Fig 1, which uses a different method of clustering than the rest of the paper (as they have stated in their reply letter and methods). Additionally, the patterns shown are not quantified at the cluster level to determine how many clusters are meaningful and what the variances within each cluster are. Also it seems inappropriate to use this method in Fig 1 and then use another method for the rest of the paper but not show the clustering or diversity of skin patterns, especially as the paper is about patterns (and not textures).

- The significance is they have demonstrated that chromatophore control systems in cuttlefish are very sophisticated (more than previously published). Their analysis seems to suggest, but they have not shown, they are producing highly tailored camouflage patterning across the dorsal surface with respect to specific visual stimuli.

- The new paper from Osorio et al. 2022 *Current Biology* should be cited and discussed in the intro and discussion as several aspects of this publication shed new light on the control system and show similar /related findings to this manuscript.

C. Data & methodology: validity of approach, quality of data, quality of presentation

The tracking method is accurate from what was reported in *Nature* and this manuscript. It is producing some very detailed information about the control of cuttlefish skin patterns. The paper remains very complicated and hard to parse. I really appreciate the very high detail quantification

approach but still raises many questions. Unfortunately I still see there are some major problems in the presentation of the research in this manuscript and potentially how it has been analyzed.

- It is unclear why the paper uses texture. The dictionary defines texture as “the way a surface, substance, or piece of cloth feels when you touch it, for example how rough, smooth, hard, or soft it is”. Given there is not testing of the stimuli or skin texture it does not seem justified in this manuscript. Pattern is far more appropriate throughout. But the authors need to clarify that pattern is referred generally to a correlated set of chromatophore activations because in most cases the ‘spatial pattern’ is not shown or provided to the reader in relation to specific stimuli. Texture is also inappropriate as cuttlefish have papillae that alter the 3D texture of the skin but papillae texture was excluded from the analysis. Some stimuli also had shading that would induce perception of depth and texture but this was not analyzed as part of the manuscript from what I understand making “texture” and “visual texture” not relevant to this manuscript as no 3D analysis was done of the skin or stimuli.
- The manuscript discusses patterns, but most of these instances of the word pattern are not referring to specific arrangements or configurations of expressed chromatophore patterns, but rather they are implied patterns that were extracted: either from the UMAP or other analyses. The authors provided some chromatophore activation dynamics information, which is excellent and very helpful but does not address spatial pattern ground truth and their variances (such as the 8 clusters shown in Fig 1d but no analysis of variance among the nine images shown for each). Currently they are just extracted correlations and it is unclear which are similar patterns and statistically how similar they are spatially, which is part of the pattern definition. Chromatophore pattern space makes sense, but texture space should be renamed as stimuli pattern space.
- Chromatophore spatial patterns are not obviously compared within and between trials and between animals. How can the reader determine if specific spatial patterns are in response to specific visual stimuli, given the paper is about pattern? Clearly cuttlefish express more coarse skin patterns from more coarse stimuli (this was already known from previous checkerboard stimuli papers, e.g. Barbosa et al. 2008 Vis Res.). Clearly this manuscript shows is a continuum rather than three patterns, but how precise do individuals match stimuli coarseness to skin patterns (i.e. the error of matching)? And how, spatially, are these specific chromatophore activation patterns reproducible within and between animals to a given stimuli?
- Non-Stereotypical responses query: Line 43 “...space of camouflage patterns is high-dimensional and that the process of pattern matching (search) is not stereotyped.” This was also confirmed in the rebuttal letter where repeated stimuli presentations yielded different response patterns of chromatophore activations. And repeated in the discussion. Yet Fig 2a-b indicates that as granularity of the stimuli increases (both natural and checkerboard patterns) the expressed skin patterns also become more granular or coarse. Then Fig 2c indicates complete pattern difference lacking stereotypy (but the stimulus was not given). Then Fig 3a-b start with two different patterns and end with the same approximate pattern, indicating stereotypy. Fig 4f has similar start and end patterns despite different trajectories (which we do not have variance shown, so may not differ significantly). There are several such conflicts in the paper that are confusing as the specifics are not clearly discussed (As far as I can tell). It is also hard to resolve as a reader whether responses are lacking stereotypy as the patterns are not analyzed spatially and related to specific stimuli or repeated stimuli presentations. This also raises doubt about correlations, because lack of stereotypy implies correlations between stimuli and chromatophore activations should be low but the authors suggest the 0.6-0.7 should be consider high.

- While the number of animals and stimuli are reported throughout, it remains unclear which visual stimuli were used for each figure. For example, in Fig 1c-d, was this just naturalistic or did it include checkerboard stimuli. Was the image provided for each of the eight representative clusters the same or did it differ greatly? It is clear that visual stimuli produced a vast array of skin responses but it is much harder to deduce which specific stimuli produce which specific patterns and how reproducible they were. If the 8 clusters were from one specific image then this should be stated in the figure legend and linked back to the EDF 2 panel (which should also be numbered).
- Line 243: The number of pauses and their duration increased as convergence neared. This does not appear supported by presented data from one trial in Fig 3a. How many trajectories were tracked? How many animals? What is the data for number and pause duration, frequency in seconds +- SD prior settling on specific spatial pattern?

There are some other minor suggested changes.

- Line 43: High-dimensional is vague. Do you mean more than 3, 10 or do you mean 100s? It would be more clear to cite your data from EDF3c of ~ 50-dimensional? Are these 50 dimensions due to the vast number of chromatophore activation combinations? It is unclear what is the biological basis for so many dimensions. There are many skin components or groupings of frequently co expressed activations or deactivations and so this may lead to many dimensions as per Osorio et al. 2022.
- Line 89: Do you mean HR rather than LR?
- Line 91: parameterize skin-patterns – do you mean chromatophore co-activations were determined? I am not seeing data for spatial patterns, just abstract mathematical representations. By patterns are you referring to clusters of skin activations that show high correlation to each other? It should be defined at this early part what you mean by parameterize skin-patterns as spatial configurations are not evident.
- Line 141: direction of motion - two motion-direction models were outlined in Fig 3c but not discussed here. Please put in methods and clarify what are the two motion-direction models and why they were chosen and link to the text here.
- Line 146: number of successive low-velocity regions increased as the animal approached its target. Does this suggest the animal just slowed down to choose a specific place to rest? Why does it have to do with successive error correction steps? Or anything to do with camouflage. Was pattern matching spatially tested statistically to better match the target stimuli?

D. Appropriate use of statistics and treatment of uncertainties

There is a lot of data analysis in this paper. The new figures are helpful. But without a better understanding of the variance between evoked correlations of chromatophore activated spatial patterns and their link to the visual stimuli patterns presented it is very hard to comment on the statistics and uncertainties in this paper as it relates to the control of specific spatial patterns of chromatophore activation. As it stands, there is a good correlation that as visual stimulus granularity increases the skin patterns get significantly more coarse or blocky, however it does not show how reproducible spatial patterns are by each animal and between animals (but this analysis could be done at least on fig 2b data, the legend info is not clear what it is testing). With only three animals there appears some animal specific differences for some checker stimuli. Fig 2c demonstrates the same trend but highlights the extreme diversity in pattern activation within a single animal to simple checkerboard stimuli. So the take home is there are not three patterns but a continuum of activation as granularity of the stimulus increases but also skin patterning is unpredictable in many cases

(which makes challenges for interpreting the correlations found, which would be high when there data sets are very large). Fig 2 highlights it is hard to review whether analyses demonstrate the biology of skin control of pattern in relation to stimuli as the paper outlines its new findings.

E. Conclusions: robustness, validity, reliability

See above, and it still remains, without the stimuli linked to each figure, and the data analysis performed that highlights the relationship between stimuli and the expressed specific spatial chromatophore patterns (how they differ spatially as a pattern between trials, stimuli and individuals) it is hard to understand if the conclusions are warranted.

F. Suggested improvements: experiments, data for possible revision

See above.

G. References: appropriate credit to previous work?

The method is new, so no previous comparisons. The older work seems to be referenced well.

H. Clarity and context: lucidity of abstract/summary, appropriateness of abstract, introduction and conclusions

See above.

Author Rebuttals to First Revision:

Responses to the reviewers' comments on Woo, Liang et al., manuscript.

We thank the reviewers for their detailed comments, which we address below.

Referee #1 (Remarks to the Author):

I appreciate the authors putting in the effort to respond to my comments. I think the new draft is an improvement over the previous version. I found the text to be easier to follow and much more coherent. In particular, the stronger emphasis on the dynamics of the pattern formation in the text – the potential role of the slow spots in updating the trajectory, their frequency/duration as it converges, the re-grouping of chromatophores in different but similar transitions, and how all of this is different to what happens during blanching. It's a fun story and the observations seem to provide interesting constraints for theoretical work on the dynamics of cuttlefish camouflage pattern formation. I am particularly interested in the reorganisation of the chromatophores. It does still feel a bit like a list of observations but much less than before.

Thank you very much.

The arguments and discussion about the apparent high dimensionality of the pattern space seem reasonable but perhaps not bulletproof since 1) dimensionality estimation is challenging/poorly understood in high-dimensional, noisy data sets and 2) there are a lot of processing steps taken in this analysis.

We now present additional analysis demonstrating that the high dimensionality of camouflage pattern space is not due to noisy data. We observed significantly lower dimensionality when applying the same analysis methods to visual threat response data (Extended Data Fig. 9f and below). This difference in dimensionality between datasets was further confirmed using chromatophore-resolution analysis (see below and Extended Data Fig. 9c).

To further address the concern that the high dimensionality of camouflage pattern space might be an artefact of our neural network parameterization, we tested an alternative visual texture metric. We observed similar high dimensionality when using a non-neural network model of visual textures, the Portilla and Simoncelli model (see below and Extended Data Fig. 3e).

The authors mentioned in the rebuttal that this finding goes against previous work suggesting it should be low-dimensional. I think it would be helpful to the reader (and me) to discuss what was different in the previous studies that led them to conclude low dimensionality, i.e. is it the resolution? the variety/number of input patterns? the analysis methods? etc.

The multiple possible reasons behind the finding of a high-dimensional pattern space include our increased resolution, our use of large amounts of imaging data, and our quantitative analysis methods. One clear reason is the size of our stimulus set, which is greater than in most past studies. As we mention in the text:

Using checkerboards as backgrounds²⁷ (Extended Data Fig. 2c), we observed, as others had before^{8,11,27}, that a coarse sampling of spatial frequencies (half-periods: 0.04-20cm) led to only a few clusters of correlated skin patterns. Observing that this sampling of spatial-frequencies was too sparse, we added 16 checkerboard sizes in an intermediate range (Fig. 2b).

I did not find any other major concerns.

I have a few minor comments:

- The error bars in figure 2(b) are not defined

We now indicate that they represent 95% confidence intervals.

- In the gray part of figure 3(d) the error bars are not defined and the text of the y-axis doesn't match the wording in the caption.

We have now clarified this plot and legend.

- It could be stated more clearly in the methods that the texture space used to construct the UMAP visualization, the dimensionality estimation and pattern dynamics (the one that starts with PCA to 200PCs) was the 512D representation of the max-pooled fifth layer activations of the VGG-19 neural network.

We agree, and now state in methods:

"This 512-dimensional texture representation was used to construct the UMAP visualisation, estimate the dimensionality of camouflage pattern space, and study camouflage pattern dynamics."

This is interesting work and I enjoyed learning from it. Thank you!

Thank you for your constructive comments throughout.

Referee #2 (Remarks to the Author):

Major concern:

Reading the revised manuscript has highlighted a major concern about the machine-learning analysis performed on the low resolution data, which calls into question the conclusions of Figures 1, 2 and 3. The problem of visual texture is well studied in the fields of computer vision and visual neuroscience, and there are well known parameterizations of textures (e.g. the works of Eero Simoncelli). Rather than use these existing methods, or reference them, the authors use the activations of a late layer of a standard deep-neural network to create a 'texture representation' of the cuttlefish body pattern. This deep-neural network was trained on an image classification task to discriminate objects and to ignore backgrounds, which doesn't seem well suited to use as a representation of texture. They motivate the use of this representation based on "findings from psychophysics experiments" (referring to Wallis et al.) However, Wallis et al. represent texture using the Gram matrix, which creates a set of summary statistics that are spatially invariant, a defining feature of visual texture (which is in line with previous texture synthesis work, Portilla and Simoncelli, 2000). By contrast, in this manuscript, Woo et al. simply used the activations of pool 5 - not the Gram matrix of correlations between feature maps of pool 5. By not computing a summary statistic on the network's features, the representation the authors are using is not spatially

invariant, which calls into question its accuracy as a texture representation, and therefore the fidelity of any conclusions reached using this data. To convincingly show that they are using a good representation of texture I would want to see that they could generate perceptually indistinguishable textures from this representation. This is the standard in the field, however, doing so seems outside the scope of this work. Thus, I would advise using a standard texture representation.

We apologise for our citations oversight, which we have now corrected.

We think that the reviewer is mistaken, however, when stating that our metrics differ from previous methods in not being spatially invariant. Our metrics use a global max pooling of VGG layer 5, making the representation spatially invariant. Very similar metrics have been used in the recent visual texture literature. For example, Parthasarathy & Simoncelli (2020), which the reviewer points us to, uses precisely a global pooling of VGG layer 5 as a texture metric strongly competitive to the 'V2Net' they introduce. Cimpoi et al. (2015) study how the representation of images within single deep layers of VGG16/19, after a similar spatial pooling, provides a powerful representation of texture for classifying the describable texture database they introduce. We now cite these studies. Therefore, our metric is, we believe, a standard texture representation, well studied in the literature, and spatially invariant by construction.

We agree with the reviewer that a comparison of visual texture metrics would be useful. We therefore performed additional analysis comparing our texture metric with the Gram matrix based approach of Gatys et al. 2015, as well as the Portilla and Simoncelli model. We show that, for our data, there is a correlation of 0.85 between pattern distances calculated using Gram matrices and calculated using our metric (shown in a below and in Extended Data Fig. 3a). Compared to our metric, the Portilla and Simoncelli model produces similarly high estimates for the dimensionality of camouflage pattern space (shown in e below and in Extended Data Fig. 3e). It was able to predict camouflage patterns from background images, albeit significantly worse than our own model (now shown in Extended Data Fig. 4d,e, and below next reply). Thus, our results are not an artefact of our choice of texture representation; similar results can be achieved using other methods. We retained our method because of reduced computational cost (compared to that of Gatys et al. 2015), and improved prediction of camouflage (compared to the Portilla and Simoncelli model).

Aside from this, It's not clear why the authors didn't use the images of the cuttlefish patterning as the representation of texture. Why was it necessary to transform the patterning itself? If there is a convincing reason to not use the image representation, I suggest the authors use the Portilla and Simoncelli model to parameterize the textures. The Gram matrices of VGG would highly over-parameterize the space.

We did consider taking this strategy at the outset of our study. We decided against this for two reasons. The first is that this representation is not translation invariant, agreeing with the reviewer's above point that this is a defining feature of visual texture that we didn't want to give up. The second is that this metric is extremely sensitive to small image misalignments, difficult to avoid in working with such highly deformable animals. We now provide a comparison ('images' in Extended Data Figure 4d,e and d,e below) showing that the untransformed images themselves are a poor predictor of skin pattern responses. We moved to the method of Gatys et al. (Gram matrices of VGG) in part to become invariant to these misalignments. As the reviewer correctly states, the number of parameters in this model is cumbersome (far more than pixels in our images). To reduce this problem, we moved to the max pooling of layer 5 approach.

Furthermore, the dimensions of the layers of deep neural networks vary from layer to layer and can be either larger or smaller than the dimensionality of the original images. In practice, there is a bias towards higher dimensional representations in later layers (this can be seen by looking at the cumulative eigenspectrum across different layers of a deep neural network). The fact that the authors find that cuttlefish camouflage is high dimensional based on the fact that the deep neural network representation is high dimensional is confounded by the fact that later layers in a DNN are intrinsically high dimensional. Thus, this result could just be a result of how they parameterized the texture space.

We now present additional analysis demonstrating that the high dimensionality of camouflage pattern space is not an artefact of our neural network parameterization. First, we observed similar high dimensionality when using a non-neural network model of visual textures, the Portilla and Simoncelli model suggested by the reviewer (Extended Data Fig. 3e and e below). Second, we observed significantly lower dimensionality when applying our network to visual threat response data (Extended Data Fig. 9f and f below). This difference in dimensionality between datasets was further confirmed using chromatophore-resolution analysis (Extended Data Fig. 9c and c below).

Finally, the authors' texture representation makes interpreting the result of smooth versus unsmooth trajectories in this texture space difficult. The roles of upper layers of a deep neural network are to make the representations of objects separable, and therefore it is

likely to attempt to shatter the space into regions. This is not well suited for asking whether trajectories are smooth or not because these transitions are going to be between these regions and likely not smooth. Thus, the conclusion that camouflage pattern trajectories are tortuous and intermittent may again be the result of how the author's chose to parameterize the texture space.

We now present additional analysis demonstrating that our description of camouflage pattern trajectories truly reflect biological reality, and are not merely a result of our choice of visual texture representation. Using our representation, we calculated the curvature of trajectories during camouflage and during blanching behaviour. We observed a significantly lower curvature during blanching than during camouflage (Extended Data Fig. 9g and g below, in 50-PC space). This difference is consistent with the decreased curvature we observe when comparing the same datasets at chromatophore resolution (Fig. 5c). Thus, camouflage trajectories are tortuous and blanching trajectories smooth, both at low resolution (described using our visual texture representation) and at the resolution of single chromatophores (an analysis independent of the VGG neural network).

Other feedback:

The authors have made some improvements to the description and motivation of experiments, but further improvements are needed. For instance, Line 85, the authors state “We first describe camouflage-pattern space”. The authors should describe the goal and experiment first, e.g. “To quantitatively assess cuttlefish pattern space, we presented a series of natural images to a cuttlefish using printed fabric, and filmed the cuttlefish skin at both high and low resolution (Figure 1A)” - Figure 1A should be a diagram of the experimental setup, combined with 1B. It doesn't make sense to show a diagram of the dynamics when Figure 1 is not about dynamics.

Thank you for the nice suggestion, which we follow, replacing our sentence with

“To quantitatively assess camouflage pattern space, we presented a series of natural images to a cuttlefish using printed fabric, and filmed the cuttlefish skin at both high and low resolution.”

We prefer to keep Fig. 1a as a schematic of pattern motion, which we discuss in our introduction and is our central theme. We show the different components of our experimental setup in Extended Data Fig. 1a-c.

Similarly, the authors should begin the section at line 150 with an introduction to the goals and methods. I also suggest highlighting the transition to using the high resolution data. E.g. “To identify pattern components on the cuttlefish skin, we used the high resolution imaging data to track covarying chromatophores (Figure 1A)”

As suggested, we reordered our text in this paragraph to begin with goals and methods.

“We next used high resolution imaging to identify large pattern components that might reflect the higher levels of a hypothesised control hierarchy in the chromatophore system.”

Some figures have been improved (although, see Major Concern, which pertains to all conclusions from the low resolution analysis, for instance, the trajectories in Figure 3A could be an artifact of the analysis). However, axes must always be labeled. For instance, 2A is a quantitative plot without numbers. Figure 5A should provide a quantification of how similar the starting and ending patterns are for all animals tested. The Figure 2 legend should describe what the colors are.

We address the major concern about our visual texture representation above. We have now added values to the axes of Fig. 2a. Figure 5a is indeed just an illustrative example; we provide quantification of the inter-trial similarities for all animals in the tanglegrams in Figure 5f and Extended Data Figure 8d.

Use of “texture” - the authors use visual texture in the abstract, but immediately shift to just “texture” in the main text. I think a definition is required. For instance, they say “The generation of skin textures relies on a motor system that controls...” - the generation of 3D texture also relies on a motor system, so this will confuse some readers.

We agree that this wording was confusing. We have changed this text to read “The generation of skin visual textures...”. We move to just referring to these as textures later in the manuscript, adding citations of the visual texture literature and referring readers to our methods section, where we write

“‘Texture representation’ in this study refers to 2-D visual textures³⁵⁻³⁷. Cuttlefish can produce different 2-D textures through chromatophore activity, and also alter their 3-D appearance through postural motion and contraction of papillae¹⁸. These 3-D alterations have effects on camouflage and alter the 2-D visual texture of the cuttlefish. These are detected by and incorporated into our LR analysis.”

The authors use the blanching experiments as an example of a separate control system. Blanching could be the result of uniform inhibition of the entire skin pattern, in which case it would not be surprising that there is a fast and direct change, returning to the previous state. A more informative test of a different control system would be to quantify the skin change for an innate behavior (like the zebra skin pattern).

Blanching could indeed have been the result of uniform inhibition of the entire skin pattern. Our analysis in Fig. 5, however, shows this not to be the case. Instead, spatially organised groups of chromatophores contract and later expand in stereotyped, nonuniform, sequences.

Line 34: "This suggests that motion within camouflage space should be stereotyped (being limited to few possible outputs) and direct (from lacking the possibility of correction)."

That may or may not be the case. But in these experiments, the cuttlefish is in a dynamic environment, moving freely, so each time it is exposed to one of the fabric textures, the cuttlefish is in a different location and therefore will see a different visual scene. Furthermore, since cuttlefish skin is so dynamic, the starting point for each transition will be different, so it's not surprising that no two trajectories are the same.

We observe certain camouflage patterns adopted consistently over different backgrounds (Fig. 2). From these conserved starting points, we observe a diversity of paths by which camouflage patterns change (Fig. 3,4). This is not consistent with a hypothesis whereby motion in pattern space is open loop and output patterns are few. We agree, however, that previous models of cuttlefish camouflage, which we are referring to in our manuscript, do not incorporate these behavioural realities into the description of the system.

Line 65: Chromatophores are technically organs not cells

'Chromatophore' can refer to the cell holding the sacculus of pigment granules (chromatophore proper), and to the organ composed of that cell, the radial muscles, and the support cells (Tublitz et al. 2006; Osorio 2014). We use the cellular definition in our manuscript, as this is the part of the organ that we track. We have now added this point in the Chromatophore segmentation section of methods:

"In this study, we refer to the pigmented chromatophore proper as 'chromatophore', and 'chromatophore size' as the size of the pigment cell that we track."

Line 91: "We then used machine learning to parameterize skin-patterns".

The texture representation is critical for all subsequent analyses of the LR images and thus, should be described in brief in the main text. Machine learning is too broad of a description.

We now state in the main text "We then used a pre-trained neural network to parameterize skin-patterns", and in Methods more explicitly state

"The texture representation used in our LR imaging (Fig. 1-3) was the max-pooled fifth layer activations (conv5_1) of the VGG-19 neural network with weights pre-trained with the ImageNet dataset in an object recognition task, accessed through the Keras platform³⁹."

Line 98: "Having tested the linearity of texture space we opted for a linear method"

It is not clear from the main text how the linearity of texture space was tested. Is the fact that PCA was compared to JAE what is being referred to here? If so, that doesn't follow from the discussion in the main text.

We now write "Having tested the explanatory power of linear and nonlinear methods for dimensionality estimation (Methods)".

We have also clarified our procedure for model selection in the Methods section (see below).

Line 433: "Therefore, we chose PA, a linear method, to estimate the dimensionality of visual texture space."

I don't see how this logically follows from the analysis conducted. My understanding is that the authors determined the dimensionality using PA and then computed how much variance was explained using this number of latent dimensions for a linear dimensionality reduction method (PCA) and non-linear dimensionality reduction method (JAE). This analysis does not justify the use of PA. It seems from the methods that PA was being used independently of whether they used a linear or non-linear method to determine the latent dimensions. Do the authors mean they chose to use PCA, a linear method rather than a joint autoencoder?

We are sorry for the confusion. Our method was guided by simulation constrained with neural data (Altan *et al.* 2021). It is true that PA is used in the preliminary step to determine an upper bound of the true dimensionality prior to any knowledge of the degree of nonlinearity of the space. This step was used as a principled way of defining the number of latent dimensions, so as to ensure a fair comparison of explanatory power between a linear model (PCA) and a non-linear model (JAE). The text and figure legend now read:

Then, we fitted a linear (PCA) and a nonlinear (Joint Autoencoder) model, respectively, to the data with the same number of latent dimensions as determined by PA in the previous step.

Skin texture variance explained (mean + 95% confidence interval) by principal component analysis (PCA) and Joint Autoencoder (JAE), at 59.4 ± 1.2 latent dimensions.

The quoted sentence refers to our subsequent choice of using PA (a linear method) to estimate dimensionality based on the result that PCA performed better than JAE.

Line 107: "Because natural visual textures themselves are high-dimensional and thus difficult to parameterize simply."

This statement is in conflict with the visual texture literature. What makes visual texture interesting is that, unlike other natural images, they can be parameterized simply with a low-dimensional representation. A recent work that highlights this is Parthasarathy & Simoncelli, 2020.

We have removed the word 'high-dimensional' to avoid confusing our message with the relative nature of this term. Models such as the 60-dimensional texture description of

Parthasarathy & Simoncelli, 2020 were interpreted by us as being high-dimensional, relative to the 1st and 2nd order statistics often used to quantify cephalopod camouflage.

Line 125: “None (spatial frequency included) matched the predictive power of a high-dimensional texture parameterization”

This is likely because of the methods chosen by the authors. Again, I suggest reading the visual texture literature (especially Eero Simoncelli).

The published works on visual textures of B Julesz, J Victor, E Simoncelli, M Bethge and others have been and continue to be very influential for our work; we now provide additional citations. We now compare different texture parameterization methods and show that the predictive power of our method is stronger than the suggested Portilla and Simoncelli (2000) model (see below “Text(PS)”, Extended Data Figure 4d)

Line 153-154: “Note that cluster numbers and sizes depend on arbitrary decisions about resolution.”

If this is in fact an issue with the clustering algorithm the authors should discuss how they chose the cluster numbers and sizes or explain how their arbitrary decisions don’t change the inferences taken from the analysis.

We agree and indeed did so: In ED Fig. 6e-h and ED Fig. 9a-c, we have explored the parameter space of our clustering algorithm (number and size), showing that our conclusions are stable across parameter choices (see below).

Lines 164-168: “Our pattern decomposition had high explanatory power only if the components had been derived from ...”

The author’s interpretation of a lack of generalization is very optimistic. It feels more likely that the lack of generalization could mean that they are not finding an appropriate trajectory space with their methods or there is noise in the data. Ideally, they would define one set of components from the data and then look at all trajectories within this space – which is not currently my understanding of how they went about the analysis.

In Fig. 4e, we tried to define such component sets by pooling all trajectories (“all traj.”) and found that they failed to describe most of the variance in each individual camouflage trajectory. However, in Fig. 5k-i, using the same approach, we were able to find component sets (‘cluster all trials’) that were more generalizable when describing the dynamics of blanch pattern change. Thus, the lack of generalisation is not due to limitation of our methods or noise in the data; instead it reflects mechanistic features of camouflage skin patterning.

References:

Cimpoi, M., Maji, S., & Vedaldi, A. (2015). Deep filter banks for texture recognition and segmentation. In Proceedings of the IEEE conference on computer vision and pattern recognition (pp. 3828-3836).

Gatys, L., Ecker, A. S., & Bethge, M. (2015). Texture Synthesis Using Convolutional Neural Networks. *Advances in Neural Information Processing Systems*, 28. <https://papers.nips.cc/paper/2015/hash/a5e00132373a7031000fd987a3c9f87b-Abstract.html>

Geirhos, R., Rubisch, P., Michaelis, C., Bethge, M., Wichmann, F. A., & Brendel, W. (2018, September 27). ImageNet-trained CNNs are biased towards texture; increasing shape bias improves accuracy and robustness. *International Conference on Learning Representations*. <https://openreview.net/forum?id=Bygh9j09KX>

Osorio, D. (2014). Cephalopod behaviour: skin flicks. *Current biology*, 24(15), R684-R685.

Parthasarathy, N., & Simoncelli, E. P. (2020). Self-Supervised Learning of a Biologically-Inspired Visual Texture Model (arXiv:2006.16976). arXiv. <http://arxiv.org/abs/2006.16976>

Portilla, J., & Simoncelli, E. P. (2000). A Parametric Texture Model Based on Joint Statistics of Complex Wavelet Coefficients. *International Journal of Computer Vision*, 40(1), 49–70. <https://doi.org/10.1023/A:1026553619983>

Tublitz, N. J., Gaston, M. R., & Loi, P. K. (2006). Neural regulation of a complex behavior: body patterning in cephalopod molluscs. *Integrative and Comparative Biology*, 46(6), 880-889.

Wallis, T. S. A., Funke, C. M., Ecker, A. S., Gatys, L. A., Wichmann, F. A., & Bethge, M. (2017). A parametric texture model based on deep convolutional features closely matches texture appearance for humans. *Journal of Vision*, 17(12), 5. <https://doi.org/10.1167/17.12.5>

Referee #3 (Remarks to the Author):

A. Summary of the key results

Woo et al. present an updated but still dazzling array of data and figures to convince the reader that chromatophore control in cuttlefish in response to visual stimuli is extremely complex (not just three patterns). They were able to complete this study due to a previous publication demonstrating the method of tracking of 10,000s of chromatophores simultaneously. This new manuscript version is much better in terms of how things were done, justification they provided and improved clarity in figures.

Thank you.

They present several findings as new from their study including (1) chromatophore activations are expressed in a vast array of forms, (2) chromatophore activations are not stereotypical (though they contradict this later), (3) chromatophore activations changed over time and the speed of changes altered over time, (4) analyses showed subcomponents of camouflage activation regions, and (5) different extracted chromatophore activations resulted when different stimuli were used, such as natural, vs checkerboard, vs looming, which they hypothesized was due to different control systems.

B. Originality and significance: if not novel, please include reference

- The method of tracking chromatophores is not novel and was published in Nature 2018 (<https://doi.org/10.1038/s41586-018-0591-3>).
- While the analysis methods in this manuscript are novel (sorry this was not clear previously), the relationship between visual stimuli driving expression of chromatophore activation patterns is not novel, as evidenced by many papers from the labs of Roger Hanlon and Daniel Osorio (See the massive number of papers cited in Hanlon & Messenger 2018. Cephalopod Behaviour. Cambridge University Press) and particularly the new paper from Osorio et al. 2022 Current Biology.
- The abstract and throughout talks about feedback, which I assume the authors to mean visual feedback as a closed loop visuomotor processing step that alters chromatophore expression continuously. Nearly all publications have an adaptation phase when cuttlefish are either shown, or put in a tank with, a specific background. So clearly it is closed loop rather than a reflex response that is open-loop. On a bigger scale, clearly without functional eyes the cuttlefish cannot produce appropriate skin patterns, which has been shown on nearly every publication.

We apologise for the confusion surrounding the definition of 'feedback'. Our data do not point to a particular feedback pathway (visual, other sensory, or corollary discharge). But we note that multiple authors have referred to cephalopod camouflage as an open-loop system or one that does not rely on feedback. (See for example Chapter 3 of Hanlon & Messenger 2018. Cephalopod Behaviour. Cambridge University Press.) We have now added a citation and brief summary of the Osorio et al 2022 study in our introduction and discussion.

Refs 5, 12 & 13 do not disable the visual system during pattern expression nor does these studies deactivate vision reversibly to demonstrate a lack of feedback is required for proper

chromatophore activation patterns. So there has been no way to demonstrate precisely that closed loop exists.

We agree with the reviewer that there has yet to be any evidence demonstrating precisely the existence and nature of a closed loop. Therefore, we believe that our findings in camouflage transition dynamics, suggestive of successive error-correcting steps (Fig. 3), constitute novel evidence that points to the involvement of feedback.

Certainly not enough is known about the brain and vision circuitry to implicate feed-forward and feedback neural connections related to specific chromatophore pattern expression on the skin and this study does not look at the brain connectivity. So it is not novel that incoming visual information (which could be termed visual feedback) alters the state of the expanded pattern of activated chromatophores. Given the diversity in the results between animals and stimuli (and they are not clear) it remains unclear when visual feedback determines the final expressed chromatophore pattern. This is especially true for cephalopods because in addition to a decision taking time to select a pattern, there are also skin responses to visual stimuli they experience, both static and from depth cues as they or other objects move in their environment.

We respectfully disagree with this point. Incoming visual information from static and depth cues during self locomotion might contribute to 'visual feedback' for locomotion, but not for pattern motion, and thus would not fall under 'visual feedback' in our interpretation of the results. While it is known that camouflage patterning is driven by visual input, the likely involvement of feedback during skin-pattern change is novel.

It is correct that a lack of stimulus-response stereotypy could suggest that camouflage is not hard-wired, but here we investigated the lack of stereotypy in motion within pattern space. (They are not mutually exclusive.) Our findings argue against open-loop behaviour.

Fig 4 attempts to get at this, by looking at dynamic changes, but the stimulus is not reported. Reported results are not clear for example the PC1 & PC2 may have high variance indicating the squiggly lines are not important, as they overlap and do not show interrelationships.

The top PCs maximise variance across whole trajectories (Fig. 4f). They generally emphasise salient features such as the presence of the central white square, and give little weight to the instantaneous variance ('squigginess') along individual trajectories. Such intermittency is discussed in Figure 3.

We now report the background stimuli used in Fig. 4a,f. The two trajectories in Fig. 4f overlap in space, but the paths taken through this space are different: the lack of stereotypy in pattern motion indeed suggests that the exact path taken is unimportant to camouflage. But we also show that these differences reflect real differences in chromatophore expansion profiles (Fig. 4g-i). Identical methods revealed much more stereotyped dynamics during visual threat responses (blanching, Fig. 5c, Extended Data Fig. 9), indicating that these observations are not a foregone result of technical noise.

- The novelty of this study utilizes a major increase in the spatial and temporal resolution in their tracking method to precisely quantify and show the relationships between visual stimuli and correlated chromatophore activity at the level of single chromatophores (or small patches of chromatophores). However they do not show specific full body patterns except for Fig 1, which uses a different method of clustering than the rest of the paper (as they have stated in their reply letter and methods). Additionally, the patterns shown are not quantified at the cluster level to determine how many clusters are meaningful and what the variances within each cluster are. Also it seems inappropriate to use this method in Fig 1 and then use another method for the rest of the paper but not show the clustering or diversity of skin patterns, especially as the paper is about patterns (and not textures).

We thank the reviewer for appreciating the novelty of our methods. We now clarify in our methods section that the texture space representation is in fact used throughout Figures 1-3. The only difference in Figure 1 was the usage of a non-linear embedding to visualise this high-dimensional texture space in two dimensions, motivating subsequent analyses. While this 2-D embedding is useful for providing an intuition of the diversity of the pattern repertoire, the dimensionality of which we quantify in Extended Data Figure 3a-c, it is not best-suited for the other analyses in Figures 2-3 and Extended Data Figures 3-5. We would also like to point out that we show full-body patterns along the axes in Figure 2a-b, as well as in examples of the start and end points of the trajectories in Figures 3a-b.

We did not perform clustering in Figure 1. Our findings in Figures 2-3 do not rely on clustering in visual texture space. In fact, our results present a more continuous space, as rightly pointed out by the reviewer under Section C below. (Fig. 2).

- The significance is they have demonstrated that chromatophore control systems in cuttlefish are very sophisticated (more than previously published). Their analysis seems to suggest, but they have not shown, they are producing highly tailored camouflage patterning across the dorsal surface with respect to specific visual stimuli.

In Figure 2 and Extended Data Figure 4, we show camouflage patterning to a variety of natural and artificial background stimuli (Extended Data Fig. 2). We now present additional example images showing how camouflage patterns are indeed highly tailored with respect to specific visual stimuli (Extended Data Fig. 4f,g).

- The new paper from Osorio et al. 2022 Current Biology should be cited and discussed in the intro and discussion as several aspects of this publication shed new light on the control system and show similar /related findings to this manuscript.

We now cite and discuss this interesting new study in the introduction and discussion.

C. Data & methodology: validity of approach, quality of data, quality of presentation

The tracking method is accurate from what was reported in Nature and this manuscript. It is producing some very detailed information about the control of cuttlefish skin patterns. The paper remains very complicated and hard to parse. I really appreciate the very high detail

quantification approach but still raises many questions. Unfortunately I still see there are some major problems in the presentation of the research in this manuscript and potentially how it has been analyzed.

Thank you. We hope that the new analyses and changes made to the manuscript and figures have clarified the results of our study.

- It is unclear why the paper uses texture. The dictionary defines texture as “the way a surface, substance, or piece of cloth feels when you touch it, for example how rough, smooth, hard, or soft it is”. Given there is not testing of the stimuli or skin texture it does not seem justified in this manuscript. Pattern is far more appropriate throughout. But the authors need to clarify that pattern is referred generally to a correlated set of chromatophore activations because in most cases the ‘spatial pattern’ is not shown or provided to the reader in relation to specific stimuli. Texture is also inappropriate as cuttlefish have papillae that alter the 3D texture of the skin but papillae texture was excluded from the analysis. Some stimuli also had shading that would induce perception of depth and texture but this was not analyzed as part of the manuscript from what I understand making “texture” and “visual texture” not relevant to this manuscript as no 3D analysis was done of the skin or stimuli.
- The manuscript discusses patterns, but most of these instances of the word pattern are not referring to specific arrangements or configurations of expressed chromatophore patterns, but rather they are implied patterns that were extracted: either from the UMAP or other analyses. The authors provided some chromatophore activation dynamics information, which is excellent and very helpful but does not address spatial pattern ground truth and their variances (such as the 8 clusters shown in Fig 1d but no analysis of variance among the nine images shown for each). Currently they are just extracted correlations and it is unclear which are similar patterns and statistically how similar they are spatially, which is part of the pattern definition. Chromatophore pattern space makes sense, but texture space should be renamed as stimuli pattern space.

We apologise for the confusion. We use ‘texture’ in the manuscript to refer to 2-d visual textures, terminology that has been used for many years in the computational vision literature to describe the kinds of visual patterns that cuttlefish attempt to camouflage to. We refer to visual textures in our abstract, now refer to ‘visual texture’ additional times in the main text, and have added additional references. We also refer readers to our methods section, where we define our terminology and discuss the issue of papillae:

“‘Texture representation’ in this study refers to 2-D visual textures^{35–37}. Cuttlefish can produce different 2-D textures through chromatophore activity, and also alter their 3-D appearance through postural motion and contraction of papillae¹⁸. These 3-D alterations have effects on camouflage and alter the 2-D visual texture of the cuttlefish. These are detected by and incorporated into our LR analysis.”

We take ‘camouflage patterns’ as objects that can be studied at multiple resolutions. In the introduction, we define the 2 scales used in this study: that of visual texture and chromatophore-representation spaces. We do not extract patterns from UMAP; instead we

use this method only as a tool to visualise high-dimensional texture space. We have now added text to clarify our parameterization of visual texture, writing in methods:

“The texture representation used in our LR imaging (Fig. 1-3) was the max-pooled fifth layer activations (conv5_1) of the VGG-19 neural network with weights pre-trained with the ImageNet dataset in an object recognition task, accessed through the Keras platform³⁹.”

“This 512-dimensional texture representation was used to construct the UMAP visualisation, estimate the dimensionality of camouflage pattern space, and study camouflage pattern dynamics.”

We did not perform clustering in Figure 1. We statistically analyse our texture representation, and relate it to background visual textures in Figures 2-3. These analyses likewise do not rely on clustering in visual texture space. In fact, our results present a more continuous space.

- Chromatophore spatial patterns are not obviously compared within and between trials and between animals. How can the reader determine if specific spatial patterns are in response to specific visual stimuli, given the paper is about pattern? Clearly cuttlefish express more coarse skin patterns from more coarse stimuli (this was already known from previous checkerboard stimuli papers, e.g. Barbosa et al. 2008 Vis Res.). Clearly this manuscript shows is a continuum rather than three patterns, but how precise do individuals match stimuli coarseness to skin patterns (i.e. the error of matching)? And how, spatially, are these specific chromatophore activation patterns reproducible within and between animals to a given stimuli?

To compare skin patterns and visual stimuli quantitatively, it is necessary to use a metric that can be applied to both. Much of the manuscript therefore utilises a high-dimensional and perceptually relevant metric, directly on images of cuttlefish. We show in Extended Data Fig. 4d,e that across animals and visual stimuli this metric allows one to predict visual stimuli from cuttlefish camouflage patterns better than even optimal combinations of previously used metrics. We use this metric to analyse how specific spatial patterns vary with specific visual stimuli across animals (Fig. 2a,b). Using checkerboard stimuli (Fig 2c,d and Extended Data Fig. 4a-c), we extend this analysis to the level of chromatophore activation patterns. We now present example images from these experiments, showing how camouflage patterns vary consistently with background stimuli across animals (Extended Data Fig. 4f,g).

We agree with the reviewer that a downside of the generality of our high-dimensional image metric is that it is difficult to interpret variation in terms of particular visual features of interest (e.g. coarseness). This is a general issue with high-dimensional data analysis, and we believe that the benefits of our approach (enabling us to describe novel aspects of the system) outweigh the costs in this case.

- Non-Stereotypical responses query: Line 43 “...space of camouflage patterns is high-dimensional and that the process of pattern matching (search) is not stereotyped.” This was also confirmed in the rebuttal letter where repeated stimuli presentations yielded different

response patterns of chromatophore activations. And repeated in the discussion. Yet Fig 2a-b indicates that as granularity of the stimuli increases (both natural and checkerboard patterns) the expressed skin patterns also become more granular or course. Then Fig 2c indicates complete pattern difference lacking stereotypy (but the stimulus was not given). Then Fig 3a-b start with two different patterns and end with the same approximate pattern, indicating stereotypy. Fig 4f has similar start and end patterns despite different trajectories (which we do not have variance shown, so may not differ significantly). There are several such conflicts in the paper that are confusing as the specifics are not clearly discussed (As far as I can tell). It is also hard to resolve as a reader whether responses are lacking stereotypy as the patterns are not analyzed spatially and related to specific stimuli or repeated stimuli presentations. This also raises doubt about correlations, because lack of stereotypy implies correlations between stimuli and chromatophore activations should be low but the authors suggest the 0.6-0.7 should be consider high.

We apologise for the confusing presentation of our results, which show a lack of stereotypy in the *dynamics* of camouflage transitions (Fig. 3) and in the reorganisation of chromatophore groupings during camouflage transitions (Fig. 4), while showing reliable resulting camouflage patterns (Fig. 2). We now write, when discussing the reliability of these steady-state camouflage patterns (end of results section: Camouflage-pattern space is high dimensional):

“Sepia camouflage can therefore smoothly and predictably transition from one pattern to another when challenged with appropriate sets of visual textures.”

- While the number of animals and stimuli are reported throughout, it remains unclear which visual stimuli were used for each figure. For example, in Fig 1c-d, was this just naturalistic or did it include checkerboard stimuli. Was the image provided for each of the eight representative clusters the same or did it differ greatly? It is clear that visual stimuli produced a vast array of skin responses but it is much harder to deduce which specific stimuli produce which specific patterns and how reproducible they were. If the 8 clusters were from one specific image then this should be stated in the figure legend and linked back to the EDF 2 panel (which should also be numbered).

This is a great suggestion. In Fig. 1c-d we used both naturalistic and checkerboard stimuli, and now state this in figure legend. We also number the images in Extended Data Fig. 2 and refer to these numbers in the main text. We clarify in the figure legend and methods that the coloured rectangles in Figure 1c are not labels of clusters; instead, they mark boundaries of the 3×3 grid where the examples in Figure 1d originated:

“Representative images taken from the 8 coloured regions of texture space in (c).”

“For visualisation, a 3×3 set of grid points were laid on each of the selected regions in the 2-D UMAP space, the nearest data point with distance ≤ 0.1 was selected, and the corresponding skin pattern plotted (Fig. 1d).”

We agree with the reviewer that the variability apparent in Figure 1c-d warrants quantification. We therefore quantified the dimensionality of the skin pattern visual texture

space in 12 animals in Extended Data Figure 3a-e. The stimulus-response relationship and reproducibility (stereotypy, both in camouflage response and in pattern motion, detailed above), as the reviewer pointed out, are also central to the current study. We investigated these questions in detail in Figures 2, 4 and 5 with at least 3 animals in each case. Accompanying our quantitative analyses, we now provide example images demonstrating the camouflage response to both naturalistic and checkerboard backgrounds (Extended Data Fig. 4f,g and below).

- Line 243: The number of pauses and their duration increased as convergence neared. This does not appear supported by presented data from one trial in Fig 3a. How many trajectories were tracked? How many animals? What is the data for number and pause duration, frequency in seconds \pm SD prior settling on specific spatial pattern?

We refer in text to Fig. 3d (grey histogram) here, which exactly plots frequency in seconds \pm SD as a function of distance to background for $N = 868$ slow points, from 85 trajectories in 3 animals. We state this in the figure legend.

There are some other minor suggested changes.

- Line 43: High-dimensional is vague. Do you mean more than 3, 10 or do you mean 100s? It would more clear to cite your data from EDF3c of ~ 50 -dimensional? Are these 50 dimensions due to the vast number of chromatophore activation combinations? It is unclear what is the biological basis for so many dimensions. There are many skin components or groupings of frequently co expressed activations or deactivations and so this may lead to many dimensions as per Osorio et al. 2022.

We now refer to the high dimensionality of camouflage space after describing our methods for estimating ~ 60 dimensions. We now cite and discuss the recent study of Osorio et al. 2022, which is indeed consistent with our observation of high dimensionality. We explore the mechanistic underpinnings of this high dimensionality using our chromatophore resolution data in Figures 2, 4 and 5. We identify pattern components in a data-driven way

and describe an unexpected flexibility in this system, differing markedly between camouflage and visual threat behaviours.

- Line 89: Do you mean HR rather than LR?

We confirm that we used 200,000 LR images to build the skin pattern space in Figure 1. In the current study, LR has a resolution of about 119.8 $\mu\text{m}/\text{pixel}$, 1 pixel containing 2.4 chromatophores on average, whereas HR (Fig. 2c, 4, 5) has a resolution of 17.4 $\mu\text{m}/\text{pixel}$, 1 chromatophore occupying 54 pixels on average. (Methods)

- Line 91: parameterize skin-patterns – do you mean chromatophore co-activations were determined? I am not seeing data for spatial patterns, just abstract mathematical representations. By patterns are you referring to clusters of skin activations that show high correlation to each other? It should be defined at this early part what you mean by parameterize skin-patterns as spatial configurations are not evident.

We apologise for the confusion. Here we refer to the generation of the texture representation, in which we represent the LR images containing spatial patterns in a set of perceptually-relevant parameters (methods).

- Line 141: direction of motion - two motion-direction models were outlined in Fig 3c but not discussed here. Please put in methods and clarify what are the two motion-direction models and why they were chosen and link to the text here.

We now describe the two motion-direction models in the figure legend and methods section, linking to main text.

Legend:

“c. Test of two motion-direction models (“update” and “memory”) for motion in texture space: dark green vector points to end-goal from starting point; light green vectors point to end-goal from each intermediate slow point; blue vectors: actual motion direction when exiting each slow point. Data support the “update” model: distribution of α significantly biased to 0 (Rayleigh test, $p < 10^{-10}$), but not that for β (Rayleigh test, $p > 0.01$). $N = 85$ trajectories, 3 animals on 3 backgrounds.”

Methods:

“Two motion-direction models were distinguished by measuring two angles, α and β , as an animal’s skin pattern moved from a starting pattern (start), through n intermediate slow points (steps) towards an eventual steady-state pattern (goal). α is the angle between the vector connecting step $n-1$ to step n , and the vector connecting step $n-1$ to the goal. β is the angle between the vector connecting step $n-1$ to step n , and that connecting the start to the goal. In the “memory” model, the animal follows the initial direction from start to goal, resulting in both α and β values near 0. In the “update” model, the animal updates the direction it must move to reach the goal every in every step, resulting in α values near 0.”

- Line 146: number of successive low-velocity regions increased as the animal approached its target. Does this suggest the animal just slowed down to choose a specific place to rest?

Why does it have to do with successive error correction steps? Or anything to do with camouflage. Was pattern matching spatially tested statistically to better match the target stimuli?

We apologise for the confusion: here we are not referring to the animal's movement velocity, but remain with velocity in visual-texture space, i.e. how fast the skin pattern changes. We have now updated our text to clarify this point, stating

"The number of successive low-velocity regions increased as the animal skin approached its target pattern"

Improved texture matching is quantified as decreasing distance between the current skin pattern and the background pattern (x-axis of Figure 3d, left to right). While transitioning from one camouflage pattern to another, we observed more slow-down events and longer dwell times as animals approached their target pattern. This suggests that animals compare their current skin pattern with their surroundings and can more quickly recognize a mismatch when the discrepancy is larger.

D. Appropriate use of statistics and treatment of uncertainties

There is a lot of data analysis in this paper. The new figures are helpful. But without a better understanding of the variance between evoked correlations of chromatophore activated spatial patterns and their link to the visual stimuli patterns presented it is very hard to comment on the statistics and uncertainties in this paper as it relates to the control of specific spatial patterns of chromatophore activation. As it stands, there is a good correlation that as visual stimulus granularity increases the skin patterns get significantly more coarse or blocky, however it does not show how reproducible spatial patterns are by each animal and between animals (but this analysis could be done at least on fig 2b data, the legend info is not clear what it is testing). With only three animals there appears some animal specific differences for some checker stimuli. Fig 2c demonstrates the same trend but highlights the extreme diversity in pattern activation within a single animal to simple checkboard stimuli. So the take home is there are not three patterns but a continuum of activation as granularity of the stimulus increases but also skin patterning is unpredictable in many cases (which makes challenges for interpreting the correlations found, which would be high when there data sets are very large). Fig 2 highlights it is hard to review whether

analyses demonstrate the biology of skin control of pattern in relation to stimuli as the paper outlines its new findings.

We quantify the relationship between skin patterns and visual stimuli across animals in Fig. 2 and Extended Data Fig. 4. We now provide additional example images showing how different camouflage patterns are produced on specific backgrounds across animals (Extended Data Fig. 4f,g).

E. Conclusions: robustness, validity, reliability

See above, and it still remains, without the stimuli linked to each figure, and the data analysis performed that highlights the relationship between stimuli and the expressed specific spatial chromatophore patterns (how they differ spatially as a pattern between trials, stimuli and individuals) it is hard to understand if the conclusions are warranted.

We have now added additional stimulus information, and clarified our results on the relationship between camouflage patterns and background stimuli. We would like to point out that much of our paper is concerned with the dynamics of pattern motion, a related but distinct issue from static background matching.

F. Suggested improvements: experiments, data for possible revision

See above.

G. References: appropriate credit to previous work?

The method is new, so no previous comparisons. The older work seems to be referenced well.

H. Clarity and context: lucidity of abstract/summary, appropriateness of abstract, introduction and conclusions

See above.

Reviewer Reports on the Second Revision:

Referees' comments:

Referee #1 (Remarks to the Author):

The authors have addressed all of my concerns. I have enjoyed reviewing the article and appreciate all of the work that has gone into this impressive work.

Referee #2 (Remarks to the Author):

The manuscript by Woo et al. is significantly improved - the authors performed a number of additional analyses to demonstrate that their high dimensional pattern space is likely not an artifact of their network parameterization and the text is now greatly improved. However, the authors did not include any figure legends in this re-submission, which limited my ability to fully assess the new analyses.

The remaining concern I have is the use of "camouflage". In the Discussion (L230), the authors state "camouflage pattern space is high-dimensional". In Extended Data 4f we see examples of the cuttlefish skin patterns, and it looks like they are only camouflaging in some cases. I suggest two small additions:

1. In Extended Data Figure 4f, please include the r^2 values on each image so we can see the concurrence between the observed skin/background matching ability and the authors' calculated correlation.
2. I suggest the authors reword the Abstract and Discussion to say "Cuttlefish skin pattern space is high-dimensional", or "Motor control is high-dimensional", as many or most of the skin patterns exhibited by the cuttlefish in this study don't appear to be camouflage - especially if ED4f is indicative of the data.

Referee #3 (Remarks to the Author):

A. Summary of the key results

Woo et al. present an updated but still dazzling array of data and figures to convince the reader that chromatophore control in cuttlefish in response to visual stimuli is extremely complex (not just three patterns) and that the coarseness of the stimulus patterns correlate with the chromatophore patterns expressed on the skin. They were able to complete this study due to a previous publication demonstrating the method of tracking of 10,000s of chromatophores simultaneously. They present several findings (providing you assume that principle components are a proxy for chromatophore patterns) as new from their study including (1) chromatophore activation patterns are expressed in a vast array of forms, not just three (but published earlier by Osorio et al. 2022 Current Biology),

(2) chromatophore activation pattern matches where not considered to be a reflex (when given time to decide), as the expression changes that occur before they are ultimately expressed are not stereotypical, (3) chromatophore activation patterns changed over time, both before settling on a specific pattern and after, as well as the speed of pattern change altered over time, (4) analyses showed camouflage chromatophore activation patterns were made of subcomponents, some of which overlapped, and (5) different extracted chromatophore activations resulted when different stimuli were used, such as natural, vs checkerboard, vs looming, which they hypothesized was due to different control systems (but the authors did not have any direct evidence to a mechanism of the alternate control systems). This updated manuscript version is better in terms of new justifications provided and improved clarity in figures. But I still have major concerns, see below.

B. Originality and significance: if not novel, please include reference

- The method of tracking chromatophores is not novel and was published in Nature 2018 (<https://doi.org/10.1038/s41586-018-0591-3>).
- The relationship between visual stimuli driving expression of chromatophore activation patterns is not novel, as evidenced by many papers from the labs of Roger Hanlon and Daniel Osorio (See the massive number of papers cited in Hanlon & Messenger 2018. Cephalopod Behaviour. Cambridge University Press) and particularly the new paper from Osorio et al. 2022 Current Biology. These are now all cited. However the new presentation now shows a much better and detailed correlation of stimulus pattern to expressed chromatophore pattern.
- With respect to visuomotor chromatophore feedback (the brain matching chromatophore activation patterns to a scene, without proprioception involvement), cuttlefish cannot produce skin patterns correlated to their environment, without functional eyes. Thus it is not novel that visual information, via feedback (as defined above), is constantly needed to properly express chromatophore patterns, particularly for camouflage. Refs 5, 12 & 13 do not disable the visual system during pattern expression nor does these studies deactivate vision reversibly to demonstrate that feedback is not required for proper chromatophore activation patterns. While previous such papers may state open-loop, they do not show it experimentally, it is just their opinion, probably based on seeing very rapid transformation of camouflage patterns (which can be less than 500 ms conservatively), and electrically stimulating the brain where the chromatophore cell bodies are located. So currently, there has been no way to demonstrate precisely that open- or closed-loop visuomotor chromatophore control exists and the paper should probably only leave feedback as part of the discussion.
- The novelty of this study utilizes a major increase in the spatial and temporal resolution in their tracking method to precisely quantify and show the relationships between visual stimuli and correlated chromatophore activity at the level of single chromatophores (or small patches of chromatophores).
- The significance is they have demonstrated that chromatophore control systems in cuttlefish are very sophisticated (more than previously published). Their analysis has shown, via correlation rather than a novel control mechanism, that cuttlefish given enough time to settle on a pattern, can produce highly tailored camouflage patterning across the dorsal surface with respect to specific visual stimuli.

C. Data & methodology: validity of approach, quality of data, quality of presentation

The tracking method is accurate from what was reported in Nature and this manuscript. It is producing some very detailed information about the control of cuttlefish skin patterns. I really

appreciate the very high detail quantification approach (with even more figures provided) but the manuscript still raises some questions. Unfortunately I still see there are some major problems in the presentation of the research in this manuscript.

- My most major concern is the author's unwillingness to recognize and use the naming conventions in the literature for chromatophore (activation) 'patterns', and instead using novel language such as chromatophore 'texture'. 'Texture' is not correct or appropriate as I pointed out previously, as cuttlefish produce skin texture using papillae (totally different muscles than chromatophores), and texture is already exclusively used for cuttlefish papillae in the literature (and definitely not chromatophores). Furthermore, the authors acknowledge such "papillae texture" produces a confounding signal in their data from the three-dimensional effect produced by papillae expression, and so regions affected by this are removed from the analysis in this paper (lines 668-670). Thus as this paper is about the biological activation of the chromatophores I strongly insist the authors to remove texture throughout when it is being used in conjunction with chromatophore activation and replace it with the word pattern. Such an exchange should be easy without needing to rewrite the sentence in nearly every case. This is not just my opinion, as according to John Messenger (ref 5), Roger Hanlon (ref 1) and Boycott (1961)(just naming a few key papers), chromatophores are neurally activated in patterns. So adding new terminology, as textures, solely due to using a computer science technique described in this manuscript, that discusses texture, is not appropriate in this context. There is no problem with discussing the texture-based analysis method, but for all the biological concerns related to cuttlefish chromatophore activations, it should be solely described as patterns. I note also that pattern is already used in many places throughout the manuscript appropriately with regards to chromatophore expression, thus it is unclear why texture is also used.
- If the title is about cuttlefish chromatophores (rather than papillae) then texture needs to be replaced with pattern. The paper does not measure 3D texture which is only produced by skin and papillae muscles.
- Feedback is used, but without being defined. Plus the paper is only providing input and output information, with the brain and neural systems as a black box. If you are referring to visuomotor feedback, as the chain of neuron activations from receptors to chromatophore muscles, then state this. But there is a bigger issue, as feedback generally requires something about proprioceptive feedback such as from the skin and chromatophores. Is this in your assumption about feedback? How are the cuttlefish even knowing the feedback is altering their chromatophore patterns as they cannot see most of them? Or are you talking about some corollary discharge that the brain sends back to the photoreceptors to tune the output chromatophore patterns? Given the black box approach of the control system in this case, then it would be much better to specify what you mean by feedback and keep this for the discussion.
- I will concede that indeed this manuscript shows strong correlation between the visual stimulus pattern coarseness (granularity used in some papers, extracted PC 1 in this paper) and the chromatophore patterns expressed, as evidenced in Fig 2. But correlation is not causation. Importantly nearly all publications (this manuscript included), have an adaptation phase when cuttlefish are either shown, or put in a tank with, a specific background. And the correlations are showing the static output after viewing a scene for some time. So while it is likely that chromatophore pattern expression is a closed-loop system rather than a reflex open-loop response, your correlation does not prove this (but no harm discussing this). Furthermore, the looming stimulus shows that chromatophore pattern expression can be a repeatable reflex response, demonstrating there may be an open-loop pathway for control but also that we have more to learn

(particularly how this fast and slow decision making is done). And how good are the expressed patterns without the adaptation phase? How much refinement is done over the meandering trajectory of the chromatophore pattern changes?

- Motion does not appear to be the correct term for this paper (Lines 37, 40, 46, 52, 53, 146, 158, 238, 240, 244). (1) The cuttlefish themselves are not moving when the chromatophore patterns are being analyzed (in nearly all cases right?). (2) In addition, the patterns on skin are not drifting, at least not how they are described. Otherwise there would be some discussion of correlated chromatophore neighbor activity in specific directions. (3) Furthermore the analysis is looking at pixel intensity changes, so in this case not movement. It does not appear that individual pixels are tracked over time but rather clusters are analyzed so it does not appear the movement could be determined from the data as presented. I do not see anything wrong with pattern change or pattern trajectories but motion does not seem to be part of the quantification for the three reasons outlined above. In most cases, “motion” should be replaced with “change”.

There are some other minor suggested changes.

- o Line 33-34: “Behavioral experiments also indicate that while camouflage is based on a visually acquired scene, it does not depend on feedback (5,12,13) for its execution”. This is not factually correct as animals without eyes cannot perform camouflage. Even animals temporarily blinded cannot perform camouflage.
- o Line 255: Please provide more details to explain your hypotheses as to why you think blanching control is super-imposed on camouflage control. Is it just the correlations are more stringent? Faster modes of action?
- o Line 263: The expressed pattern outcome is stereotypical but the path to the outcome is not. Right? So this sentence needs more specificity.
- o Line 266: Should be changed to pattern-matching, as stated above to align with previous nomenclature published in many papers about chromatophores. Texture should be exclusively used in cuttlefish when describing the skin surface or in relation to the papillae activation.
- o Fig 4 caption: panels a-c need to indicated it is from a single individual.
- o Fig 4f: Is this from the same animal but two trajectories? Needs to be specified in figure caption.
- o Extended Data Fig 3f: There is chromatophore expansion change prior to movement. Is this an artefact of timing determination, or do cuttlefish consistently and pre-emptively change their patterns prior to moving? If such timing is accurate, this appears to be a novel finding and should be discussed, both in terms of biology but also whether this impacts the data analysis of this manuscript.
- o Extended Data Fig 6d: Description needs to be corrected to indicate difference between trials.

D. Appropriate use of statistics and treatment of uncertainties

Chromatophore spatial “patterns” (not PCs) are not obviously compared between trials within animals, instead they seem to be lumped together. There is no demonstration that data is normally distributed, nor has it been shown that there is no statistical difference between trials and between animals and so they were lumped for the correlation testing. Extended Data Fig 6d indicates there is significant variation between trials which I did not find discussed elsewhere?

I do see that in figure 2, there is correlation generally between deconvolved skin patterns represented as PC1 and stimuli patterns. But I do not see there is statistical comparisons between animals (different colors). What is the correlation for each animal? Do they significantly differ? At the moment Fig 2a is just the extracted raw data. But there are between animal comparisons done in Fig 2b which is great. As there is also a lot of noise, it is important the readers can see where the variance is partitioned between treatments (stimuli) and animals.

How can the reader determine if individuals precisely match stimuli coarseness to skin patterns (i.e. the error of matching)? And how, spatially, are these specific chromatophore activation patterns reproducible within and between animals to a given stimuli? Your response to my previous questions, was that Extended Data Fig 4 addressed this with some text about prediction. But I do not see any computer generated stimuli patterns, and some sort of matching metric to biological patterns. Instead you used GLM (Ext. Fig 4d) to show your previous correlations hold. My understanding is that these GLM models provide some confidence as a prediction that one part of the data will predict another part. If this is central to your paper title, as in there is predictive matching, then it needs to be in the main part of the paper. Given the lack of spatial overlap of components, see Extended Data Figure 7, it is unclear why this method was not combined to emphasize pattern changes relative to stimuli tested. Also given the ranked chromatophores could be mapped back onto the mantle, Extended Data Figure 10, this also suggests there may be a way to convey pattern changes rather than just PC1 components.

Can you please explain if you actually have predicted patterns (rather than predicted principle components)? The problem is that we as reviewers and ultimately the readers cannot tell if the visual stimuli patterns relate to specific skin patterns expressed. Fig 4f-g is just images, no analysis, just a security blanket that your correlations match. Which is great to see but does not show prediction. You have correlations of abstract principle components (PCs) as your 'pattern' data but these are not spatial patterns right, just some output number from image analysis, so I cannot see how there is any matching mechanism demonstrated. The title promises "Visual texture-matching search in camouflaging cuttlefish". But where is the pattern "match"? It is just a correlation using PCs not actual patterns, and sometimes it can be quite poor, plus we do not get a metric of how good a cuttlefish is at pattern matching. Particularly as some animals may be better than others. If the title was "Camouflaging cuttlefish use image statistics to search for pattern-similarities", I could see that your correlation level analysis would be wholly appropriate. But "pattern-matching" implies there is some mechanism, some image coarseness parameter the cuttlefish are visually extracting and correlating to body patterns they express on their skin. But by using principle component analysis you have not isolated what are the actual image statistics the animals are extracting and using.

E. Conclusions: robustness, validity, reliability

The data analysis performed correctly highlights the correlation between stimuli and the expressed specific spatial chromatophore patterns but this is purely as extracted principle components (and does not tease apart the difference between animals for naturalistic stimuli nor differences among repeated stimuli). They did this in a number of ways and the GLM analysis does show some principle component predictions (but not pattern predictions). However, the manuscript as I see it, does not show how specific spatial patterns of chromatophore activations on specific regions of mantle are

turned on by the animal's neural system as they relate to specific visual stimuli. Extended Fig 4a-c does not do this convincingly. Thus it is hard to understand if the "pattern matching" discussed throughout is warranted. Perhaps it should be "image statistics" rather than "pattern matching"?

F. Suggested improvements: experiments, data for possible revision

See above.

G. References: appropriate credit to previous work?

The method is new, so no previous comparisons. The older work seems to be referenced well.

H. Clarity and context: lucidity of abstract/summary, appropriateness of abstract, introduction and conclusions

See above.

**Author Rebuttals to Second Revision:
Responses to the reviewers' comments on Woo, Liang et al., manuscript.**

We thank the reviewers for their detailed comments, which we address below.

Referee #1 (Remarks to the Author):

The authors have addressed all of my concerns. I have enjoyed reviewing the article and appreciate all of the work that has gone into this impressive work.

Thank you very much.

Referee #2 (Remarks to the Author):

The manuscript by Woo et al. is significantly improved - the authors performed a number of additional analyses to demonstrate that their high dimensional pattern space is likely not an artifact of their network parameterization and the text is now greatly improved. However, the authors did not include any figure legends in this re-submission, which limited my ability to fully assess the new analyses.

The remaining concern I have is the use of “camouflage”. In the Discussion (L230), the authors state “camouflage pattern space is high-dimensional”. In Extended Data 4f we see examples of the cuttlefish skin patterns, and it looks like they are only camouflaging in some cases. I suggest two small additions:

1. In Extended Data Figure 4f, please include the r^2 values on each image so we can see the concurrence between the observed skin/background matching ability and the authors' calculated correlation.

We have added the correlation coefficient to each image as requested.

2. I suggest the authors reword the Abstract and Discussion to say “Cuttlefish skin pattern space is high-dimensional”, or “Motor control is high-dimensional”, as many or most of the skin patterns exhibited by the cuttlefish in this study don't appear to be camouflage - especially if ED4f is indicative of the data.

Thank you for this point, we have changed to ‘skin pattern space’.

Referee #3 (Remarks to the Author):

A. Summary of the key results

Woo et al. present an updated but still dazzling array of data and figures to convince the reader that chromatophore control in cuttlefish in response to visual stimuli is extremely complex (not just three patterns) and that the coarseness of the stimulus patterns correlate with the chromatophore patterns expressed on the skin. They were able to complete this study due to a previous publication demonstrating the method of tracking of 10,000s of chromatophores simultaneously. They present several findings (providing you assume that principle components are a proxy for chromatophore patterns) as new from their study including (1) chromatophore activation patterns are expressed in a vast array of forms, not just three (but published earlier by Osorio et al. 2022 Current Biology), (2) chromatophore activation pattern matches where not considered to be a reflex (when given time to decide), as the expression changes that occur before they are ultimately expressed are not stereotypical, (3) chromatophore activation patterns changed over time, both before settling on a specific pattern and after, as well as the speed of pattern change altered over time, (4) analyses showed camouflage chromatophore activation patterns were made of subcomponents, some of which overlapped, and (5) different extracted chromatophore activations resulted when different stimuli were used, such as natural, vs checkerboard, vs looming, which they hypothesized was due to different control systems (but the authors did not have any direct evidence to a mechanism of the alternate control systems). This updated manuscript version is better in terms of new justifications provided and improved clarity in figures. But I still have major concerns, see below.

B. Originality and significance: if not novel, please include reference

- The method of tracking chromatophores is not novel and was published in Nature 2018 (<https://doi.org/10.1038/s41586-018-0591-3>).

This comment is not entirely correct. The data-processing pipelines have been significantly modified from our previous work and the image data are from an array of 15-24 HR cameras plus one low-R camera. Also, our analysis of low-R camera data has no parallel in the 2018 study. These are very significant methodological changes, not emphasised in the main text which focuses instead on the biology, but described in detail in the methods section.

- The relationship between visual stimuli driving expression of chromatophore activation patterns is not novel, as evidenced by many papers from the labs of Roger Hanlon and Daniel Osorio (See the massive number of papers cited in Hanlon & Messenger 2018. Cephalopod Behaviour. Cambridge University Press) and particularly the new paper from Osorio et al. 2022 Current Biology. These are now all cited. However the new presentation now shows a much better and detailed correlation of stimulus pattern to expressed chromatophore pattern.

Thank you.

- With respect to visuomotor chromatophore feedback (the brain matching chromatophore activation patterns to a scene, without proprioception involvement), cuttlefish cannot produce skin patterns correlated to their environment, without functional eyes. Thus it is not novel that visual information, via feedback (as defined above), is constantly needed to properly express chromatophore patterns, particularly for camouflage. Refs 5, 12 & 13 do not disable the visual system during pattern expression nor does these studies deactivate vision reversibly to demonstrate that feedback is not required for proper chromatophore activation patterns. While previous such papers may state open-loop, they do not show it experimentally, it is just their opinion, probably based on seeing very rapid transformation of camouflage patterns (which can be less than 500 ms conservatively), and electrically stimulating the brain where the chromatophore cell bodies are located. So currently, there

has been no way to demonstrate precisely that open- or closed-loop visuomotor chromatophore control exists and the paper should probably only leave feedback as part of the discussion.

- The novelty of this study utilizes a major increase in the spatial and temporal resolution in their tracking method to precisely quantify and show the relationships between visual stimuli and correlated chromatophore activity at the level of single chromatophores (or small patches of chromatophores).
- The significance is they have demonstrated that chromatophore control systems in cuttlefish are very sophisticated (more than previously published). Their analysis has shown, via correlation rather than a novel control mechanism, that cuttlefish given enough time to settle on a pattern, can produce highly tailored camouflage patterning across the dorsal surface with respect to specific visual stimuli.

Thank you very much.

C. Data & methodology: validity of approach, quality of data, quality of presentation

The tracking method is accurate from what was reported in Nature and this manuscript. It is producing some very detailed information about the control of cuttlefish skin patterns. I really appreciate the very high detail quantification approach (with even more figures provided) but the manuscript still raises some questions. Unfortunately I still see there are some major problems in the presentation of the research in this manuscript.

- My most major concern is the author's unwillingness to recognize and use the naming conventions in the literature for chromatophore (activation) 'patterns', and instead using novel language such as chromatophore 'texture'. 'Texture' is not correct or appropriate as I pointed out previously, as cuttlefish produce skin texture using papillae (totally different muscles than chromatophores), and texture is already exclusively used for cuttlefish papillae in the literature (and definitely not chromatophores). Furthermore, the authors acknowledge such "papillae texture" produces a confounding signal in their data from the three-dimensional effect produced by papillae expression, and so regions affected by this are removed from the analysis in this paper (lines 668-670). Thus as this paper is about the biological activation of the chromatophores I strongly insist the authors to remove texture throughout when it is being used in conjunction with chromatophore activation and replace it with the word pattern. Such an exchange should be easy without needing to rewrite the sentence in nearly every case. This is not just my opinion, as according to John Messenger (ref 5), Roger Hanlon (ref 1) and Boycott (1961)(just naming a few key papers), chromatophores are neurally activated in patterns. So adding new terminology, as textures, solely due to using a computer science technique described in this manuscript, that discusses texture, is not appropriate in this context. There is no problem with discussing the texture-based analysis method, but for all the biological concerns related to cuttlefish chromatophore activations, it should be solely described as patterns. I note also that pattern is already used in many places throughout the manuscript appropriately with regards to chromatophore expression, thus it is unclear why texture is also used.

- If the title is about cuttlefish chromatophores (rather than papillae) then texture needs to be replaced with pattern. The paper does not measure 3D texture which is only produced by skin and papillae muscles.

Our initial disagreement on ‘visual texture’ vs. ‘pattern’ stems from different uses of the term “texture” in different fields. “Texture” (referring to 2D statistics) is indeed routine in human and non-human primate visual psychophysics and in machine vision. To satisfy the reviewer, however, we have clarified our use of the terms in introduction (see below), and changed the wording from ‘texture’ to ‘pattern’ throughout the manuscript (including the title) when referring to cuttlefish skin patterns, retaining ‘texture’ when discussing backgrounds or computational models as suggested:

New paragraph in introduction: *“Cephalopod camouflage consists of matching the animal’s appearance to that of its substrate and typically contains two- and three-dimensional components. Although both components are technically textural^{4,15,16}, in this field the term “texture” is often applied only to 3-D features, caused for instance by the contraction of skin papillae^{5,17}. We studied here the 2-D features of camouflage and thus refer to them as (skin) patterns and to the process as pattern matching. Pattern matching does not consist of a faithful reproduction of the substrate’s appearance but rather of the visually initiated statistical estimation and generation of that appearance⁵.”*

• Feedback is used, but without being defined. Plus the paper is only providing input and output information, with the brain and neural systems as a black box. If you are referring to visuomotor feedback, as the chain of neuron activations from receptors to chromatophore muscles, then state this. But there is a bigger issue, as feedback generally requires something about proprioceptive feedback such as from the skin and chromatophores. Is this in your assumption about feedback? How are the cuttlefish even knowing the feedback is altering their chromatophore patterns as they cannot see most of them? Or are you talking about some corollary discharge that the brain sends back to the photoreceptors to tune the output chromatophore patterns? Given the black box approach of the control system in this case, then it would be much better to specify what you mean by feedback and keep this for the discussion.

We use ‘feedback’ in the general sense, that is as referring to closed-loop rather than open-loop control. Our results suggest the existence of closed-loop control. This feedback could indeed be proprioceptive, visual, or via corollary discharge or efference copy. Those possibilities are now mentioned and discussed at the end of the discussion section. This new paragraph reads:

“... Where could such feedback originate from? A first possibility is proprioceptors in or around each chromatophore. There is, so far, no evidence for such proprioceptors around cephalopod chromatophores⁵. A second possibility is that cuttlefish use vision to assess the match between their immediate skin-patterning output and the background, for example during each low-velocity segment in pattern-space motion. This could be tested by masking the animal’s skin during camouflaging. A third possibility is efference copy of the motor command to the chromatophore array. This would require the existence of appropriate motor collaterals (not described so far), some calibration of the copy and some form of integrator so that the copy accurately represents the true generated output. Our results will inform the mechanistic studies required to understand this remarkable system.”

- I will concede that indeed this manuscript shows strong correlation between the visual stimulus pattern coarseness (granularity used in some papers, extracted PC 1 in this paper) and the chromatophore patterns expressed, as evidenced in Fig 2. But correlation is not causation. Importantly nearly all publications (this manuscript included), have an adaptation phase when cuttlefish are either shown, or put in a tank with, a specific background. And the correlations are showing the static output after viewing a scene for some time. So while it is likely that chromatophore pattern expression is a closed-loop system rather than a reflex open-loop response, your correlation does not prove this (but no harm discussing this).

As the reviewer writes, figure 2 does not concern the open- vs closed-loop control issue. Instead it shows the ability of cuttlefish to finely modulate their camouflage patterns to match a range of artificial and natural backgrounds.

Furthermore, the looming stimulus shows that chromatophore pattern expression can be a repeatable reflex response, demonstrating there may be an open-loop pathway for control but also that we have more to learn (particularly how this fast and slow decision making is done).

Figure 5 indeed highlights the fact that the dynamics of skin pattern change observed in response to looming stimuli and in camouflaging animals are very different. We discuss the consequences and possible interpretations of these quantitative observations on the neural control systems for these two different behaviours.

And how good are the expressed patterns without the adaptation phase?

This question is addressed in Figure 3, in which we describe the trajectories of camouflage pattern change.

How much refinement is done over the meandering trajectory of the chromatophore pattern changes?

This was also described in Fig. 3. In this revision, we have added a new panel (Fig 3e, below) to better illustrate this point, plotting the increased correlation between skin pattern and background (i.e., the progressive refinement of the match) over the successive steps of the approach to the chosen pattern.

New methods text: *In Fig. 3e (above insert), we calculated for each step (i.e., at each local minimum of pattern motion velocity) the correlation between the skin pattern at that time and the background, in the space defined by PCs 1-50. The difference between this instantaneous correlation and that measured at behaviour onset was then averaged across all trials analysed above.*

• Motion does not appear to be the correct term for this paper (Lines 37, 40, 46, 52, 53, 146, 158, 238, 240, 244). (1) The cuttlefish themselves are not moving when the chromatophore patterns are being analyzed (in nearly all cases right?). (2) In addition, the patterns on skin are not drifting, at least not how they are described. Otherwise there would be some discussion of correlated chromatophore neighbor activity in specific directions. (3) Furthermore the analysis is looking at pixel intensity changes, so in this case not movement. It does not appear that individual pixels are tracked over time but rather clusters are analyzed so it does not appear the movement could be determined from the data as presented. I do not see anything wrong with pattern change or pattern trajectories but motion does not seem to be part of the quantification for the three reasons outlined above. In most cases, “motion” should be replaced with “change”.

We wish to retain ‘motion’, as it is established terminology in the neuroscience and dynamical systems literature (broadly speaking, ‘motion’ is not restricted to positional changes of objects in the physical world). In discussing skin-pattern space, the process of evolving from one pattern to another indeed corresponds to motion in that space. We also think that referring to ‘change in skin-pattern space’ is inappropriate, for it could be misinterpreted as a modification of the space itself rather than as motion within that space.

There are some other minor suggested changes.

o Line 33-34: “Behavioral experiments also indicate that while camouflage is based on a visually acquired scene, it does not depend on feedback (5,12,13) for its execution”. This is not factually correct as animals without eyes cannot perform camouflage. Even animals temporarily blinded cannot perform camouflage.

There is a misunderstanding: it is clear that the initiation of a camouflage requires vision. This is indeed what we say: “camouflage is based on a visually acquired scene”. What we mean when

mentioning feedback is whether feedback is necessary for the realisation of the behaviour, after the initiation of the behaviour. This is the same issue as with throwing or hitting a baseball: identifying the target requires vision, but the process of throwing or hitting, once the goal has been identified, runs in open loop, i.e., without the need for feedback. This is the question that we address and our results (velocity and tortuosity profiles) do suggest that feedback is required during camouflage, but likely not during blanching. The issue of feedback is thus now discussed in the last paragraph of the discussion (as quoted above).

o Line 255: Please provide more details to explain your hypotheses as to why you think blanching control is super-imposed on camouflage control. Is it just the correlations are more stringent? Faster modes of action?

In describing our hypothesis we now write:

“The contrast between the repeatability of skin-pattern restoration following blanching (Fig. 5k,l) and the variability of camouflage pattern composition (Fig. 4f-i) supports the hypothesis that camouflage and blanching are under different control systems.”

o Line 263: The expressed pattern outcome is stereotypical but the path to the outcome is not. Right? So this sentence needs more specificity.

We have modified this sentence to read:

“In conclusion, camouflage control in Sepia appears to be both extremely flexible and to follow non stereotypical paths when analysed at cellular resolution.”

o Line 266: Should be changed to pattern-matching, as stated above to align with previous nomenclature published in many papers about chromatophores. Texture should be exclusively used in cuttlefish when describing the skin surface or in relation to the papillae activation.

The line now reads:

“Our results will inform the mechanistic studies required to understand this remarkable system.”

o Fig 4 caption: panels a-c need to indicated it is from a single individual.

Thank you: now added.

*“a. ... (1 representative animal from 3 analysed)”
b. Of the 12 components segmented in a, ...
c. ... averaged activity shown as red and blue traces in top row of b) ...”*

o Fig 4f: Is this from the same animal but two trajectories? Needs to be specified in figure caption.

Thank you: now specified, it now reads:

“Trajectories (projected onto the same PC1-PC2 plane as a) and segmented images of two similar pattern transitions (teal: 87.2 s; pink: 87.1 s) of an animal responding to the same background switch (N13 to N29) as a.”

o Extended Data Fig 3f: There is chromatophore expansion change prior to movement. Is this an artefact of timing determination, or do cuttlefish consistently and pre-emptively change their patterns prior to moving? If such timing is accurate, this appears to be a novel finding and should be discussed, both in terms of biology but also whether this impacts the data analysis of this manuscript.

We indeed often observed skin pattern changes slightly before animal movement. We also observed pattern changes without movement, especially when background stimuli were changed (EDF. 3g). We found no difference in the time course of skin pattern change between these two conditions, as shown in EDF. 3h. The detailed relationship between animal motion and skin pattern change represents an interesting direction for future work.

o Extended Data Fig 6d: Description needs to be corrected to indicate difference between trials.

Thank you: We have added lines to EDF6d to indicate which trials were of the same types as trial #0.

D. Appropriate use of statistics and treatment of uncertainties

Chromatophore spatial “patterns” (not PCs) are not obviously compared between trials within animals, instead they seem to be lumped together. There is no demonstration that data is normally distributed, nor has it been shown that there is no statistical difference between trials and between animals and so they were lumped for the correlation testing. Extended Data Fig 6d indicates there is significant variation between trials which I did not find discussed elsewhere?

We describe the dynamics of skin pattern change between trials within animals in Fig. 3 and 4. Principal component analysis is an appropriate and routinely used technique with high-dimensional data of this type. Tests of normality are not appropriate in this context. EDF 6D refers to the rearrangement of chromatophore use over multiple trials. We discuss this significant variation at length in Fig. 4, Extended Data Fig. 6 and 7, and contrast it with the reaction to visual threat displays in Fig. 5 and Extended Data Fig. 9.

I do see that in figure 2, there is correlation generally between deconvolved skin patterns represented as PC1 and stimuli patterns. But I do not see there is statistical comparisons between animals (different colors). What is the correlation for each animal? Do they significantly differ? At the moment Fig 2a is just the extracted raw data. But there are between animal comparisons done in Fig 2b which is great. As there is also a lot of noise, it is important the readers can see where the variance is partitioned between treatments (stimuli) and animals.

We provide Pearson's r value for each animal in the caption of figure 2a.

How can the reader determine if individuals precisely match stimuli coarseness to skin patterns (i.e. the error of matching)? And how, spatially, are these specific chromatophore activation patterns reproducible within and between animals to a given stimuli? Your response to my previous questions, was that Extended Data Fig 4 addressed this with some text about prediction. But I do not see any computer generated stimuli patterns, and some sort of matching metric to biological patterns. Instead you used GLM (Ext. Fig 4d) to show your previous correlations hold. My understanding is that these GLM models provide some confidence as a prediction that one part of the data will predict another part. If this is central to your paper title, as in there is predictive matching, then it needs to be in the main part of the paper. Given the lack of spatial overlap of components, see Extended Data Figure 7, it is unclear why this method was not combined to emphasize pattern changes relative to stimuli tested. Also given the ranked chromatophores could be mapped back onto the mantle, Extended Data Figure 10, this also suggests there may be a way to convey pattern changes rather than just PC1 components.

The different statistical and visualisation methods used in Fig. 2a, EDFig. 4d-e and Fig. 2b were intentional, and motivated by differences in experimental designs and objectives.

As the reviewer rightly stated above, Fig. 2a presents a *correlation* analysis between two dependent variables, namely the visual texture of the background and the skin (pattern). While the 30 sampled natural images served as stimuli for studying camouflage behaviour, the visual texture of the background patches around the animal (x-axis) were not the independent variable in the experimental design. This figure demonstrates a correlation between background visual texture and skin visual texture (pattern). We provide the correlation coefficient for each animal in the legend.

EDFig. 4d-e describes a modelling experiment using the natural background data from Fig. 2a. We use the GLM as a method to compare the ability of different, or combinations of different texture and image statistics in modelling a stimulus-response relationship between the background and skin pattern, assuming that such a relationship exists. These figures demonstrate the superiority of the presented pattern/texture representation in describing the observed camouflage behaviour.

Our findings with natural backgrounds led to the experiment with artificial backgrounds (checkerboards) and to testing explicitly the relationship between background and skin pattern. We performed regression analysis in Fig. 2b, with the generated logarithmic spatial series as the independent variable (x-axis), and the measured skin pattern as the dependent variable (y-axis). This figure not only demonstrates a causal relationship but also the response sensitivity in texture/pattern space. We provide the r^2 value for each animal, which indicates the predictive power of the regression models.

Together with refinement dynamics shown in Figure 3, we believe that the use of "matching" is indeed warranted.

Can you please explain if you actually have predicted patterns (rather than predicted principle components)? The problem is that we as reviewers and ultimately the readers cannot tell if the visual stimuli patterns relate to specific skin patterns expressed. Fig 4f-g is just images, no analysis, just a security blanket that your correlations match. Which is great to see but does not show prediction. You have correlations of abstract principle components (PCs) as your 'pattern' data but these are not spatial patterns right, just some output number from image analysis, so I cannot see how there is any matching mechanism demonstrated. The title promises "Visual texture-matching search in camouflaging cuttlefish". But where is the pattern "match"? It is just a correlation using PCs not actual patterns, and sometimes it can be quite poor, plus we do not get a metric of how good a cuttlefish is at pattern matching. Particularly as some animals may be better than others. If the title was "Camouflaging cuttlefish use image statistics to search for pattern-similarities", I could see that your correlation level analysis would be wholly appropriate. But "pattern-matching" implies there is some mechanism, some image coarseness parameter the cuttlefish are visually extracting and correlating to body patterns they express on their skin. But by using principle component analysis you have not isolated what are the actual image statistics the animals are extracting and using.

Quantifying pattern matching requires a metric. Any such metric is a function of image data, that is, an image statistic. We tested many such statistics, some more complex than others. We agree that it would be satisfying if the statistics that did the best job of describing pattern matching were easily or instinctively interpretable by humans. Indeed, the current tradeoff between predictive power and interpretability is a widespread issue within visual neuroscience. But for our purpose here, which is to describe the dynamics of pattern matching, the most important issue is to use a good metric for pattern matching. We tested many, and use the best ones.

E. Conclusions: robustness, validity, reliability

The data analysis performed correctly highlights the correlation between stimuli and the expressed specific spatial chromatophore patterns but this is purely as extracted principle components (and does not tease apart the difference between animals for naturalistic stimuli nor differences among repeated stimuli). They did this in a number of ways and the GLM analysis does show some principle component predictions (but not pattern predictions). However, the manuscript as I see it, does not show how specific spatial patterns of chromatophore activations on specific regions of mantle are turned on by the animal's neural system as they relate to specific visual stimuli. Extended Fig 4a-c does not do this convincingly. Thus it is hard to understand if the "pattern matching" discussed throughout is warranted. Perhaps it should be "image statistics" rather than "pattern matching"?

As discussed above, measuring pattern matching requires a metric, and this metric is by definition an image statistic. We write now in Methods:

"This skin-pattern representation can be interpreted as a metric that captures textural information using 512 variables derived objectively from the visual world."

F. Suggested improvements: experiments, data for possible revision
See above.

G. References: appropriate credit to previous work?
The method is new, so no previous comparisons. The older work seems to be referenced well.

H. Clarity and context: lucidity of abstract/summary, appropriateness of abstract, introduction and conclusions
See above.

Reviewer Reports on the Third Revision:

Referees' comments:

Referee #3 (Remarks to the Author):

The authors have addressed the grand majority of my concerns since the previous version and the manuscript is now much improved. It is excellent to see the focus on the biology of the animals (rather than the methods) and read about how cuttlefish are able to perform some intricate camouflage. I have some minor suggestions/comments remaining to help the manuscript appeal to a greater audience.

- It was excellent to read that the data-processing pipelines had been significantly modified and analysis of low-R camera data had no parallel. I suggest these upgrades are highlighted in the main text. On line 73, "Here, we use these techniques" could be embellished to be "Here we update these techniques and build new analysis pipelines to".
- The new panel (Fig 3e) was a fabulous illustration of the camouflage refinement over the successive steps of the approach to the chosen pattern. Great stuff. Might be worth highlighting this in the discussion if not already.
- Motion in pattern tuning over time: As this is established in the neuroscience literature, then it would be ideal to introduce this concept early in the manuscript and highlight a reference to lead the reader to a more detailed understanding of this concept if they are interested. I did not find one in manuscript.
- Line 81: "a dense sampling of the animal's generative repertoire" This is a little abstract. Maybe something like "a dense videography sampling of the animal's generative skin response repertoire"
- Line 93. Add some clarification about the new methods, perhaps put at the end of the sentence "..... that had not previously been used for cuttlefish camouflage analysis."
- Line 220: "camouflage and blanching are under different control systems." There is no evidence for different systems, so need to change to "camouflage and blanching are controlled differently". There is only correlation level support here and so no way to disprove that the same control system can be triggered by the brain in different ways. While this is a subtle point, it is worth considering. Also links to the control flexibility argument you make further down.
- Line 231: replace textures with patterns.
- Line 251: components. "This suggests that blanching is superimposed on camouflage". Again there is no evidence of chromatophore control tiers in the data. Much more suitable would be "This suggests blanching is controlled differently to camouflage".

- I am not going to comment any further on the analysis. Most statistical testing needs normality testing, even tests from PCA. It is not clear from what is written that normality testing is needed or not, so perhaps where appropriate this should be highlighted why not. Also there is no causal relationships shown in the manuscript as you are doing a regression analysis and correlation. You have not disrupted the control systems in these experiments (physically or psychophysically) to separate how information flows among known neuron pathways. So I do not see how you can demonstrate causality. You do have strong correlation evidence though.
- Figure 2b. Can you please add the r^2 for each animal like you did for Fig 2a.
- Strictly speaking the manuscript has only shown “pattern correlating”, not “pattern matching” as pattern matching would have required some demonstration of similarities in the spatial frequencies of the background and skin pattern (which while your measures attempt to get at, they do not provide specific numbers). I do hope and very much look forward to read soon that you are able to show this in your next paper.

Author Rebuttals to Third Revision:

Responses to the reviewers' comments on Woo, Liang et al., manuscript.

We thank the reviewer for their detailed comments, which we address below.

Referee #3 (Remarks to the Author):

The authors have addressed the grand majority of my concerns since the previous version and the manuscript is now much improved. It is excellent to see the focus on the biology of the animals (rather than the methods) and read about how cuttlefish are able to perform some intricate camouflage. I have some minor suggestions/comments remaining to help the manuscript appeal to a greater audience.

Thank you very much.

- It was excellent to read that the data-processing pipelines had been significantly modified and analysis of low-R camera data had no parallel. I suggest these upgrades are highlighted in the main text. On line 73, "Here, we use these techniques" could be embellished to be "Here we update these techniques and build new analysis pipelines to".

Thank you. This sentence now reads:

"Here, we improved on these techniques and reported a new complementary analysis to describe quantitatively the space, dynamics and reliability of camouflage patterns and through this, gain insights into its control system."

- The new panel (Fig 3e) was a fabulous illustration of the camouflage refinement over the successive steps of the approach to the chosen pattern. Great stuff. Might be worth highlighting this in the discussion if not already.

Thank you. We now added the following line in Discussion:

"The correlation between skin and background patterns increased as the number of pattern-motion steps increased."

- Motion in pattern tuning over time: As this is established in the neuroscience literature, then it would be ideal to introduce this concept early in the manuscript and highlight a reference to lead the reader to a more detailed understanding of this concept if they are interested. I did not find one in manuscript.

Thank you. As suggested by Referee #2, we have added references pointing to relevant neuroscience literature:

"Together these results suggest that camouflage relies on feedback during the approach to an adaptive pattern, more akin to correction of hand reaching movements in primates^{33,34} or of tongue reaching in rodents³⁵, than to ballistic motion to a memorised target."

- Line 81: “a dense sampling of the animal’s generative repertoire” This is a little abstract. Maybe something like “a dense videography sampling of the animal’s generative skin response repertoire”

We modified this sentence, which now reads:

“Using natural and artificial 2-D backgrounds (Methods, Extended Data Fig. 1,2), we acquired a dense videographic sampling of the animal’s generative pattern repertoire and analysed motion within skin-pattern space (Fig 1a).”

- Line 93. Add some clarification about the new methods, perhaps put at the end of the sentence “..... that had not previously been used for cuttlefish camouflage analysis.”

We added the following line in Methods:

“To our knowledge, this method has not been previously used to study cuttlefish camouflage.”

- Line 220: “camouflage and blanching are under different control systems.” There is no evidence for different systems, so need to change to “camouflage and blanching are controlled differently”. There is only correlation level support here and so no way to disprove that the same control system can be triggered by the brain in different ways. While this is a subtle point, it is worth considering. Also links to the control flexibility argument you make further down.

We changed this line to now read:

“... supports the hypothesis that camouflage and blanching are under differential control.”

- Line 231: replace textures with patterns.

Here, we would retain the term “visual texture” when discussing the representation of the backgrounds.

- Line 251: components. “This suggests that blanching is superimposed on camouflage”. Again there is no evidence of chromatophore control tiers in the data. Much more suitable would be “This suggests blanching is controlled differently to camouflage”.

We changed this line to the following:

“This suggests that blanching co-occurs with camouflage.”

- I am not going to comment any further on the analysis. Most statistical testing needs normality testing, even tests from PCA. It is not clear from what is written that normality testing is needed or not, so perhaps where appropriate this should be highlighted why not. Also there is no causal relationships shown in the manuscript as you are doing a regression analysis and correlation. You have not disrupted the control systems in these experiments (physically or psychophysically) to separate how information flows among known neuron

pathways. So I do not see how you can demonstrate causality. You do have strong correlation evidence though.

We appreciate the recognition of the strength of the evidence reported. We will respectfully refrain from further discussion on the statistical inference of causality, as it was not central to the manuscript. We performed correlation and regression analyses, interpreted them accordingly and provided the relevant statistics.

- Figure 2b. Can you please add the r^2 for each animal like you did for Fig 2a.

The r and r^2 values of Fig. 2a and 2b are found in their respective legends.

- Strictly speaking the manuscript has only shown “pattern correlating”, not “pattern matching” as pattern matching would have required some demonstration of similarities in the spatial frequencies of the background and skin pattern (which while your measures attempt to get at, they do not provide specific numbers). I do hope and very much look forward to read soon that you are able to show this in your next paper.

Thank you.